# ClimSim: A large multi-scale dataset for hybrid physics-ML climate emulation

**Sungduk Yu**[1]*, **Walter M. Hannah**[2], **Liran Peng**[1], **Jerry Lin**[1], **Mohamed Aziz Bhouri**[3], **Ritwik Gupta**[4], **Björn Lütjens**[5], **Justus C. Will**[1], **Gunnar Behrens**[6], **Julius J. M. Busecke**[3], **Nora Loose**[7], **Charles Stern**[3], **Tom Beucler**[8], **Bryce E. Harrop**[9], **Benjamin R. Hillman**[10], **Andrea M. Jenney**[1,11], **Savannah L. Ferretti**[1], **Nana Liu**[1], **Anima Anandkumar**[12], **Noah D. Brenowitz**[12], **Veronika Eyring**[6], **Nicholas Geneva**[12], **Pierre Gentine**[3], **Stephan Mandt**[1], **Jaideep Pathak**[12], **Akshay Subramaniam**[12], **Carl Vondrick**[3], **Rose Yu**[13], **Laure Zanna**[14], **Tian Zheng**[3], **Ryan P. Abernathey**[3], **Fiaz Ahmed**[15], **David C. Bader**[2], **Pierre Baldi**[1], **Elizabeth A. Barnes**[16], **Christopher S. Bretherton**[17], **Peter M. Caldwell**[2], **Wayne Chuang**[3], **Yilun Han**[18], **Yu Huang**[3], **Fernando Iglesias-Suarez**[6], **Sanket Jantre**[19], **Karthik Kashinath**[12], **Marat Khairoutdinov**[20], **Thorsten Kurth**[12], **Nicholas J. Lutsko**[13], **Po-Lun Ma**[9], **Griffin Mooers**[1], **J. David Neelin**[15], **David A. Randall**[16], **Sara Shamekh**[3], **Mark A. Taylor**[10], **Nathan M. Urban**[19], **Janni Yuval**[5], **Guang J. Zhang**[13], **Michael S. Pritchard**[1,12]

[1]UCI, [2]LLNL, [3]Columbia, [4]UCB, [5]MIT, [6]DLR, [7]Princeton, [8]UNIL, [9]PNNL, [10]SNL, [11]OSU, [12]NVIDIA, [13]UCSD, [14]NYU, [15]UCLA, [16]CSU, [17]Allen AI, [18]Tsinghua, [19]BNL, [20]SUNY

## Abstract

Modern climate projections lack adequate spatial and temporal resolution due to computational constraints. A consequence is inaccurate and imprecise predictions of critical processes such as storms. Hybrid methods that combine physics with machine learning (ML) have introduced a new generation of higher fidelity climate simulators that can sidestep Moore's Law by outsourcing compute-hungry, short, high-resolution simulations to ML emulators. However, this hybrid ML-physics simulation approach requires domain-specific treatment and has been inaccessible to ML experts because of lack of training data and relevant, easy-to-use workflows. We present ClimSim, the largest-ever dataset designed for hybrid ML-physics research. It comprises multi-scale climate simulations, developed by a consortium of climate scientists and ML researchers. It consists of 5.7 billion pairs of multivariate input and output vectors that isolate the influence of locally-nested, high-resolution, high-fidelity physics on a host climate simulator's macro-scale physical state.

The dataset is global in coverage, spans multiple years at high sampling frequency, and is designed such that resulting emulators are compatible with downstream coupling into operational climate simulators. We implement a range of deterministic and stochastic regression baselines to highlight the ML challenges and their scoring. The data (https://huggingface.co/datasets/LEAP/ClimSim_high-res[2]) and code (https://leap-stc.github.io/ClimSim) are released openly to support the development of hybrid ML-physics and high-fidelity climate simulations for the benefit of science and society.

---

*Corresponding author: sungduk@uci.edu

[2]Also available in a low-resolution version (https://huggingface.co/datasets/LEAP/ClimSim_low-res) and an aquaplanet version (https://huggingface.co/datasets/LEAP/ClimSim_low-res_aqua-planet).

37th Conference on Neural Information Processing Systems (NeurIPS 2023) Track on Datasets and Benchmarks.

# 1 Introduction

## 1.1 Overview

Predictions from numerical physical simulations are the primary tool informing policy on climate change. However, current climate simulators poorly represent cloud and extreme rainfall physics [1, 2] despite stretching the limits of the world's most powerful supercomputers. The complexity of the Earth system imposes significant restrictions on the spatial resolution we can use in these simulations [3]. Physics occurring on scales smaller than the temporal and/or spatial resolutions of climate simulations are commonly represented using empirical mathematical representations called "parameterizations". Unfortunately, assumptions in these parameterizations often lead to errors that can grow into inaccuracies in the future predicted climate.

Machine learning (ML) is an attractive approach to emulate the complex nonlinear sub-resolution physics—processes occurring on scales smaller than the resolution of the climate simulator—at a lower computational complexity. Their implementation has the exciting possibility of resulting in climate simulations that are both cheaper and more accurate than they currently are [4, 5]. Current climate simulators have a typical smallest resolvable scale of 80–200 km, equivalent to the size of a typical U.S. county. However, accurately representing cloud formation requires a resolution of 100 m or finer, demanding six orders of magnitude increase in computational intensity. Exploiting ML remains a conceivable solution to sidestep the limitations of classical computing [5]: resulting hybrid-ML climate simulators combine traditional numerical methods—which solve the equations governing large-scale fluid motions of Earth's atmosphere—with ML emulators of the macro-scale effects of small-scale physics. Instead of relying on heuristic assumptions about these small-scale processes, the emulators learn directly from data generated by short-duration, high-resolution simulations [4, 6–18]. The task is essentially a regression problem: in the climate simulation, an ML parameterization emulator returns the large-scale outputs—changes in wind, moisture, or temperature—that occur due to unresolved small-scale (sub-resolution) physics, given large-scale resolved inputs (e.g., temperature, wind velocity; see Section 4).

While several proofs of concept have emerged in recent years, hybrid-ML climate simulators have yet to be advanced to operational use. Obtaining sufficient training data is a major challenge impeding interest from the ML community. This data must contain all macro-scale variables that regulate the behavior of sub-resolution physics and be compatible with downstream hybrid ML-climate simulations. Addressing this using training data from uniformly high-resolution simulations has proven to be very expensive and can lead to issues when coupled to a host climate simulation.

A promising solution is to utilize multi-scale climate simulation methods to generate training data. Crucially, these provide a clean interface between the emulated high-resolution physics and the host climate simulator's planetary-scale dynamics [19]. In theory, this makes downstream hybrid coupled simulation approachable and tractable. In practice, the full potential of multi-scale methods remains largely untapped due to a scarcity of existing datasets, exacerbated by the combination of operational simulation code complexity and the need for domain expertise in choosing variables.

We introduce ClimSim, the largest and most physically comprehensive dataset for training ML emulators of atmospheric storms, clouds, turbulence, rainfall, and radiation for use in hybrid-ML climate simulations. ClimSim is a comprehensive collection of inputs and outputs from physical climate simulations using the multi-scale method. ClimSim was prepared by atmospheric scientists and climate simulator developers to lower the barriers to entry for ML experts on this important problem. Our benchmark dataset serves as a foundation for developing robust frameworks that emulate parameterizations for cloud and extreme rainfall physics, and their interaction with other sub-resolution processes. These frameworks enable online coupling within the host coarse-resolution climate simulator, ultimately improving the performance and accuracy of climate simulators used for long-term projections.

## 1.2 Concepts and Terminology from Earth Science

**Convective Parameterization:** In atmospheric science, "convection" refers to storm cloud and rain development, as well as the associated turbulent air motions. Convective parameterizations represent the integrated effects of these processes, such as the vertical transport of heat, moisture, and momentum within the atmosphere, and condensational heating and drying, on the temporal and spatial

scale of the host climate simulator [20–22]. Stochastic parameterizations represent sub-resolution ("sub-grid scale" in the terminology of Earth science) effects as stochastic processes, dependent on grid-scale variable inputs [23, 24] to capture variations arising from sub-grid scale dynamics.

**Multi-Scale Climate Simulators:** Multi-scale climate simulation is a technique that represents convection without a convective parameterization, by deploying a smaller-scale, high-resolution cloud-resolving simulator nested within each host grid column of a climate simulator [25–29]. The smaller-scale simulator explicitly resolves the detailed behavior of clouds and their turbulent motions at both a higher spatial and temporal resolution (but with a smaller domain) than the host simulator. This improves the accuracy of the host simulations, but comes at a high computational cost [30, 31]. The time-integrated and horizontally averaged influence of the resolved convection is fed upscale to the host climate simulator, and is the target of hybrid ML-climate simulation approaches.

**Significance of Precipitation Processes for Climate Impacts:** In climate simulations, changes in precipitation with warming is a particularly important issue. The frequency of extreme precipitation events increases with warming [32–34], with corresponding societal impacts [35]. Current climate simulators agree on the direction of this change, but exhibit large spread in the quantitative rate of increase with warming [36, 37].

## 2   Related Work

There have been several recent efforts to produce hybrid-ML emulators using multi-scale climate simulations, analogous to ClimSim [4, 10–16, 38]. Most of these focused on simple aquaplanets [4, 10–13, 16, 38] and those that included real geography [14, 15] did not include enough variables for complete land-surface coupling, to our knowledge. Most examine simple multi-layer perceptrons except for [12, 15], who used a ResNet architecture, and [39] who used a variational encoder-decoder that accounts for stochasticity. Although downstream hybrid testing in real-geography settings is error-prone, [15] demonstrates some hybrid stability. Compressing input data to avoid causal confounders may improve downstream accuracy [16], and methods have been proven to enforce physical constraints [40, 41].

Compared to the training data used above, ClimSim's comprehensive variable coverage is unprecedented, including all variables needed to couple to and from a land system simulator and enforce physical constraints. Its availability across coarse-resolution, high-resolution, aquaplanet and real-geography use cases is also new to the community. Successful ML innovations with ClimSim can have a downstream impact since it is based on a state-of-the-art multi-scale climate simulator that is actively supported by a mission agency (U.S. Department of Energy).

In non-multi-scale settings, an important body of related work [6–9] has made exciting progress on using analogous hybrid ML approaches to reduce biases in uniform resolution climate simulations, including in an operational climate code with land coupling and downstream hybrid stability [17, 18] (see Supplementary Information; SI). Other related work includes full model emulation (FME) for short-term weather prediction [42–44]. Whether this approach is possible for climate simulation using the high-frequency output of its state variables remains an open question. For instance, it has recently been shown that incorporating spherical geometry and resolution invariance through spherical Fourier neural operators leads to stability of long rollouts [43]. While ClimSim is focused on hybrid-ML climate simulation and we do not demonstrate FME baselines, ClimSim contains full atmospheric state variable sampling well suited for the task.

## 3   ClimSim Dataset Construction

**Experiment Outline:** ClimSim presents a regression problem with mapping from a multivariate input vector, with inputs $x \in \mathbb{R}^{d_i}$ of size $d_i = 124$ and targets $y \in \mathbb{R}^{d_o}$ of size $d_o = 128$ (Figure 1). The input represents the local vertical structure (in horizontal location and time) of macro-scale state variables in a multi-scale physical climate simulator before any adjustments from sub-grid scale convection and radiation are made. The input also includes concatenated scalars containing boundary conditions of incoming radiation at the top of the atmospheric column, and land surface model constraints at its base. The target vector contains the tendencies of the same state variables representing the redistribution of mass and water, microphysical water species conversions, and radiative heating feedbacks associated with explicitly resolved convection. This brackets the change

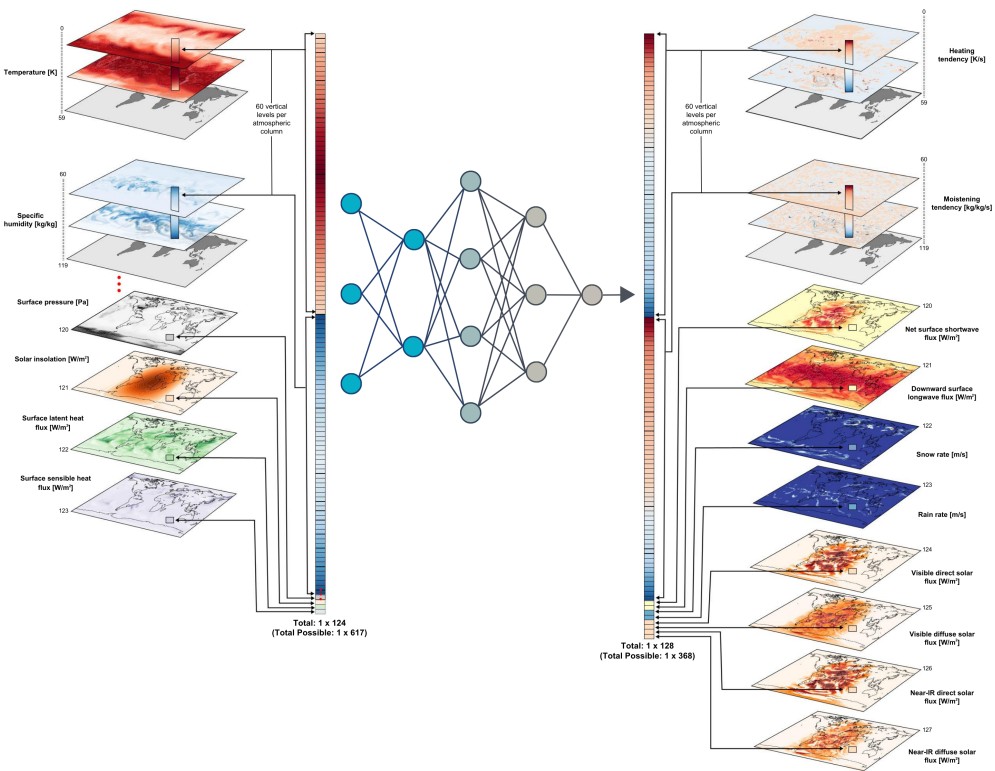

Figure 1: The spatially-local version of ClimSim that our baselines are scored on. A spatially-global version of the problem that expands to the full list of variables would be useful to try.

in atmospheric state after tens of thousands of computationally intensive, spatially nested simulators of explicit cloud physics have completed a temporally-nested integration. The ultimate goal is to outsource these physics to ML by mapping inputs to targets at comparable fidelity. The target vector includes scalar fields and fluxes from the bottom of the atmospheric column expected by the land surface model component that it must couple to; land-atmosphere coupling is important to predicting regional water cycle dynamics [45, 46]. Importantly, ClimSim also includes the option for *expanded inputs* $x \in \mathbb{R}^{d_i}$ of size $d_i = 617$ and targets $y \in \mathbb{R}^{d_o}$ of size $d_o = 368$, which we demonstrate in one of our experiments.

**Locality vs. Nonlocality:** A spatially-global version of the problem could be of practical use for improving ML via helpful spatial context [47, 48]. In such a case, the problem becomes 2D → 2D regression, and would encompass inputs $x \in \mathbb{R}^{d_i}$ of maximum size $d_i = 617 \times 21,600$ (grid columns) and targets, $y \in \mathbb{R}^{d_o}$, of maximum size $d_o = 368 \times 21,600$. Here the second dimension represents the unstructured "cube-sphere" computational mesh used by the climate model, which is a list of grid cell locations that span the surface of the sphere [49]. In contrast to typical image-to-image translation or spatio-temporal prediction problems in ML that involve data on a structured grid (i.e. rectilinear), the task at hand is of lower dimensionality. Further details about the climate simulator configuration, simulations, and data, including complete variable lists, can be found in SI.

**Dataset Collection:** We ran the E3SM-MMF multi-scale climate simulator [28, 29, 49, 50], using multiple NVIDIA A100 GPUs for a total of ∼ 9,800 GPU-hours. We saved global instantaneous values of the atmospheric state before and after high-resolution calculations occurred, isolating state updates due to explicitly-resolved moist convection, boundary layer turbulence, and radiation; details of the climate simulator configuration can be found in SI. These data were saved at 20-minute intervals (i.e. the time step of the climate model) for 10 simulated years, resulting in 5.7 billion samples for the high-resolution simulation that uses an unstructured "cube-sphere" horizontal grid with 21,600 grid columns spanning the globe. This grid yields an *approximate* horizontal grid spacing of 1.5°, but unlike a traditional climate model that maps points across the sphere using two dimensions aligned with cardinal north/south and east/west directions, unstructured grids use a single dimension to organize the horizontal location of points. The atmospheric columns at each location and time are

treated as independent samples. Thus, the total number of samples can be understood by considering that atmospheric columns at each location and time are treated as independent samples, such that 5.7 billion ≈ 21,600 horizontal locations per time step × 72-time steps per simulated day × 3,650 simulated days). It is important to note that each sample retains a 1D structure corresponding to the vertical variation across 60 levels. We also ran two additional simulations with approximately ten times less horizontal resolution, with only 384 grid columns spanning the globe, resulting in 100 million samples for each simulation. These low-resolution options allow for fast prototyping of ML models, due to smaller training data volumes and less geographic complexity. One low-resolution simulation uses an "aquaplanet" configuration, i.e., a lower boundary condition of specified sea surface temperature, invariant in the longitudinal dimension with no seasonal cycle. This is the simplest prototyping dataset, removing variance associated with continents and time-varying boundary conditions. The total data volume is 41.2TB for the high-resolution dataset and 744GB for each of the low-resolution datasets.

**Dataset Interface:** Raw model outputs emerge from the climate simulator as standard NetCDF files which can be easily parsed in any language. Each timestep yields files containing input and target vectors separately, resulting in a total of 525,600 files for each of the three datasets. To prevent redundancy, variable metadata and grid information was saved separately.

The raw tensors from the climate simulations are initially either 2D or 3D, depending on the variable. For 2D tensors, the dimensions represent time and horizontal location. While these variables actually depend on three physical dimensions (time and 2D space), since each location on the sphere is indexed along a single axis due to the climate model's unstructured horizontal grid, the apparent dimensionality is lower. Such variables include solar insolation, snow depth over land, surface energy fluxes, and surface precipitation rate. 3D tensors include the additional dimension representing altitude relative to the Earth's surface, for height-varying state variables like temperature, humidity, and wind vector components. Separate files are used to store each timestep and variable. ClimSim includes a total of 24 2D variables and 10 3D variables (see Table 1 in SI).

**Dataset Split:** The 10-year datasets are divided into: (a) a training and validation spanning the first 8 years (0001-02 to 0009-01; YYYY-MM), excluding the first simulated month for numerical spin-up, and (b) a test set spanning the remaining two years (i.e., 0009-03 to 0011-02). A one-month gap is intentionally introduced between the two sets to prevent test set contamination via temporal correlation. Both sets are stored separately in our data repositories.

**Energy use:** The computing and energy costs of generating ClimSim could be viewed as wasteful and having a negative consequence for society through associated emissions. We emphasize that while it can appear large, the compute used is actually orders of magnitude less than what is consumed by operational climate prediction. Associated emissions are minimized given that our integrations were performed on energy-efficient GPU hardware. The cost must also be weighed against the potential social benefit of mitigating future energy consumption by eliminating end users' need for costly physics-based MMF simulations. Meanwhile, a large consortium of interested parties have helped agree on this dataset, to help ensure it is not wasted.

## 4 Experiments

To guide ML practitioners using ClimSim, we provide an example ML workflow using the low-resolution, real-geography dataset for the task described in Section 1. All but one of our baselines focuses on emulating the subset of total available input and target variables illustrated in Figure 1, with the following inputs $x \in \mathbb{R}^{d_i}$ of size $d_i = 124$, and targets $y \in \mathbb{R}^{d_o}$ of size $d_o = 128$ (Figure 1, Table 1), chosen for its similarity to recent attempts in the literature.

**Training/Validation Split:** We divide the 8-year training/validation set into the first 7 years (i.e., 0001-02 to 0008-01 in the raw filenames' "year-month" notation) for training and the subsequent 1 year (0008-02 to 0009-01) for validation.

**Preprocessing Workflow:** Our preprocessing steps were (1) downsample in time by using every 7th sample, (2) collapse horizontal location and time into a single sample dimension, (3) normalize variables by subtracting the mean and dividing by the range, with these statistics calculated separately at each of the 60 vertical levels for the four variables with vertical dependence, and (4) concatenate variables into multi-variate input and output vectors for each sample (Figure 1). The heating tendency

| Input | Size | Target | Size |
|---|---|---|---|
| Temperature [K] | 60 | Heating tendency, $dT/dt$ [K/s] | 60 |
| Specific humidity [kg/kg] | 60 | Moistening tendency, $dq/dt$ [kg/kg/s] | 60 |
| Surface pressure [Pa] | 1 | Net surface shortwave flux, NETSW [W/m$^2$] | 1 |
| Insolation [W/m$^2$] | 1 | Downward surface longwave flux, FLWDS [W/m$^2$] | 1 |
| Surface latent heat flux [W/m$^2$] | 1 | Snow rate, PRECSC [m/s] | 1 |
| Surface sensible heat flux [W/m$^2$] | 1 | Rain rate, PRECC [m/s] | 1 |
| | | Visible direct solar flux, SOLS [W/m$^2$] | 1 |
| | | Near-IR direct solar flux, SOLL [W/m$^2$] | 1 |
| | | Visible diffused solar flux, SOLSD [W/m$^2$] | 1 |
| | | Near-IR diffused solar flux, SOLLD [W/m$^2$] | 1 |

Table 1: The subset of input and target variables used in most of our experiments (Figure 1). Dimension length 60 corresponds to the total number of vertical levels (discretized altitudes) of the climate simulator.

target $dT/dt$ (i.e., time rate of temperature $T$) was calculated from the raw climate simulator output as $(T_{after} - T_{before})/\Delta t$, where $\Delta t = 1200$ s) is the climate simulator's known macro-scale timestep. Likewise, the moisture tendency was calculated via taking the difference of humidity state variables recorded before versus after the convection and radiation calculations. This target variable transformation is done so that we can compare the performance of our baseline models to that of previously published models that reported errors of emulated tendencies [14, 39]. Additionally, this transformation implicitly normalizes the target variables leading to better convergence properties for ML algorithms. Given the domain-specific nature of the preprocessing workflow, we provide scripts in the GitHub repository for workflow reproduction.

## 4.1 Baseline Architectures

Six baseline models used in our experiment are briefly described here. Refer to SI for further details.

**Convolutional Neural Network (CNN)** uses a 1D ResNet-style network. Each ResNet block contains two 1D convolutional layers and a skip connection. CNNs can learn spatial structure and have outperformed MLP and graph-based networks in [51]. The inputs and outputs for the CNN are stacked in the channel dimensions, such that the mapping is $60 \times 6 \rightarrow 60 \times 10$. Accordingly, global variables have been repeated along the vertical dimension.

**Encoder-Decoder (ED)** consists of an Encoder and a Decoder with 6 fully-connected hidden layers each [39]. The Encoder of ED condenses the original dimensionality of the input variables down to only 5 nodes inside the latent space. This enhances the interpretability of ED and makes the model beneficial for advanced postprocessing of multivariate climate data [39].

**Heteroskedastic Regression (HSR)** [52] predicts a separate mean and standard deviation for each output variable, using a regularized MLP.

**Multi-layer Perceptron (MLP)** is a fully connected, feed-forward neural network. The MLP architecture used for our experiments is optimized via an extensive hyperparameter search with 8,257 trials.

**Randomized Prior Network (RPN)** is an ensemble model [53]. Each member of the RPN is built as the sum of a trainable and a non-trainable (so-called "prior") surrogate model; we used MLP for simplicity. Multiple replicas of the networks are constructed by independent and random sampling of both trainable and non-trainable parameters [54, 55]. RPNs also resort to data bootstrapping (e.g., subsampling and randomization) in order to mitigate the uncertainty collapse of the ensemble method when tested beyond the training data points [55].

**Conditional Variational Autoencoder (cVAE)** uses amortized variational inference to fit a deep generative model that is conditioned on the input and can produce samples from a complex predictive distribution.

| Variable | MAE [W/m$^2$] | | | | | | R$^2$ | | | | | |
|---|---|---|---|---|---|---|---|---|---|---|---|---|
| | CNN | ED | HSR | MLP | RPN | cVAE | CNN | ED | HSR | MLP | RPN | cVAE |
| $dT/dt$ | **2.585** | 2.864 | 2.845 | 2.683 | 2.685 | 2.732 | **0.627** | 0.542 | 0.568 | 0.589 | 0.617 | 0.590 |
| $dq/dt$ | **4.401** | 4.673 | 4.784 | 4.495 | 4.592 | 4.680 | – | – | – | – | – | – |
| NETSW | 18.85 | 14.968 | 19.82 | **13.36** | 18.88 | 19.73 | 0.944 | 0.980 | 0.959 | **0.983** | 0.968 | 0.957 |
| FLWDS | 8.598 | 6.894 | 6.267 | **5.224** | 6.018 | 6.588 | 0.828 | 0.802 | 0.904 | **0.924** | 0.912 | 0.883 |
| PRECSC | 3.364 | 3.046 | 3.511 | **2.684** | 3.328 | 3.322 | – | – | – | – | – | – |
| PRECC | 37.83 | 37.250 | 42.38 | **34.33** | 37.46 | 38.81 | **0.077** | -17.909 | -68.35 | -38.69 | -67.94 | -0.926 |
| SOLS | 10.83 | 8.554 | 11.31 | **7.971** | 10.36 | 10.94 | 0.927 | 0.960 | 0.929 | **0.961** | 0.943 | 0.929 |
| SOLL | 13.15 | 10.924 | 13.60 | **10.30** | 12.96 | 13.46 | 0.916 | 0.945 | 0.916 | **0.948** | 0.928 | 0.915 |
| SOLSD | 5.817 | 5.075 | 6.331 | **4.533** | 5.846 | 6.159 | 0.927 | 0.951 | 0.923 | **0.956** | 0.940 | 0.921 |
| SOLLD | 5.679 | 5.136 | 6.215 | **4.806** | 5.702 | 6.066 | 0.813 | 0.857 | 0.797 | **0.866** | 0.837 | 0.796 |

Table 2: MAE and R$^2$ for target variables averaged globally and temporally (from 0009-03 to 0011-02). Variables include heating tendency ($dT/dt$), moistening tendency ($dq/dt$), net surface shortwave flux (NETSW), downward surface longwave flux (FLWDS), snow rate (PRECSC), rain rate (PRECC), visible direct solar flux (SOLS), near-IR direct solar flux (SOLL), visible diffused solar flux (SOLSD), and near-IR diffused solar flux (SOLLD). Units of non-energy flux variables are converted to a common energy unit, W/m$^2$. Best model performance for each variable is bolded.

## 4.2 Skill Boost from Expanding Features and Targets

We performed an ablation of our best performing MLP baseline to demonstrate the added value of the expanded inputs and targets available in ClimSim, i.e. using inputs $x$ of size $d_i = 617$ and targets $y \in \mathbb{R}^{d_o}$ of size $d_o = 368$; see Table 1 in SI for the full list of variables. We use the same transformation described in our preprocessing workflow to compute and add condensate (cloud liquid and cloud ice) and momentum (zonal and meridional winds) tendencies to the target vector. We conducted this ablation study with both the low-resolution and the high-resolution datasets (see Section 3.1 in SI for further details regarding these MLP variants). For common elements of the target vector, using all available variables leads to a uniform improvement in prediction accuracy, especially for precipitation, in both resolutions (Figures SI7, SI8 and Table SI4). The larger errors (e.g., MAE and RMSE) observed in the high-resolution emulators are anticipated due to the increased variance of higher-resolution data. Nevertheless, the similarity of their R$^2$ values to those of the corresponding low-resolution emulators confirms their adequate performance.

## 4.3 Evaluation Metrics

Our evaluation metrics are computed separately for each variable in the output vector. Mean absolute error (MAE) and the coefficient of determination (R$^2$) are calculated independently at each horizontal and vertical location, and then averaged horizontally and vertically to produce the summary statistics in Figure 2. For the vertically-varying fields, we first form a mass-weighting and then convert moistening and heating tendencies into common energy units in Watts per square meter as in [56]. We also report continuous ranked probability scores (CRPS) for all considered models in SI.

## 4.4 Baseline Model Results

Figure 2 summarizes the error characteristics. Whereas heating and moistening rates have comparable global mean MAE, behind a common background vertical structure (Figure 2 b,c) the coefficient of determination R$^2$ (d,e) reveals that certain architectures (RPN, HSR, cVAE, CNN) consistently perform better in the upper atmosphere (model level < 30) whereas the highly optimized MLP model outperforms in the lower atmosphere (model level > 30) and therefore the global mean (Table 2). For the global mean MAE we see the largest averaged errors for PRECC and NETSW (mean MAE > 15 W/m², Figure 2 and Table 2), where MLP clearly has the best the best skill compared to all other benchmark models. For the other variables, the global mean MAE is considerably smaller and the skill of the benchmarks model appears to be more similar in absolute numbers. While for the global mean R$^2$ we find the lowest measurable performance for dT/dt and PRECC (mean R$^2$ < 0.7) and in these cases, CNN gives the most skillful predictions. The other variables have larger R$^2$ of order 0.8 or higher, which suggests that these quantities are easier to deep-learn (Table 2). For dq/dt and PRECSC global mean R$^2$ is not an ideal evaluation metric due to negligible variability in dq/dt in the upper atmosphere and for PRECSC in the tropics in the dataset (Table 2).

Additional tables and figures that reveal the geographic and vertical structure of these errors, fit quality, and analysis of stochastic metrics, are included in SI (Sections 4.3, 8.1, and 8.2 in SI).

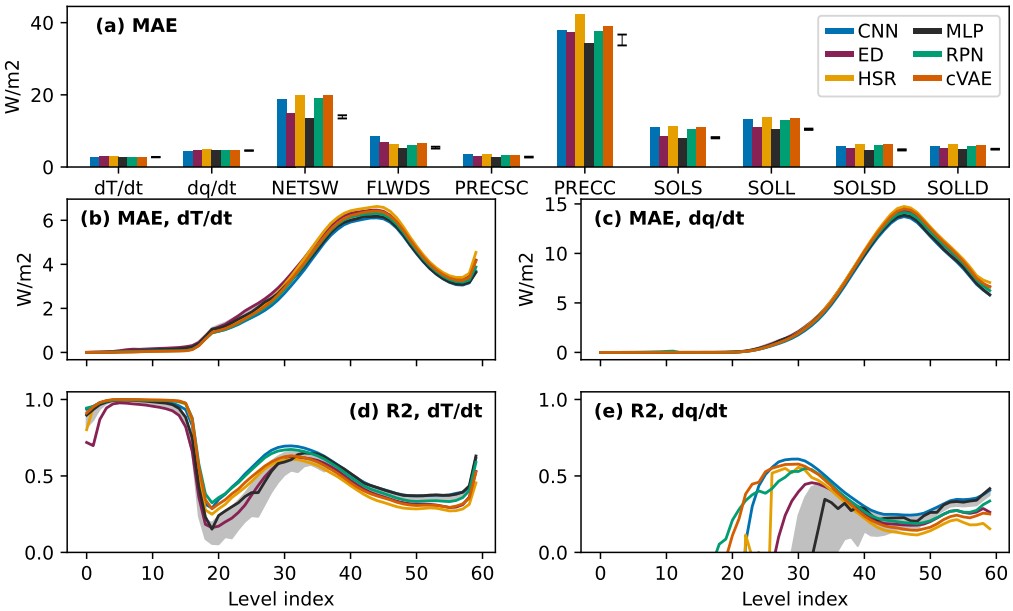

Figure 2: (a) Summary, where $dT/dt$ and $dq/dt$ are the tendencies of temperature and specific humidity, respectively, and were vertically integrated with mass weighting. (b,c) retain the vertical structure of MAE and (d,e) $R^2$. Error bars and grey shadings show the the 5- to 95-percentile range of MLP. Refer to Table 1 for variable definitions.

## 4.5 Physics-Informed Guidance to Improve Generalizability and Coupled Performance

**Physical Constraints:** Mass and energy conservation are important criteria for Earth system simulation. If these terms are not conserved, errors in estimating sea level rise or temperature change over time may become as large as the signals we hope to measure. Enforcing conservation on emulated results helps constrain results to be physically plausible and reduce the potential for errors accumulating over long time scales. We discuss how to do this and enforce additional constraints, such as non-negativity for precipitation, condensate, and moisture variables in the Supporting Information.

**Stochasticity and Memory:** The results of the embedded convection calculations regulating $d_o$ are chaotic, and thus worthy of stochastic architectures, as in our RPN, HSR, and cVAE baselines. These solutions are likewise sensitive to sub-grid initial state variables from an interior nested spatial dimension that has not been included in our data.

**Temporal Locality:** Incorporating the previous timesteps' target or feature in the input vector inflation could be beneficial as it captures some information about this convective memory and utilizes temporal autocorrelations present in atmospheric data.

**Causal Pruning:** A systematic and quantitative pruning of the input vector based on objectively assessed causal relationships to subsets of the target vector has been proposed as an attractive preprocessing strategy, as it helps remove spurious correlations due to confounding variables and optimize the ML algorithm [16].

**Normalization:** Normalization that goes beyond removing vertical structure could be strategic, such as removing the geographic mean (e.g., latitudinal, land/sea structure) or composite seasonal variances (e.g., local smoothed annual cycle) present in the data. For variables exhibiting exponential variation and approaching zero at the highest level (e.g., metrics of moisture), log-normalization might be beneficial.

**Expanded Resolution and Complete Inputs and Outputs:** Our baseline models have focused on the low-resolution dataset, for ease of data volume, and using only a subset of the available inputs and outputs. This illustrates the essence of the ML challenge. However, we show in our ablation study, using MLPs, that including all input variables yields generally an improved reproduction of the target variables in both the low-resolution and the high-resolution dataset (Figures SI7 and SI8 and Table SI4). Accordingly, we encourage users who discover competitive fits in this approachable limit to expand to all inputs/outputs in the high-resolution, real-geography dataset, for which successful fits become operationally relevant.

**Further ML Approaches:** Recent methods to capture multi-scale processes using neural operators that learn in a discretization-invariant manner and can predict at higher resolutions than available during training time [57] may be attractive. Their performance can be further enhanced by incorporating physics-informed losses at a higher resolution than available training data [58]. Ideas on ML modeling for sub-grid closures from adjacent fields like turbulent flow physics and reactive flows can also be leveraged for developing architectures with an inductive bias for known priors [59], easing prediction of stiff non-linear behavior [60–62], generative modeling with physical constraints [63, 64] and for interpretability of the final trained models [60].

## 5    Limitations and Other Applications

**Idealizations:** A limitation of the multi-scale climate simulator used to produce ClimSim (E3SM-MMF) is that it assumes scale separation, i.e., that convection can be represented as laterally periodic within the grid size of the host simulator, and neglects sub-grid scale representations of topographic and land-surface variability. Despite these simplifications, the data adequately captures many essential aspects of the ML problem, such as stochasticity, and interactions across radiation, microphysics, and turbulence.

**Hybrid testing:** Inclusion of a natural path for downstream testing of learned physics emulators as fully coupled components of a hybrid-ML climate simulator is vital. However, such a workflow is not yet included in ClimSim, since there is no easy way for the ML community to run many hybridized variants of the E3SM-MMF in a distributed high-performance GPU computing infrastructure via a lightweight API. It is our eventual goal to tackle the software engineering needed to enable such a protocol, since, in the long term, it is in this downstream environment where ML researchers should expect to have their maximum impact on the field of hybrid-ML climate simulation. Meanwhile, ClimSim provides the first step.

**Stochasticity:** One open problem that the dataset may allow assessing is understanding the role of stochasticity in hybrid-ML simulation. While primarily used as a dataset for regression, it would be also interesting to assess and understand the degree to which different variables are better modeled as stochastic or deterministic, or if the dataset gives rise to heavy-tailed or even multi-modal conditional distributions that are important to capture. To date, these questions have been raised based on physical conjectures [e.g., 65] but remain to be addressed in the ML-based parameterization literature. For instance, precipitation distributions have long tails that are projected to lengthen under global warming [34, 66]—and will thus tend to generate out-of-sample extremes. ClimSim could help construct optimal architectures to capture precipitation tails and other impactful climate variables such as surface temperature.

**Interpretability:** This dataset could also be utilized to discover physically interpretable models for atmospheric convection, radiation, and microphysics. A possible workflow would apply dimensionality reduction techniques to identify dominant vertical variations, followed by symbolic regression to recover analytic expressions [67, 68].

**Generalizability:** Although the impacts of global warming and inter-annual variability are absent in this initial version of ClimSim, important questions surrounding climate-convection interactions can begin to be addressed. One strategy would involve partitioning the data such that the emulator is trained on cold columns, but validated on warm columns, where warmth could be measured by surface temperatures, as in [56]. However, the results from this approach may also reflect the dependence of convection on the geographical distribution of surface temperatures in the current climate and should be interpreted with caution. To optimally engage ML researchers in solving the climate generalization problem, a multi-climate extension of ClimSim should be developed that includes physical simulations that samples future climate states and more internal variability.

**Relevance determination and active learning:** While the climate simulator code offers data generation flexibility, guidance on ideal regimes to target for improved learning would benefit the domain scientists able to run it. This question can be addressed with the current data and metrics of interest provided.

# 6    Conclusion and Future Work

We introduce ClimSim, the most physically comprehensive dataset yet published for training ML emulators of atmospheric storms, clouds, turbulence, rainfall, and radiation for use in hybrid-ML climate simulation. It contains all inputs and outputs necessary for downstream coupling in a full-complexity multi-scale climate simulator. We conduct a series of experiments on a subset of these variables that demonstrate the degree to which climate data scientists have been able to fit their deterministic and stochastic components.

We hope ML community engagement in ClimSim will advance fundamental ML methodology and clarify the path to producing increasingly skillful sub-grid physics emulators that can be reliably used for operational climate simulation. To facilitate two-way commications between ML practitioners and climate scientists, we incorporate many desired characteristics for an ideal benchmark dataset suggeted in [69]. Such interdisciplinary collaboration will open up an exciting future in which the computational limits that currently constrain climate simulation can be reconsidered.

We plan to soon extend ClimSim to include, first, a sampling of multiple future climate states. Second, we aim to provide a protocol for downstream hybrid simulation testing. We hope lessons learned in our chosen limit of multi-scale atmospheric simulation will have applicability in other sub-fields of Earth System Science where computational constraints are currently a barrier to including explicit representations of more systems of nested complexity.

## Acknowledgements

This work is broadly supported across countries and agencies. Primary support is by the National Science Foundation (NSF) Science and Technology Center (STC) Learning the Earth with Artificial Intelligence and Physics (LEAP), Award # 2019625-STC and the Exascale Computing Project (17-SC-20-SC), a collaborative effort of the U.S. Department of Energy (DOE) Office of Science (SC), the National Nuclear Security Administration, and the Energy Exascale Earth System Model project, funded by DOE grant DE-SC0022331. M.S.P, S.Y., L.P, A.M.J., J.L., N.L., and G.M. further acknowledge support from the DOE (DE-SC0023368) and NSF (AGS-1912134). R.Y, S.M, P.G, M.P. acknowledge funding from the DOE Advanced Scientific Computing Research (ASCR) program (DE-SC0022255). V.E., P.G., G.B., and F.I.-S. acknowledge funding from the European Research Council Synergy Grant (Agreement No. 855187) under the Horizon 2020 Research and Innovation Programme. E.A.B. was supported, in part, by NSF grant AGS-2210068. S.J. acknowledges funding from DOE ASRC under an Amalie Emmy Noether Fellowship Award in Applied Mathematics (B&R #KJ0401010). M.A.B acknowledges NSF funding from an AGS-PRF Fellowship Award (AGS-2218197). R.G. acknowledges funding from the NSF (DGE-2125913) and the U.S. Department of Defense (DOD). S.M. acknowledges support from an NSF CAREER Award and NSF grant IIS-2007719. L.Z. and N.L. received $M^2$LInES research funding by the generosity of Eric and Wendy Schmidt by recommendation of the Schmidt Futures program. This research used resources of the National Energy Research Scientific Computing Center (NERSC), a DOE SC User Facility operated under Contract No. DE-AC02-05CH11231. The Pacific Northwest National Laboratory is operated by Battelle for the DOE under Contract DE-AC05-76RL01830. This work was performed under the auspices of the DOE by Lawrence Livermore National Laboratory under Contract DE-AC52-07NA27344. This work used Bridges2 at the Pittsburgh Supercomputing Center through allocation ATM190002 from the Advanced Cyberinfrastructure Coordination Ecosystem: Services & Support (ACCESS) program, which is supported by NSF grants #2138259, #2138286, #2138307, #2137603, and #2138296. This work also utilized the DOD High Performance Computing Modernization Program (HPCMP).

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
