# ClimSim: Supplementary Information

**Sungduk Yu**[1][*]**, Walter M. Hannah**[2]**, Liran Peng**[1]**, Jerry Lin**[1]**, Mohamed Aziz Bhouri**[3]**,
Ritwik Gupta**[4]**, Björn Lütjens**[5]**, Justus C. Will**[1]**, Gunnar Behrens**[6]**, Julius J. M. Busecke**[3]**,
Nora Loose**[7]**, Charles Stern**[3]**, Tom Beucler**[8]**, Bryce E. Harrop**[9]**, Benjamin R. Hillman**[10]**,
Andrea M. Jenney**[1,11]**, Savannah L. Ferretti**[1]**, Nana Liu**[1]**, Anima Anandkumar**[12]**,
Noah D. Brenowitz**[12]**, Veronika Eyring**[6]**, Nicholas Geneva**[12]**, Pierre Gentine**[3]**,
Stephan Mandt**[1]**, Jaideep Pathak**[12]**, Akshay Subramaniam**[12]**, Carl Vondrick**[3]**, Rose Yu**[13]**,
Laure Zanna**[14]**, Tian Zheng**[3]**, Ryan P. Abernathey**[3]**, Fiaz Ahmed**[15]**, David C. Bader**[2]**,
Pierre Baldi**[1]**, Elizabeth A. Barnes**[16]**, Christopher S. Bretherton**[17]**, Peter M. Caldwell**[2]**,
Wayne Chuang**[3]**, Yilun Han**[18]**, Yu Huang**[3]**, Fernando Iglesias-Suarez**[6]**, Sanket Jantre**[19]**,
Karthik Kashinath**[12]**, Marat Khairoutdinov**[20]**,Thorsten Kurth**[12]**, Nicholas J. Lutsko**[13]**,
Po-Lun Ma**[9]**, Griffin Mooers**[1]**, J. David Neelin**[15]**, David A. Randall**[16]**, Sara Shamekh**[3]**,
Mark A. Taylor**[10]**, Nathan M. Urban**[19]**, Janni Yuval**[5]**, Guang J. Zhang**[13]**,
Michael S. Pritchard**[1,12]

[1]UCI, [2]LLNL, [3]Columbia, [4]UCB, [5]MIT, [6]DLR, [7]Princeton, [8]UNIL, [9]PNNL, [10]SNL, [11]OSU,
[12]NVIDIA, [13]UCSD, [14]NYU, [15]UCLA, [16]CSU, [17]Allen AI, [18]Tsinghua, [19]BNL, [20]SUNY

## Contents

---

[*]Corresponding author: sungduk@uci.edu

37th Conference on Neural Information Processing Systems (NeurIPS 2023) Track on Datasets and Benchmarks.

# 1   Climate Simulations

Climate models divide the Earth's atmosphere, land surface, and ocean into a 3D grid, creating a discretized representation of the planet. Somewhat like a virtual Lego construction of Earth, with each brick representing a small region (grid cell). Earth system models are made up of independent component models for the atmosphere, land surface, rivers, ocean, sea ice, and glaciers. Each of

these component models is developed independently and can run by itself when provided with the appropriate input data. When running as a fully coupled system the "component coupler" handles the flow of data between the components.

Within each grid cell of the component models, a series of complex calculations are performed to account for various physical processes, such as phase changes of water, radiative heat transfer, and dynamic transport (referred to as "advection"). Each component model uses the discretized values of many quantities (such as temperature, humidity, and wind speed) as inputs to parameterizations and fluid solvers to output those same values for a future point in time.

The atmosphere and ocean components are the most expensive pieces of an Earth system model, which is largely due to the computation and inter-process communication associated with their fluid dynamics solvers. Furthermore, a significant portion of the overall cost is attributed to the atmospheric physics calculations that are performed locally within each grid column. It is important to note that atmospheric physics serves as a major source of uncertainty in climate projections, primarily stemming from the challenges associated with accurately representing cloud and aerosol processes.

## 1.1   Model Description

The data that comprise ClimSim are from simulations with the Energy Exascale Earth System Model-Multiscale Modeling Framework version 2.1.0 (E3SM-MMF v2) [1]. Traditionally, global atmospheric models parameterize clouds and turbulence using crude, low-order models that attempt to represent the aggregate effects of these processes on larger scales. However, the complexity and nonlinearity of cloud and rainfall processes make them particularly challenging to represent accurately with parameterizations. The MMF approach replaces these conventional parameterizations with a cloud resolving model (CRM) in each cell of the global grid, so that cloud and turbulence can be explicitly represented. Each of these independent CRMs is spatially fixed and exchange coupling tendencies with a parent global grid column. This novel approach to representing clouds and turbulence can improve various aspects of the simulated climate, such as rainfall patterns [2].

The configuration of E3SM-MMF used here shares some details with E3SMv2. The dynamical code of E3SM uses a spectral element approach on a cubed-sphere geometry. Physics calculations are performed on an unstructured, finite-volume grid that is slightly coarser than the dynamics grid, following Hannah et al. (2021) [3], which is better aligned with the effective resolution of the dynamics grid. Cases with realistic topography include an active land model component that responds to atmospheric conditions with the appropriate fluxes of heat and momentum.

The embedded CRM in E3SM-MMF is adapted from the System for Atmospheric Modeling (SAM) described by Khairoutdinov and Randall (2003) [4]. While the CRM does explicitly represent clouds and turbulence, it still cannot represent the smallest scales of turbulence and microphysics, and, therefore, these processes still need to be parameterized within each CRM grid cell. Microphysical processes are parameterized with a single-moment scheme, and sub-grid scale turbulent fluxes are parameterized using a diagnostic Smagorinsky-type closure. Convective momentum transport in the nested CRM is handled using the scalar momentum tracer approach of Tulich (2015) [5]. The CRM uses an internal timestep of 10 seconds, while the global calculations use a timestep of 20 minutes.

Despite recent efforts to accelerate E3SM-MMF with GPUs and algorithmic techniques [6], the CRM domain size strongly affects the computational throughput and limits the type of experiments that can be conducted. However, the MMF approach is quite flexible in how the CRM size is specified. E3SM-MMF is typically run with a 2D CRM that neglects one of the horizontal dimensions, and employs relatively coarse grid spacing that cannot represent small clouds. Increasing the size of this

2D domain by adding further columns (more CRM cells) generally improves the realism of the model solution. Reducing the model grid spacing can also improve the model to a certain degree, although the number of columns often needs to be increased to avoid the degradation associated with a small CRM. Ideally, the CRM would always be used in a 3D configuration to fully capture the complex, chaotic turbulence that dictates the life cycle of each individual cloud, but this approach is generally limited to special experiments that can justify the extra computational cost. The simulations for ClimSim utilize a 2D CRM with 64 columns and 2 km horizontal grid spacing within each grid cell.

The atmospheric component of E3SM uses a hybrid vertical grid that is "terrain-following" near the surface, and transitions to be equivalent to pressure levels near the top (e.g., https://www2.cesm.ucar.edu/models/atm-cam/docs/usersguide/node25.html). The vertical levels are specified to be thin near the surface to help capture turbulent boundary layer processes, and are gradually stretched to be very coarse in the stratosphere. E3SM-MMF uses 60 levels for the global dynamics with a top level around 65 km. The CRM used for atmospheric physics uses 50 levels, ignoring the upper 10 levels, to avoid problems that arise from using the anelastic approximation with very low densities. This does not create any issues, because cloud processes are generally confined to the troposphere where the anelastic approach is valid. The hybrid grid can be converted to pressure levels using Equation 1, where $P_0 = 100,000$ Pa is a reference pressure, and $P_s(\mathbf{x}, t)$ is the surface pressure which varies in location $\mathbf{x}$ and time $t$:

$$P_k = A_k P_0 + B_k P_s \tag{1}$$

$A_k$ and $B_k$—where the subscript $k$ denotes the index of vertical coordinate—are the fixed, prescribed coefficients that define how the "terrain-following" and "pure pressure" coordinates are blended to define the hybrid coordinate at each vertical level. $A_k$ and $B_k$ are provided as a part of the dataset with variable names of `hyam` and `hyai` or `hybm` and `hybi`, depending on whether mid-level or interface values are needed. The third character of the variable names ("a" and "b") in Equation 1 denotes $A_k$ and $B_k$ coefficients, respectively. Note that the indexing of the vertical coordinate starts from the top of the atmosphere due to the construct of $A_k$ and $B_k$ coefficients, e.g., $k = 0$ for the top and $k = 59$ for the surface in E3SM-MMF.

In the E3SM-MMF framework, the sequencing of atmospheric processes can be conceptualized as follows. It starts with a surface coupling step that receives fluxes from the surface component models (i.e., land, ocean, and sea ice). This is followed by a set of relatively inexpensive physics parameterizations that handle processes such as airplane emissions, boundary layer mixing, and unresolved gravity waves. The global dynamics then takes over to evolve the winds and advect tracers on the global grid. Finally, there is another set of physics calculations to handle clouds, chemistry, and radiation, which are relatively expensive. This final physics section is where the embedded CRM of E3SM-MMF is used, and is the ideal target for surrogate model emulation due to its outsized computational expense. Accordingly, this step represents the target of ClimSim.

One area where E3SM-MMF significantly differs from E3SMv2 is in the treatment of aerosols and chemistry. The embedded CRM in E3SM-MMF predicts the mass of water species (i.e., cloud and rain droplet mass mixing ratios) but does not predict the number concentration (i.e., number of drops per mass of air). One consequence of this limitation is that E3SM-MMF cannot represent complex cloud aerosol interactions that can impact droplet number concentrations and cloud radiative feedbacks. Therefore, E3SM-MMF cannot use the more sophisticated aerosol and chemistry package used by E3SMv2, and instead uses prescribed aerosol and ozone amounts to account for the direct radiative impact of these tracers. Current efforts are addressing this limitation for future versions of E3SM-MMF.

## 1.2 Model Configurations

The simulations used for ClimSim were performed on the NERSC Perlmutter machine. E3SM-MMF is unique among climate models in that it can leverage hybrid CPU/GPU architectures on machines such as NERSC Perlmutter (https://www.nersc.gov/systems/perlmutter), which has 4 NVIDIA A100 GPUs per node. All simulations were configured to run with 4 MPI ranks and 16 OpenMP threads per node. The low-resolution (real geography and aquaplanet) cases used 2 nodes, and the high-resolution (real geography) case used 32 nodes. The throughput of these configurations was roughly 11.5 simulated years per day (sypd) for low-resolution cases and 3.3 sypd for the high-resolution case, averaged over multiple batch submissions. The total simulation length in all cases was 10 model years and 2 model months.

Boundary conditions over maritime regions are constrained by prescribed sea surface temperatures and sea ice amount. Various input data are needed for the cases with realistic topography, such as ozone concentrations and sea surface temperatures, which have been generated to reflect a climatological average of the 2005-2014 period. The aquaplanet configuration does not have a land component, but otherwise has similar input requirements using idealized data to produce a climate that is symmetric along lines of constant latitude.

# 2 Dataset and Code Access

## 2.1 Code Access

Following NeurIPS Dataset and Benchmark Track guidelines, we have uploaded our datasets to Hugging Face:

- E3SM-MMF High-Resolution Real Geography dataset:
  https://huggingface.co/datasets/LEAP/ClimSim_high-res
- E3SM-MMF Low-Resolution Real Geography dataset:
  https://huggingface.co/datasets/LEAP/ClimSim_low-res
- E3SM-MMF Low-Resolution Aquaplanet dataset:
  https://huggingface.co/datasets/LEAP/ClimSim_low-res_aqua-planet

We have documented all code (including the code to preprocess the data, create, train, and evaluate the baseline models, and visualize data and metrics) in an openly-available GitHub repository: https://leap-stc.github.io/ClimSim.

## 2.2 Variable List

All variables included in our dataset are listed in Table 1.

## 2.3 Dataset Statistics

Here, we present some distribution statistics to aid in understanding the dataset. Detailed distributions for all variables are provided in https://github.com/leap-stc/ClimSim/tree/main/dataset_statistics. These statistics are calculated for each vertical level individually for the vertically-resolved variables (e.g., state_t and state_q0001). For each variable (additionally, at each level for the vertically-resolved variables), a histogram is provided to visualize the distribution using 100 bins. Additionally, a text file accompanies each histogram, containing key statistical measures such as the mean, standard deviation, skewness, kurtosis, median, deciles, quartiles, minimum, maximum, and mode. The text file also includes the bin edges and the corresponding frequency

| In | Out | Variable | Dimensions | Units | Description |
|---|---|---|---|---|---|
| × | | pbuf_SOLIN | ncol | W/m$^2$ | Solar insolation |
| × | | pbuf_COSZRS | ncol | | Cosine of solar zenith angle |
| × | | pbuf_LHFLX | ncol | W/m$^2$ | Surface latent heat flux |
| × | | pbuf_SHFLX | ncol | W/m$^2$ | Surface sensible heat flux |
| × | | pbuf_TAUX | ncol | W/m$^2$ | Zonal surface stress |
| × | | pbuf_TAUY | ncol | W/m$^2$ | Meridional surface stress |
| × | | pbuf_ozone | lev, ncol | mol/mol | Ozone volume mixing ratio |
| × | | pbuf_N2O | lev, ncol | mol/mol | Nitrous oxide volume mixing ratio |
| × | | pbuf_CH4 | lev, ncol | mol/mol | Methane volume mixing ratio |
| × | | state_ps | ncol | Pa | Surface pressure |
| × | × | state_q0001 | lev, ncol | kg/kg | Specific humidity |
| × | × | state_q0002 | lev, ncol | kg/kg | Cloud liquid mixing ratio |
| × | × | state_q0003 | lev, ncol | kg/kg | Cloud ice mixing ratio |
| × | × | state_t | lev, ncol | K | Air temperature |
| × | × | state_u | lev, ncol | m/s | Zonal wind speed |
| × | × | state_v | lev, ncol | m/s | Meridional wind speed |
| × | | state_pmid | lev, ncol | Pa | Mid-level pressure |
| × | | cam_in_ASDIR | ncol | | Albedo for direct shortwave radiation |
| × | | cam_in_ASDIF | ncol | | Albedo for diffuse shortwave radiation |
| × | | cam_in_ALDIR | ncol | | Albedo for direct longwave radiation |
| × | | cam_in_ALDIF | ncol | | Albedo for diffuse longwave radiation |
| × | | cam_in_LWUP | ncol | W/m$^2$ | Upward longwave flux |
| × | | cam_in_SNOWHLAND | ncol | m | Snow depth over land (liquid water equivalent) |
| × | | cam_in_SNOWHICE | ncol | m | Snow depth over ice |
| × | | cam_in_LANDFRAC | ncol | | Land area fraction |
| × | | cam_in_ICEFRAC | ncol | | Sea-ice area fraction |
| | × | cam_out_NETSW | ncol | W/m$^2$ | Net shortwave flux at surface |
| | × | cam_out_FLWDS | ncol | W/m$^2$ | Downward longwave flux at surface |
| | × | cam_out_PRECSC | ncol | m/s | Snow rate (liquid water equivalent) |
| | × | cam_out_PRECC | ncol | m/s | Rain rate |
| | × | cam_out_SOLS | ncol | W/m$^2$ | Downward visible direct solar flux to surface |
| | × | cam_out_SOLL | ncol | W/m$^2$ | Downward near-IR direct solar flux to surface |
| | × | cam_out_SOLSD | ncol | W/m$^2$ | Downward visible diffuse solar flux to surface |
| | × | cam_out_SOLLD | ncol | W/m$^2$ | Downward near-IR diffuse solar flux to surface |

Table 1: Overview of input variables (first column) and output variables (second column) of the E3SM-MMF physics calculations (including the CRM) that are stored in ClimSim. The other columns indicate the variable name, dimensions, units, and a brief description. IR is short for infrared, which is also often referred to as "longwave" radiation among atmospheric scientists.

values used to generate the histogram figures. This comprehensive approach allows for a detailed analysis of the dataset's distributions.

## 2.4 Dataset Applications

Our data can benefit a broader audience beyond climate modelers wishing to explore ML for sub-grid parameterization. For climate studies, while high-frequency timestep-level outputs from simulations are rarely archived, they offer insights into convective extremes and diurnal variability. Such data opens the path to explore multi-scale interactions between rapid dynamics and broader weather and climate fluctuations. This includes a detailed examination of variables needed to constrain vertically resolved energy and water budgets and understand their variability. For the machine learning community, this dataset addresses the scarcity of large-scale regression benchmarks, common in the sciences. Such benchmarks are less common compared to prevalent industrial datasets that emphasize classification, computer vision, and NLP tasks.

## 2.5 Target Audiences

In essence, this benchmark aims to democratize and expand access to advanced climate modeling. High-potential architectures will undergo testing in the superparameterized version of the DOE's primary climate model, E3SM. Successful integration would substantially reduce computational costs for the DOE when contemplating the deployment of MMF technology in climate prediction. E3SM's external user community, typically deterred by the extensive computational demands of superparameterized simulators, also stands to benefit. Currently, only a minority with substantial computing resources can engage with such models. A successful recipe for ClimSim could thus democratize the use of explicit convection for a broader user base. If performant architectures also prove effective in the NCAR Community Earth System Model (CESM) - the world's most widely used open source climate simulator - the user base could expand significantly. Given its software similarities to E3SM, it is logical to expect that ClimSim's learnt parameterizations will be readily adaptable to CESM. Moreover, we anticipate that a successful hybrid machine learning climate simulator will bring benefits to a diverse range of industry sectors, including those vulnerable to climate risks (such as agriculture, energy, and tourism), as well as the climate risk industry itself (such as insurance and risk assessment).

## 3 Baseline Models

This section offers a detailed depiction of six baseline models. Every facet of model designs, excluding the dimensions of the input and output layers, differs among the models. We recognize that while this approach maximizes the differentiation among baseline models, such extensive degrees of freedom complicate the complete isolation of the effects arising from optimization parameter choices and those originating from the model architecture itself. In future ClimSim releases, baseline models will share more constraints (including optimization parameters) to highlight the performance difference due to model architectures.

### 3.1 Multilayer Perceptron (MLP)

A multilayer perceptron (MLP) is a basic, densely connected artificial neural network. We used KerasTuner [7] with a random search algorithm for hyperparameter optimization. The following hyperparameters were optimized: the number of hidden layers ($N_{\text{layers}}$), the number of nodes per layer ($N_{\text{nodes}}$), activation function, and batch size. The search domains were:

- $N_{\text{layers}}$: [3, 4, 5, 6, 7, 8, 9, 10, 11, 12, 13]

- $N_{\text{nodes}}$: [128, 256, 384, 512, 640, 768, 896, 1024]
- Activation function: [ReLU, LeakyReLU ($\alpha = 0.15$), eLU ($\alpha = 1.0$)]
- Batch size: [48, 96, 192, 384, 768, 1152, 1536, 2304, 3072]
- Optimizer: [Adam, RAdam, RMSprop, SGD]

Note that $N_{\text{nodes}}$ was selected independently for each hidden layer. For example, for $N_{\text{layers}} = k$, $N_{\text{nodes}}$ was drawn from the search domain $k$ times. The width of the last hidden layer was fixed at 128. The output layer utilized the linear activation function for the first 120 outputs (corresponding to the heating and moistening tendencies), and ReLU for the remaining 8 variables (corresponding positive-definite surface variables). The loss function was taken as the mean squared error (MSE), and the learning rate was defined using a cyclic scheduler, with an initial learning rate of $2.5 \times 10^{-4}$, maximum of $2.5 \times 10^{-3}$, and step size of 4 epochs.

Following Yu et al. (2023) [8], we conducted the hyperparameter search in two stages. In the first stage, a total of 8,257 randomly-drawn hyperparameter configurations were trained and evaluated with a tiny subset of the full training set, sub-sampled in the time dimension with a stride of 37. In the second stage, the top 0.2% candidates (160 hyperparameter configurations) were re-trained with a larger fraction of the full training set (sub-sampled with a stride of 7), and then evaluated for our MLP baseline. After this two-step search process, the best hyperparameter configuration was identified as: $N_{\text{layers}} = 5$, $N_{\text{nodes}} = [768, 640, 512, 640, 640]$, LeakyReLU activation, a batch size of 3,072, and RAdam optimizer. The MLP baseline has approximately 1.75 million parameters and executes 3.50 MFlops on one data point, the architecture of which is summarized in Figure 1.

To provide some context on the amount of variance in model performance that can be attributed to random effects of optimization, the top 160 models were selected from our pool of 8,257 trials and scored on the validation set; the 5th to 95th percentile range of this ensemble is shown by the error bars in Figures 2a and SI3, and by the grey shading in Figures 2b-e, SI4, and SI5.

**MLP with expanded features and targets**: We built MLP with an expanded set of input and output variables, as elaborated in Section 4.2 of the main text. For the sake of clarity, we designate an MLP model employing the subset of available variables (outlined in Section 4 of the main text) as "MLPv1," while an MLP model utilizing the expanded variables is referred to as "MLPv2." The hyperparameter optimization for MLPv2 followed a similar process as MLPv1, with the exception that the search domain of batch size was defined as [2700, 5400, 10800, 21600, 43200, 64800, 86400, 129600, 172800]. After 11,851 search trials, the best hyperparameter configuration was identified as: $N_{\text{layers}} = 3$, $N_{\text{nodes}} = [384, 1024, 640]$, ReLU activation, a batch size of 2,304, and Adam optimizer. The MLPv2 baseline has approximately 1.59 million parameters and executes 3.17 MFlops on one data point.

**MLP with the high-resolution dataset**: In conjunction with the MLP featuring expanded features and targets, we also constructed MLP models using the high-resolution dataset for both MLPv1 and MLPv2. To differentiate these models from those constructed with the low-resolution dataset, we add the suffix "-ne30" to their names. The hyperparameters for MLPv1-ne30 and MLPv2-ne30 were optimized using the same methodology as was applied to their low-resolution counterparts. For MLPv1-ne30, after 10,296 search trials, the best hyperparameter configuration was identified as: $N_{\text{layers}} = 4$, $N_{\text{nodes}} = [1024, 128, 128, 768]$, leaky ReLU activation, a batch size of 5,400, and Adam optimizer. The MLPv1-ne30 baseline has approximately 0.49 million parameters and executes 0.98 MFlops on one data point. For MLPv2-ne30, after 10,440 search trials, the best hyperparameter configuration was identified as: $N_{\text{layers}} = 3$, $N_{\text{nodes}} = [640, 128, 1024]$, ReLU activation, a batch size of 2,700, and Adam optimizer. The MLPv2-ne30 baseline has approximately 1.00 million parameters and executes 2.00 MFlops on one data point.

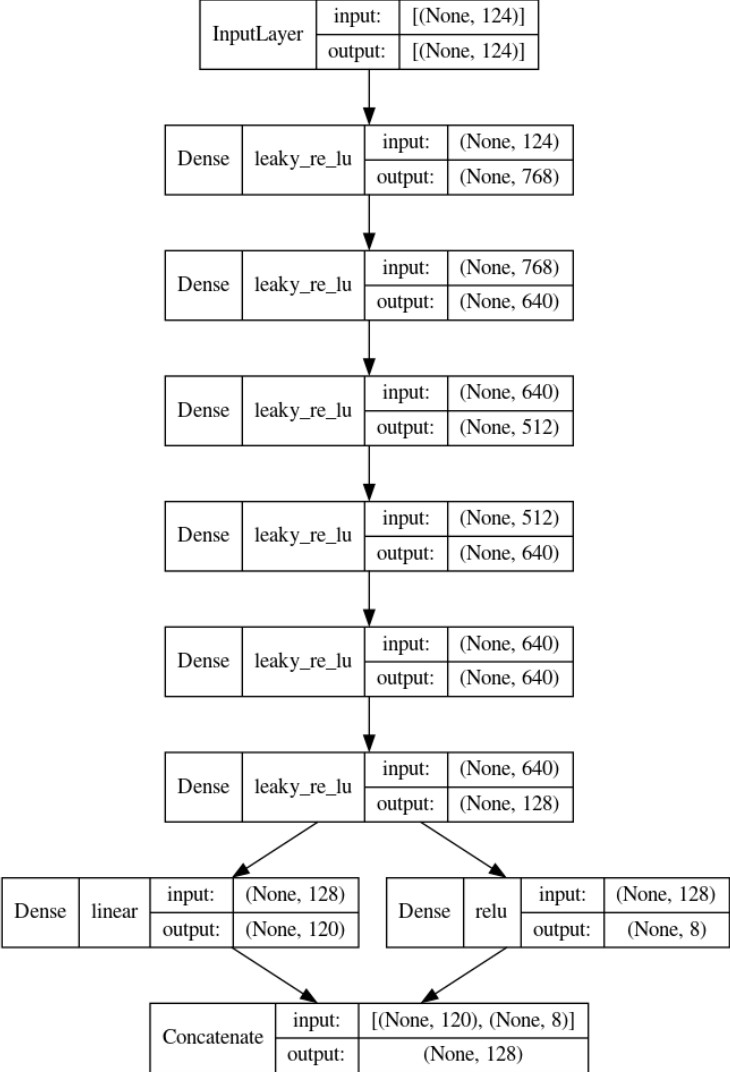

Figure 1: The architecture of the MLP baseline model.

The model performance comparison between MLPv1, MLPv2, MLPv1-ne30, and MLPv2-ne30 is presented in SI Section 8.1.

## 3.2 Randomized Prior Network (RPN)

A randomized prior network (RPN) is an ensemble model [9]. Each member of the RPN is built as the sum of a trainable and a non-trainable (so-called "prior") surrogate model; we used MLP for simplicity. Multiple replicas of the networks are constructed by independent and random sampling of both trainable and non-trainable parameters [10, 11]. RPNs also resort to data bootstrapping in order to mitigate the uncertainty collapse of the ensemble method when tested beyond the training data points [11]. Data bootstrapping consists of sub-sampling and randomization of the data each network in the ensemble sees during training. Hyperparameters of individual MLPs (i.e., $N_{\text{layers}}$, $N_{\text{nodes}}$, batch size) did not need to be tuned from scratch, and were instead chosen based on the hyperparameter search mentioned in Section 3.1. RPN ensembles of 128 networks were considered justified [10].

In particular, individual MLPs forming the RPN were considered as fully connected neural networks with $N_{\text{layers}} = 5$, $N_{\text{nodes}} = [768, 640, 512, 640, 640]$, and a batch size of 3,072, as in Section 3.1. We utilized ReLU activation (with a negative slope of 0.15) for all layers except for the output layer, where the linear activation function was used.

The MLPs were trained for a total of 13,140 stochastic gradient descent (SGD) steps using the Adam optimizer. The learning rate was initialized at $5 \times 10^{-4}$ with an exponential decay at a rate of 0.99 for every 1,000 steps. The RPN baseline has approximately 222.3 million parameters ($\sim$1.74 million per MLP) and executes 0.89 GFlops on one data point.

### 3.3 Convolutional Neural Network (CNN)

The convolutional neural network (CNN) used is a modified version of a residual network (ResNet). Each ResNet block is composed of two, 1D convolutions (Conv1D) with a $3 \times 3$ kernel using "same" padding, and an output feature map size of 406. Each Conv1D is followed by ReLU activation and dropout (with rate = 0.175). Residuals were also 1D convolved using a $1 \times 1$ kernel, and added back to the output of the main ResNet block.

The CNN composes 12 such ResNet blocks, followed by "flattening" of the feature map via a $1 \times 1$ convolution and eLU activation. Two separate Dense layers (and their corresponding activations) map the output feature map to their respective co-domains: one to $(-\infty, \infty)$ assuming that vertically-resolved variables have no defined range, and another to $[0, \infty)$ for all globally-resolved variables. These were concatenated as the output of the network.

A hyperparameter search was conducted on depth, width, kernel size, activation functions, loss functions, and normalization types using the Hyperband [12] strategy with the KerasTuner [7] framework. The search domains were:

- Model depth/number of ResNet blocks: [2, 15]
- Model width: [32, 512]
- Kernel width: [3, 5, 7, 9]
- Activation function: [GeLU, eLU, ReLU, Swish]
- Layer normalization: [True, False]
- Dropout: [0.0, 0.5]
- Optimizer: [SGD, Adam]

The CNN was trained for 10 epochs with an Adam optimizer with standard hyperparameters ($\beta_1 = 0.9$, $\beta_2 = 0.999$, $\epsilon = 1 \times 10^{-7}$). The learning rate was defined using a cyclic scheduler, with an initial learning rate of $1 \times 10^{-4}$, a maximum of $1 \times 10^{-3}$, and a step size of $2 \times \lfloor \frac{10,091,520}{12} \rfloor$. A scaling function of $\frac{1}{2.0^{x-1}}$ was applied to the scheduler per step $x$.

The hyperparameter search was conducted for 12 hours on 8 NVIDIA Tesla V100 32GB cards, with one model executing on each card. A weighted mean absolute error (MAE) was used as the loss function for optimization. We down-weighted the standard MAE loss to de-emphasize repeated scalar values provided to the network as input. The weighted MAE function is defined below:

```python
def mae_adjusted(y_true, y_pred):
    abs_error = K.abs(y_pred - y_true)
    vertical_weights = K.mean(abs_error[:,:,0:2])*(120/128)
    scalar_weights   = K.mean(abs_error[:,:,2:10])*(8/128)
    return vertical_weights + scalar_weights
```

The CNN baseline has approximately 13.2 million parameters and executes 1.59 GFlops on one data point. The architecture is visualized below in Figure 2.

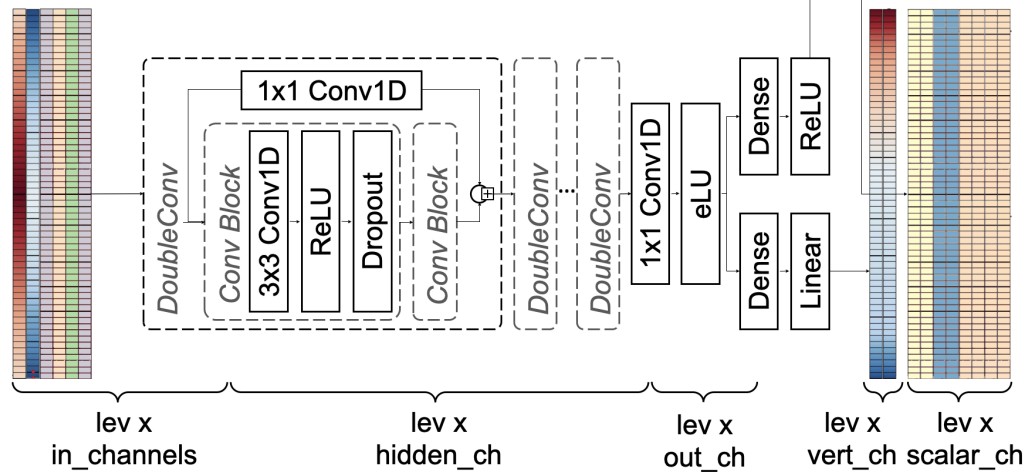

Figure 2: The ResNet-style CNN baseline is comprised of multiple ResNet blocks (i.e., DoubleConv), and applies different activation to the outputs for vertically-resolved and global variables. The channel dimensions are [in_channels, hidden_ch, out_ch, vert_ch, scalar_ch] = [6, 406, 10, 8, 2].

### 3.4 Heteroskedastic Regression (HSR)

We quantified the inherent stochasticity in the data $\mathcal{D} = \{(\mathbf{x}_1, \mathbf{y}_1), \ldots, (\mathbf{x}_n, \mathbf{y}_n)\}$, and the uncertainty in our prediction by providing a distributional prediction instead of a point estimate. In hetereoskedastic regression (HSR), this predictive distribution is modeled explicitly; here as independent Gaussians with unique mean $\mu_k$ and precision (inverse variance) $\tau_k$ for each variable. We assumed

$$\mathbf{y}_i \mid \mathbf{x}_i \sim \mathcal{N}(\mu(\mathbf{x}_i), \mathrm{Diag}(\tau(\mathbf{x}_i)^{-1})),$$

and parameterized both $\mu$ and $\tau$ as over-parameterized feed-forward neural networks (i.e., MLPs) $\hat{\mu}_\theta(\mathbf{x})$ and $\hat{\tau}_\phi(\mathbf{x})$, respectively. This yielded the corresponding predictive distribution

$$\hat{\mathbf{y}}_i \mid \mathbf{x}_i \sim \mathcal{N}(\hat{\mu}_\theta(\mathbf{x}_i), \mathrm{Diag}(\hat{\tau}_\theta(\mathbf{x}_i)^{-1})),$$

which was fitted with maximum likelihood estimation (MLE) by minimizing the objective

$$\mathcal{L}(\theta, \phi) = \frac{1}{2n} \sum_{i=1}^{n} \left[ \|\hat{\tau}_\phi(\mathbf{x}_i)(\mathbf{y}_i - \hat{\mu}_\theta(\mathbf{x}_i))\|_2^2 - \mathbf{1}^T \log(\hat{\mu}_\theta(\mathbf{x}_i)) \right].$$

Note that, due to the flexibility of the neural networks, this formulation is ill-posed. It may lead to cases of extreme overfitting where $\hat{\tau}_\phi(\mathbf{x}_i) \approx \mathbf{y}_i$, $\hat{\tau}_\phi(\mathbf{x}_i) \approx \mathbf{0}$, thus making $\mathcal{L}(\theta, \phi)$ completely unstable. Hence, we instead minimized a modified objective that included L2-regularization via

$$\mathcal{L}_{\rho, \gamma}(\theta, \phi) := \rho \mathcal{L}(\theta, \phi) + (1 - \rho) \left[ \gamma \|\theta\|_2^2 + (1 - \gamma) \|\phi\|_2^2 \right],$$

where $\rho, \gamma \in (0, 1)$ determines the trade-off between MLE estimation, mean regularization, and precision regularization. We follow [13] and set $\rho = 1 - \gamma$ to reduce the hyperparameter search domain.

Specifically, we used two MLPs with layer normalization and ReLU activation, and trained them with gradient-based stochastic optimization. To improve stability, the first third of training was

spent on exclusively training $\hat{\mu}_\theta(\mathbf{x}_i)$ with an MSE loss. To optimize hyperparameters, we selected a configuration from 300 trials with a random number of $N_{\text{layers}} = [2, 3, 4]$, $N_{\text{nodes}} = [256, 512, 1{,}024, 2{,}048]$, $\gamma$ (log-uniform in $[0.001, 0.1]$), optimizer = [SGD, Adam] with hyperparameters ($\beta_1 = 0.9$, $\beta_2 = 0.999$), learning rate $\lambda$ (log-uniform in $[10^{-6}, 10^{-3}]$), and batch size = [1024, 2048, 4096, 8192, 16384]. Each run was trained for 12 epochs total on one NVIDIA GeForce RTX 4080 16GB. We chose the run with the lowest CRPS on the validation data, yielding $N_{\text{layers}} = 4$, $N_{\text{nodes}} = 1{,}024$, $\gamma = 2.2 \times 10^{-2}$, $\lambda = 7 \times 10^{-6}$, and a batch size of 16,384, trained with Adam. The HSR baseline has approximately 6.63 million parameters and executes 6.85 MFlops per data point.

### 3.5 Conditional Variational Autoencoder (cVAE)

A conditional generative latent variable model first samples—from a prior $p(\mathbf{z})$—a point $\mathbf{z}$ in a low-dimensional latent space, which then informs a conditional distribution $p_\theta(\mathbf{y}|\mathbf{z}, \mathbf{x})$ over the target domain. This allows for a complex and flexible predictive distribution. In our case, we used feed-forward neural networks (i.e., MLPs) $\mu_\theta(\mathbf{z}, \mathbf{x})$ and $\sigma_\theta(\mathbf{z}, \mathbf{x})$ with combined parameters $\theta$ and model:

$$\mathbf{z} \sim \mathcal{N}(\mathbf{0}, \mathcal{I})$$
$$\mathbf{y}|\mathbf{z}, \mathbf{x} \sim \mathcal{N}\big(\mu_\theta(\mathbf{z}, \mathbf{x}), \text{Diag}(\sigma_\theta(\mathbf{x})^2)\big) \tag{2}$$

To fit the model to data $\mathcal{D} = \{(\mathbf{x}_1, \mathbf{y}_1), \ldots, (\mathbf{x}_n, \mathbf{y}_n)\}$, we minimized the negative evidence lower bound (NELBO) $\mathcal{L}_\theta(\mathbf{q})$ that bounds the intractable negative marginal likelihood from above via

$$\mathcal{L}_\theta(q) := -\mathbb{E}_{\mathbf{z}_i \sim q}\left[\log \frac{p_\theta(\mathbf{y}_i, \mathbf{z}_i | \mathbf{x}_i)}{q(\mathbf{z}_i | \mathbf{x}_i)}\right] = -\log p_\theta(\mathbf{y}_i | \mathbf{x}_i) + \underbrace{\text{KL}\big(q \,\|\, p_\theta(\mathbf{z}_i | \mathbf{y}_i, \mathbf{x}_i)\big)}_{\geq 0},$$

using an approximation $q$ to the posterior $p_\theta(\mathbf{z}_i|\mathbf{y}_i, \mathbf{x}_i)$. The conditional variational autoencoder (cVAE) [14] uses amortized variational inference to optimize $\theta$ and $q$ jointly by approximating the latter with e.g., $q_\psi(\mathbf{z}_i) = \mathcal{N}\big(g_\psi(\mathbf{x}_i), \text{Diag}(h_\psi(\mathbf{x}_i)^2)\big)$, where we again chose $g_\psi(\mathbf{x}_i)$ and $h_\psi(\mathbf{x}_i)$ to be MLPs. This allowed us to optimize for $\theta$ and $\psi$ by minimizing

$$\mathcal{L}_\theta(q) \overset{\beta=1}{=} \mathbb{E}_{\mathbf{z}_i \sim q_\psi}\left[\frac{1}{2}\left\|\frac{\mathbf{y}_i - \mu_\theta(\mathbf{z}_i, \mathbf{x}_i)}{\sigma_\theta(\mathbf{z}_i, \mathbf{x}_i)}\right\|_2^2 + \mathbf{1}^T \log\big(\sigma_\theta(\mathbf{z}_i, \mathbf{x}_i)\big)\right] + \beta\text{KL}(q_\psi(\mathbf{z}_i) \,\|\, p(\mathbf{z}_i)) + \text{const}$$

with a Monte Carlo approximation by first sampling $\mathbf{z}_i$ (once) from the variational encoder $q_\psi(\mathbf{z}_i)$. After which, we decoded the predictive mean and standard deviation with the variational decoder $\mu_\theta(\mathbf{z}, \mathbf{x})$ and $\sigma_\theta(\mathbf{z}, \mathbf{x})$. We then computed NELBO as a sum of a reconstruction term and a KL term that regularizes the latent space, averaged over all samples, and back-propagated the gradients. By letting $\beta$ be a hyperparameter, we manually determined the trade-off between reconstruction quality and latent space structure. Finally, at inference time, we used Equation 2 to sample from the predictive distribution

$$p_\theta(\hat{\mathbf{y}}|\mathbf{x}) = \int p_\theta(\hat{\mathbf{y}}|\mathbf{x}, \mathbf{z})p(\mathbf{z})\, d\mathbf{z}.$$

For both the variational encoder and decoder, we used an MLP with layer normalization, ReLU activation, dropout with $p = 0.05$, and two branching final layers that produced the mean and standard deviation, respectively. We trained both MLPs jointly—with gradient-based stochastic optimization—on the objective described above.

To optimize hyperparameters, we ran 300 trials with a random number of hidden layers $N_{\text{layers}} = [2, 3, 4]$, $N_{\text{nodes}} = [256, 512, 1024, 2048]$, size of the latent space = [4, 8, 16, 32], $\beta$ (log-uniform in $[0.01, 10]$), optimizer = [SGD, Adam] with ($\beta_1 = 0.9$, $\beta_2 = 0.0999$), learning rate $\lambda$ (log-uniform in $[10^{-6}, 10^{-3}]$), L2 regularization $\alpha$ (log-uniform in $[10^{-6}, 10^{-3}]$), and batch size = [1024, 2048, 4096, 8192,

16384]. Each run was trained for 5 epochs total on one NVIDIA GeForce RTX 4080 16GB. The run with the lowest CRPS on the validation data yielded $N_{\text{layers}} = 3$, $N_{\text{nodes}} = 1{,}024$, and a batch size of 4,096, trained with Adam. In a second step, we fixed these hyperparameters and further fine-tuned $\beta$, $\lambda$, and $\alpha$ by training for 20 epochs every time, for 10 trials. We found the best model with $\beta = 0.5$, $\lambda = 5 \times 10^{-5}$, $\alpha = 10^{-3}$. The cVAE baseline has approximately 4.9 million parameters and executes 4.88 MFlops per data point.

## 3.6 Encoder-Decoder (ED)

The Encoder-Decoder (ED) is an adjusted version of the ED presented in Behrens et al. (2022) [15]. We keep all tuneable hyperparameters except for the learning rate and the node sizes of input and output layer of ED fixed to the original values that were optimized with a detailed hyperparameter search for the superparameterization of the Community Atmosphere Model version 3 in an aquaplanet setup [15]. The Encoder consists of 6 hidden fully-connected layers. The Encoder decreases progressively the dimensionality of the input variables down to 5 nodes in the latent space of the network. These 5 latent nodes are the only input to the decoding part of ED. The Decoder maps the information from the latent space back to 128 nodes in the output layer through 6 progressively wider fully-connected hidden layers [15]. We train ED over 40 epochs with a learning rate step after each $7^{\text{th}}$ epoch, which reduces the learning rate by factor 5 [15]. The adjusted initial learning rate has a value of $1 \times 10^{-4}$. The batch size has a value of 714 samples. As activation functions of all hidden layers we use ReLU and the output layer of the Decoder is ELU-activated [15]. As an optimizer during training we use Adam. As a loss function of ED we use a MSE loss and as additional metric the MAE during training. The following list summarizes the key hyperparameters of ED:

- Learning rate: $1 \times 10^{-4}$, learning rate decrease after every $7^{\text{th}}$ epoch

- Batch size: 714

- Latent space width: 5 Nodes

- Encoder node size: [124, 463, 463, 232, 116, 58, 29, 5]

- Decoder node size: [5, 29, 58, 116, 232, 463, 463, 128]

- Encoder activation functions: [Input, ReLU, ReLU, ReLU, ReLU, ReLU, ReLU, ReLU]

- Decoder activation functions: [Input, ReLU, ReLU, ReLU, ReLU, ReLU, ReLU, ELU]

- Optimizer: Adam

To prevent overfitting we shuffle the training data set before each epoch. ED baseline has approximately 832,000 parameters, with 415,000 parameters in the Encoder and 417,000 parameters in the Decoder. In total, ED executes 1.66 MFlops per data point, with 829 kFLops per data point for the Encoder and 832 kFlops per data point for the Decoder.

## 3.7 Inference Cost

|  | CNN | ED | HSR | MLP | RPN | cVAE |
|---|---|---|---|---|---|---|
| Number of Parameters | 13,200,000 | 832,000 | 6,630,000 | 1,750,000 | 222,300,000 | 4,900,000 |
| MFlops Per Data Point | 1590 | 1.66 | 6.85 | 3.50 | 890 | 4.88 |

Table 2: The number of learnable parameters and Megaflops (MFlops) per data point for each of the six baseline models.

# 4 Baseline Model Evaluations

## 4.1 Metrics

### 4.1.1 Deterministic Metrics

Mean Absolute Error (MAE):

$$\text{MAE} = \frac{1}{n} \sum_{i=1}^{n} |X_i - y| \tag{3}$$

Root Mean Squared Error (RMSE):

$$\text{RMSE} = \sqrt{\frac{1}{n} \sum_{i=1}^{n} (X_i - y)^2} \tag{4}$$

Coefficient of Determination ($R^2$):

$$R^2 = 1 - \frac{\sum_{i=1}^{n}(X_i - y)^2}{\sum_{i=1}^{n}(X_i - \bar{X})^2} \tag{5}$$

In Equations 3–5, $X_i$ and $y$ represent the true and predicted values, respectively. The mean of the true values of the dependent variable is denoted by $\bar{X}$.

### 4.1.2 Stochastic Metric (CRPS)

The continuous ranked probability score (CRPS) is a generalization of the MAE for distributional predictions. CRPS penalizes over-confidence in addition to inaccuracy in ensemble predictions—a lower CRPS is better. For each variable, it compares the ground truth target $y$ with the cumulative distribution function (CDF) $F$ of the prediction via

$$\text{CRPS}(F, y) := \int \left( F(x) - \mathbf{1}_{\{x \geq y\}} \right)^2 dx$$

$$= \mathbb{E}[|X - y|] - \frac{1}{2} \mathbb{E}[|X - X'|],$$

where $X, X' \sim F$ are independent and identically distributed ($iid$) samples from the distributional prediction. We use the non-parametric "fair estimate to the CRPS" [16], estimating $F$ with the empirical CDF of $n = 32$ $iid$ samples $X_i \sim F$:

$$\hat{\text{CRPS}}(\mathbf{X}, y) := \frac{1}{n} \sum_{i=1}^{n} |X_i - y| - \frac{1}{2n(n-1)} \sum_{i=1}^{n} \sum_{j=1}^{n} |X_i - X_j| \tag{6}$$

The first term in Equation 6 is the MAE between the target and samples of the predictive distribution, while the second term is small for small predictive variances, vanishing completely for point estimates. Note that this definition extends to ensemble models, where we take the prediction of each ensemble member as a sample of an implicit predictive distribution.

## 4.2 Results

MAE and $R^2$ of the baseline models are presented in the main text (e.g., Table 1 and Figure 2 in the main text). Here, we show RMSE and CRPS in Table 3 and Figures 3, 4, and 5.

We also present the spatial structure of the metrics. Figure 6 shows the latitude-height structure of $R^2$.

| Variable | RMSE [W/m$^2$] | | | | | | CRPS [W/m$^2$] | | | | | |
|---|---|---|---|---|---|---|---|---|---|---|---|---|
| | CNN | ED | HSR | MLP | RPN | cVAE | CNN | ED | HSR | MLP | RPN | cVAE |
| $dT/dt$ | **4.369** | 4.696 | 4.825 | 4.421 | 4.482 | 4.721 | – | – | **2.158** | – | 2.305 | 2.708 |
| $dq/dt$ | **7.284** | 7.643 | 7.896 | 7.322 | 7.518 | 7.780 | – | – | **3.645** | – | 4.100 | 4.565 |
| NETSW | 36.91 | 28.537 | 37.77 | **26.71** | 33.60 | 38.36 | – | – | **14.62** | – | 14.82 | 20.53 |
| FLWDS | 10.86 | 9.070 | 8.220 | **6.969** | 7.914 | 8.530 | – | – | 4.561 | – | **4.430** | 6.732 |
| PRECSC | 6.001 | 5.078 | 6.095 | **4.734** | 5.511 | 6.182 | – | – | 2.905 | – | **2.729** | 3.513 |
| PRECC | 85.31 | 76.682 | 90.64 | **72.88** | 76.58 | 88.71 | – | – | 34.30 | – | **30.08** | 40.17 |
| SOLS | 22.92 | 17.999 | 23.61 | **17.40** | 20.61 | 23.27 | – | – | 8.369 | – | **8.309** | 11.91 |
| SOLL | 27.25 | 22.540 | 27.78 | **21.95** | 25.22 | 27.81 | – | – | **10.14** | – | 10.49 | 14.42 |
| SOLSD | 12.13 | 9.917 | 12.40 | **9.420** | 11.00 | 12.64 | – | – | 4.773 | – | **4.649** | 5.945 |
| SOLLD | 12.10 | 10.417 | 12.47 | **10.12** | 11.25 | 12.63 | – | – | **4.599** | – | 4.682 | 5.925 |

Table 3: Globally-averaged RMSE and CRPS. Each metric is calculated at each grid point, then horizontally-averaged and (for $dT/dt$ and $dq/dt$) vertically-averaged. The units of non-energy flux variables are converted to a common energy unit, W/m$^2$, following Section 5.2. Best model performance for each variable is highlighted in bold.

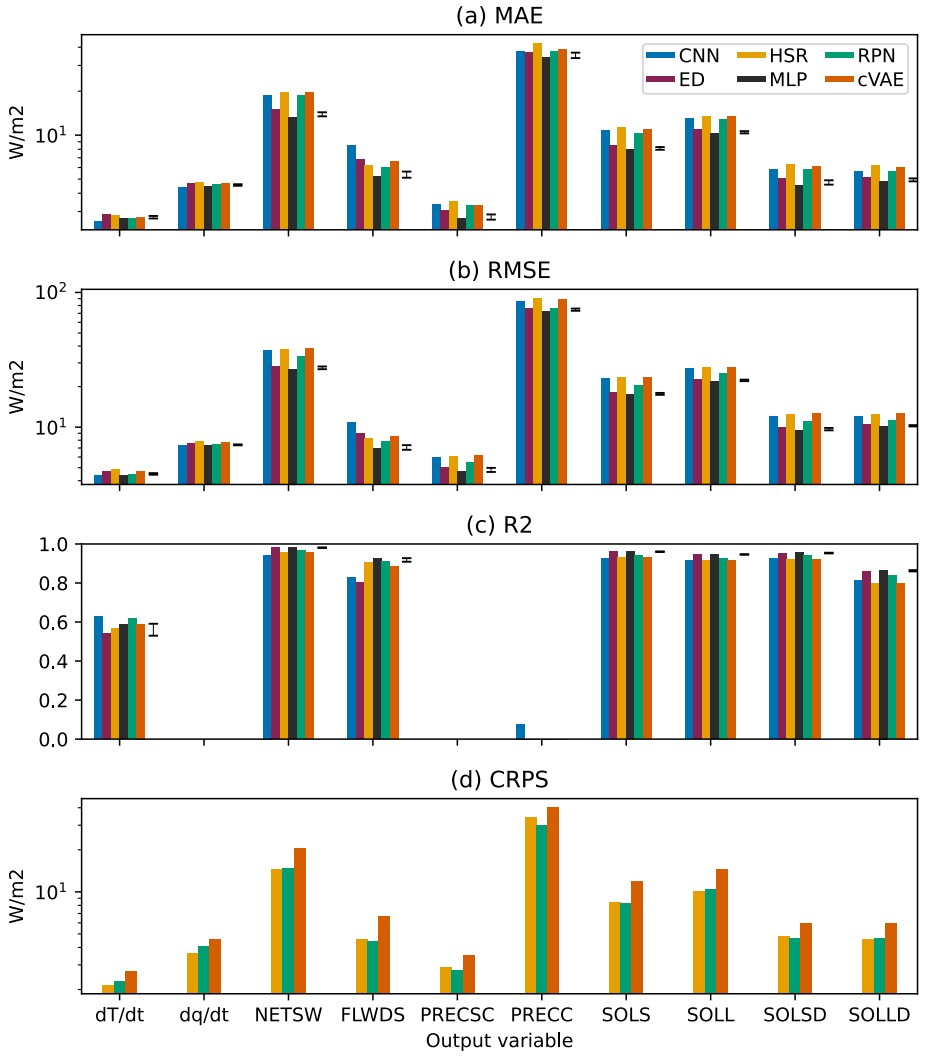

Figure 3: Averaged (a) MAE, (b) RMSE, (c) R$^2$, and (d) CRPS. Each metric is calculated at each grid point, then horizontally-averaged and (for $dT/dt$ and $dq/dt$) vertically-averaged. For MAE, RMSE, and CRPS, the units of non-energy flux variables are converted to a common energy unit, W/m$^2$, following Section 5.2. Negative values are not shown for R$^2$. Error bars show the 5- to 95-percentile range of MLP.

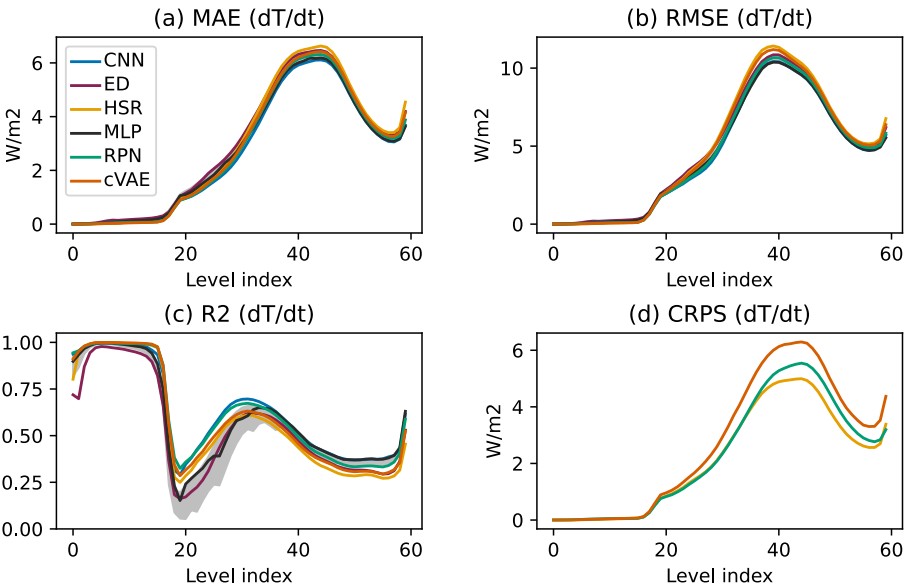

Figure 4: Vertical structures of horizontally-averaged (a) MAE, (b) RMSE, (c) $R^2$, and (d) CRPS of $dT/dt$. For MAE, RMSE, and CRPS, the units of non-energy flux variables are converted to a common energy unit, W/m$^2$, following Section 5.2. Negative values are not shown for $R^2$. Grey shadings show the 5- to 95-percentile range of MLP.

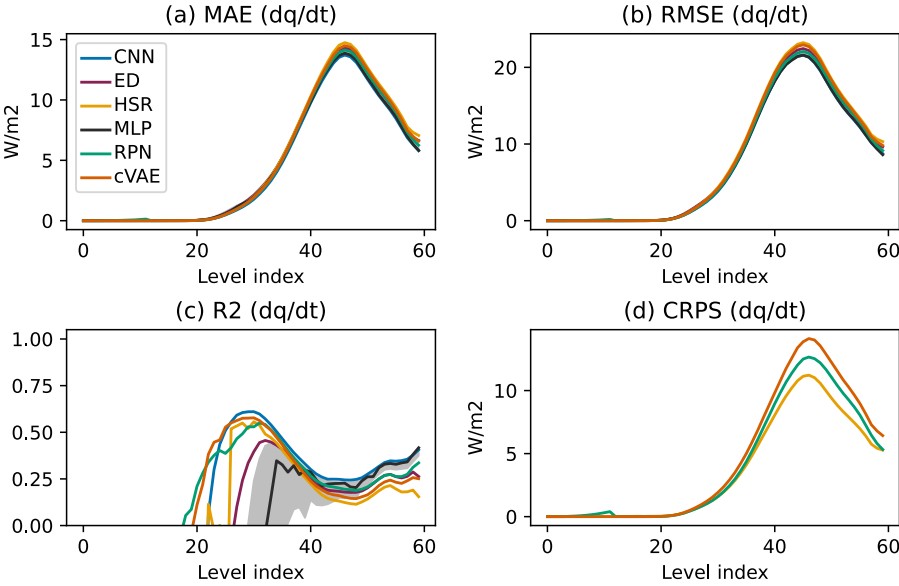

Figure 5: The vertical structures of horizontally-averaged (a) MAE, (b) RMSE, (c) $R^2$, and (d) CRPS of $dq/dt$. For MAE, RMSE, and CRPS, the units of non-energy flux variables are converted to a common energy unit, W/m$^2$, following Section 5.2. Negative values are not shown for $R^2$. Grey shadings show the 5- to 95-percentile range of MLP.

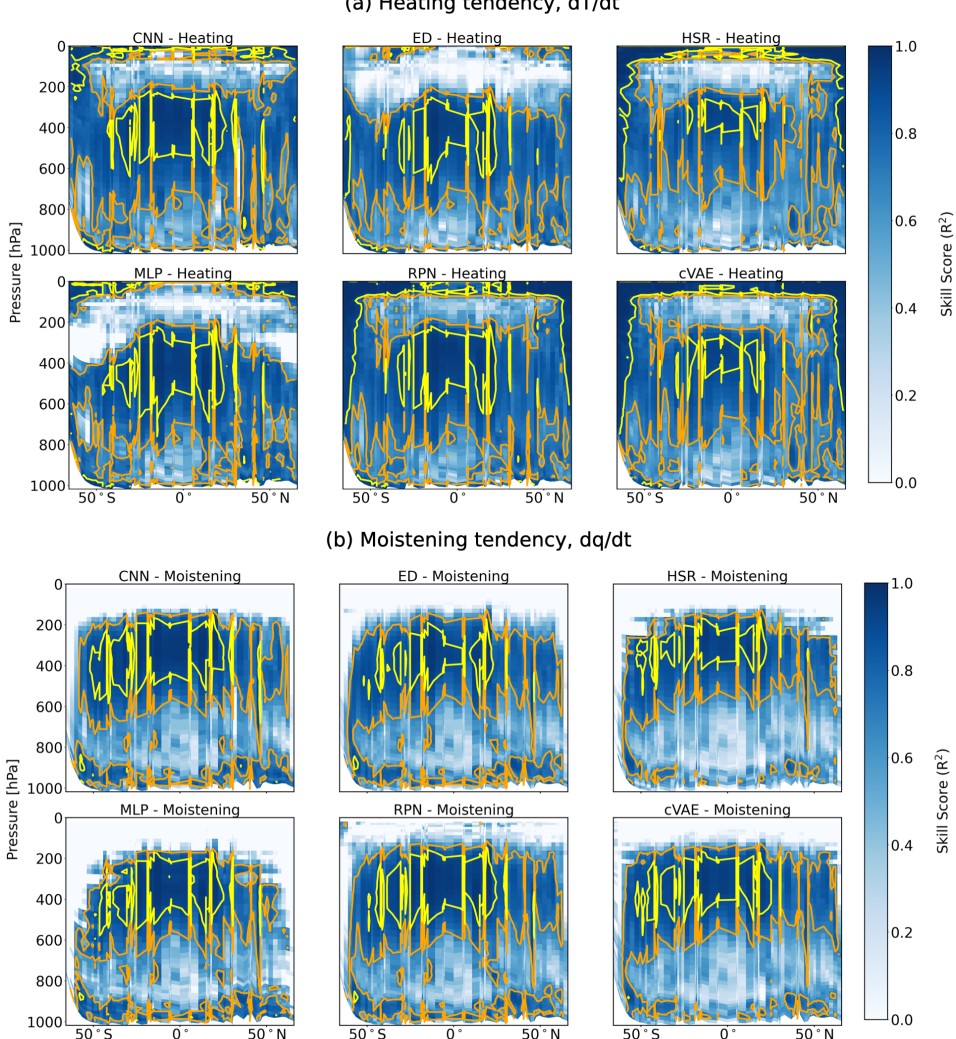

Figure 6: $R^2$ of daily-mean, zonal-mean (a) heating tendency and (b) moistening tendency. Yellow contours surround regions of $> .9R^2$ while orange contours surround regions of $> .7R^2$. Negative values are not plotted (white). Sin(latitude) is used for x-axis to account for the curvature of Earth. The pressure levels on Y-axis are approximated values.

## 4.3 Fit Quality

Scatter plots of truth versus prediction are shown in this section (Figures SI9 to SI16 in SI Section 8). While many variables exhibit consistent fit quality, some show notable variability between baselines, as seen with snow precipitation rate predictions. The performance of our optimized deterministic baseline (MLP) suggests these issues are avoidable. However, note that our prediction problem has a multi-variate and multi-dimensional nature.

# 5 Guidance

## 5.1 Physical Constraints

Mass and energy conservation are important criteria for Earth system modeling. If these terms are not conserved, errors in estimating sea level rise or temperature change over time may become as large as

the signals we hope to measure. Enforcing conservation on emulated results helps constrain results to be physically plausible and reduce the potential for errors accumulating over long time scales.

In the atmospheric component of the E3SM climate model, mass is composed of "dry air" (i.e., well-mixed gases such as molecular nitrogen and oxygen) and water vapor. During the physics parameterizations we seek to emulate, there is no lateral exchange of mass across columns of the host model, and the model assumes that the total mass in each column and level remains unchanged. Thus, while surface pressure (`state_ps`) is part of the state structure we seek to emulate, that surface pressure component must be held fixed. The water mass, however, is not held fixed, requiring fictitious sources and sinks of dry air, which are corrected later in the model—outside of the "emulated" part of the code—and is not addressed within the emulator.

Changes in column water mass should balance the sources and sinks of water into and out of the column through surface fluxes. The surface source of water is an input to the emulator via the `cam_in` structure. The surface sink of water is generated by the model, and hence emulated in our case. The net surface water flux (source minus sink) should be equal to the tendency of water mass within the column (7). The mass of water is held in five separate terms within the `state` structure: water vapor ($q_v$), cloud liquid condensate ($q_l$), cloud ice ($q_i$), rain ($q_r$), and snow ($q_s$). These terms are held as ratios of their mass to the sum of dry air plus water vapor (referred to as specific humidity). The "$\delta$" refers to the difference (after minus before computation) in each quantity owing to the CRM physics. The layer mass (sum of dry air and water vapor) of level $k$ is equal to the pressure thickness of that layer $\Delta p_i$ (the difference between top and bottom interface pressure for level $i$) divided by the gravitational acceleration $g$ (assumed constant). The timestep length is $\delta t$. In addition to conserving water mass, we required each individual water constituent to remain greater than or equal to zero in every layer within the column. In Equation 7, $E$ is the surface source of water (evapotranspiration) and $P$ is the surface sink of water (precipitation):

$$\sum_i \left( \delta q_v + \delta q_l + \delta q_i + \delta q_r + \delta q_s \right) \frac{\Delta p_i}{g \delta t} = E - P \tag{7}$$

For the portion of the code that we try to emulate, the water source $E$ is not applied such that the only surface flux to account for when constraining water conservation is the precipitation flux ($P$, `cam_out_PRECC`). Unfortunately, only the input and output state variables for water vapor (`state_q0001`), cloud liquid (`state_q0002`), and cloud ice (`state_q0003`) are available. Additional storage terms related to precipitating water that have not exited the column over the course of a model timestep are unavailable in the current output. Therefore, we are unable to exactly enforce water conservation. Estimates show relative errors of a couple percent resulting from the lack of these precipitation mixing ratios. We can still require that the relative error be small. To accomplish this, we compared the "expected" total water, based on the combination of the input and surface fluxes, to the predicted total water. In the equations below, superscript $o$ denotes output and superscript $i$ denotes input:

$$\text{Total Water (Actual)} = \sum_i \left( \delta q_v^o + \delta q_l^o + \delta q_i^o \right) \frac{\Delta p_i}{g}$$

$$\text{Total Water (Expected)} = \sum_i \left( \delta q_v^i + \delta q_l^i + \delta q_i^i \right) \frac{\Delta p_i}{g} - P \delta t$$

$$\text{Relative Error} = \frac{\text{Total Water (Expected)} - \text{Total Water (Actual)}}{\text{Total Water (Actual)}}$$

We required the model to keep the relative error small (e.g., below 5%). Anything further is beyond the limit of the current data.

Like mass conservation, energy conservation can generally be enforced by requiring that the total change within the column is exactly balanced by the fluxes into and out of that column. Because the emulator does not predict upwelling radiative fluxes at the model top (a sink term for energy), we do not have the boundary conditions necessary to constrain column energy tendencies. However, we still required certain criteria be met for physical consistency. First, the downwelling surface shortwave radiative flux cannot exceed the downwelling shortwave flux at the model top (prescribed input `pbuf_SOLIN`). Likewise, the net surface shortwave flux should also be bounded between zero (100% reflection) and the surface downwelling shortwave flux (100% absorption). Additionally, the downwelling longwave flux should not exceed the blackbody radiative flux from the warmest temperature in the column.

## 5.2  Unit Conversion and Weighting for Interpretable Evaluation

To facilitate the objective evaluation of the model's prediction, we provided a weight tensor of shape $(d_o, N_{\boldsymbol{x}})$ to convert raw outputs to area-weighted outputs with consistent energy flux units [W/m$^2$]. More details are given below.

To ensure that our evaluation takes the Earth's spherical geometry into account, we designed an area weighting factor $a$ that depends on the horizontal position $\boldsymbol{x}$:

$$a\left(\boldsymbol{x}\right) = \mathcal{A}_{\mathrm{col}}\left(\boldsymbol{x}\right) / \left\langle \mathcal{A}_{\mathrm{col}} \right\rangle_{\boldsymbol{x}}$$

where $\mathcal{A}_{\mathrm{col}}$ is the area of an atmospheric column and $\left\langle \mathcal{A}_{\mathrm{col}} \right\rangle_{\boldsymbol{x}}$ the horizontal average of all atmospheric columns' areas. This formula gives more weight to outputs if their grid cell has a larger horizontal area. To ensure that our evaluation is physically-consistent, we convert all predicted variables to energy flux units $\left[\mathrm{W/m}^2\right]$ (power per unit area). This has to be done for each variable separately.

- For heating tendencies $\dot{T}\,[\mathrm{K/s}]$, which depends on the horizontal position $\boldsymbol{x}$ and vertical level lev, this was be done using the specific heat capacity constant at constant pressure $c_p\,[\mathrm{J/\left(K \times kg\right)}]$, where $\Delta p_i\,[\mathrm{Pa}]$ is the layer's pressure thickness, calculated as the difference between the pressure at the layer's top and bottom interfaces:

$$\dot{T}\left[\mathrm{W/m}^2\right] = \frac{c_p}{g} \times a\left(\boldsymbol{x}\right) \times \Delta p_i\left(\mathrm{lev}\right) \times \dot{T}\,[\mathrm{K/s}]$$

- For water concentration tendencies $\dot{q}\left[\mathrm{s}^{-1}\right]$, which also depends on $\boldsymbol{x}$ and lev, this was be done using the latent heat of vaporization of water vapor at constant pressure $L_v\,[\mathrm{J/kg}]$:

$$\dot{q}\left[\mathrm{W/m}^2\right] = \frac{L_v}{g} \times a\left(\boldsymbol{x}\right) \times \Delta p_i\left(\mathrm{lev}\right) \times \dot{q}\left[\mathrm{s}^{-1}\right]$$

Note that there is some level of arbitrariness, as the exact latent heat depends on which water phase is assumed to calculate the energy transfer. Here, we chose to weigh all phases using $L_v$ to give them comparable weights in the evaluation metrics.

- For momentum tendencies $\dot{u}\left[\mathrm{m/s}^2\right]$, which also depend on $\boldsymbol{x}$ and lev, we used a characteristic wind magnitude $|\boldsymbol{U}|\,[\mathrm{m/s}]$ to convert these tendencies into turbulent kinetic energy fluxes, in units W/m$^2$, making them comparable to $\dot{T}\left[\mathrm{W/m}^2\right]$ and $\dot{q}\left[\mathrm{W/m}^2\right]$:

$$\dot{u}\left[\mathrm{W/m}^2\right] = \frac{|\boldsymbol{U}|}{g} \times a\left(\boldsymbol{x}\right) \times \Delta p_i\left(\mathrm{lev}\right) \times \dot{u}\left[\mathrm{m/s}^2\right]$$

Note that there is some level of arbitrariness in the choice of $|\boldsymbol{U}|\,[\mathrm{m/s}]$, which could e.g., be chosen so that the variances of $\dot{u}\,\left[\mathrm{W/m^2}\right]$ and $\dot{T}\,\left[\mathrm{W/m^2}\right]$ are comparable.

- Precipitation rate variables $P\,[\mathrm{m/s}]$ were also be converted to energy fluxes using $L_v$ and the density of liquid water $\rho_w\,\left[\mathrm{kg/m^3}\right]$ (or the density of snow/ice for solid precipitation), though they do not require vertical integration:

$$P\left[\mathrm{W/m^2}\right] = L_v \times \rho_w \times a\left(\boldsymbol{x}\right) \times P\left[\mathrm{m/s}\right]$$

- Finally, surface energy fluxes $\mathcal{F}\left[\mathrm{W/m^2}\right]$ were simply multiplied by $a\left(\boldsymbol{x}\right)$ to account for area-weighting.

Note that while these choices ensured unit consistency, facilitating the physical interpretation of our evaluation metrics, we recommend tailoring the exact choice of physical constants to the application of interest.

### 5.3 Additional Guidance

**Stochasticity and Memory:** The results of the embedded convection calculations regulating $d_o$ come from a chaotic dynamical system and thus could be worthy of architectures and metrics beyond the deterministic baselines in this paper. These solutions are likewise sensitive to sub-grid initial state variables from an interior nested spatial dimension that have not been included in our data.

**Temporal Locality:** Incorporating the previous timesteps' target or feature in the input vector inflation could be beneficial as it captures some information about this convective memory and utilizes temporal autocorrelations present in atmospheric data.

**Causal Pruning**: A systematic and quantitative pruning of the input vector based on objectively assessed causal relationships to subsets of the target vector has been proposed as an attractive preprocessing strategy, as it helps remove spurious correlations due to confounding variables and optimize the machine learning (ML) algorithm [17].

**Normalization:** Normalization that goes beyond removing vertical structure could be strategic, such as removing the geographical mean (e.g., latitudinal, land/sea structure) or composite seasonal variances (e.g., local smoothed annual cycle) present in the data. For variables exhibiting exponential variation and approaching zero at the highest level (e.g., metrics of moisture), log-normalization might be beneficial.

## 6 Other Related Work

Several benchmark datasets have been developed to facilitate AI tasks in weather and climate. ClimateNet [18] and Extremeweather [19] were both designed for AI-based feature detection of extreme weather events in forecasts of Earth's future climate made using conventional climate models. WeatherBench [20] provides data specifically designed for data-driven weather forecasting, focusing on periods ranging from 3 to 5 days into the future. PDEBench [21] provides data from numerical simulations of several partial differential equations (PDEs) for benchmarking AI PDE emulators. ClimateBench [22] was designed for emulators that produce annual mean global predictions of temperature and precipitation given greenhouse gas concentrations and emissions. ClimART [23] was designed for the development of radiative energy transfer parameterization emulators for use in weather and climate modeling. These benchmark datasets play a vital role in advancing AI and ML research within the weather and climate domains.

ClimSim, a dataset for parameterization emulators trained on high-resolution data from small-scale embedded models, is unique compared to other benchmark datasets designed for emulators in

climate simulation (ClimateBench, ClimART, and PDEBench). While PDEBench provides data for developing AI emulators of the same PDEs commonly used in climate simulation, ClimSim is uniquely tailored to address the challenging task of replacing a sophisticated parameterization for the combined effects of clouds, rain, radiation, and storms. Specifically, models trained using ClimSim will learn to emulate the nonlinear effect of clouds, rain, and storms resolved on the 1 km (20 s) space (time) scale, which is a collection of hundreds of equations rather than one, to represent their upscale impacts on the 100 km (30 min) scale. Hybrid simulation is also the goal of ClimART, which is designed specifically for the narrower and less computationally costly task of radiative energy transfer parameterization, rather than cloud and rain emulators. ClimateBench, on the other hand, is not an attempt at hybrid simulation, but rather for "whole-model" emulators that reproduce the annual mean global predictions of climate that a conventional climate model would simulate given unseen greenhouse gas concentrations and emissions. This does not attempt to sidestep Moore's Law or admit previously unattainable resolution, i.e., any error or bias related to the parameterizations used to create the training data are part of what is learned by the emulator.

In contrast, the goal of ClimSim is to develop an emulator for the *explicitly resolved* effect of clouds and storms on climate, so that, down the road, the emulator can be used to replace parameterizations in a climate model, enabling more realistic climate simulation without the typical computational overhead. ClimSim builds off work by a few climate scientists who have been exploring since 2017 to apply ML for hybrid multi-scale climate modeling. [24] first demonstrated that using simple ML models, and a simple atmosphere test-bed, certain atmospheric patterns of convective heating and moistening could be effectively predicted, particularly in the tropics and mid-latitude storm tracks. However, when these models were integrated into broader climate simulations, except for lucky fits that demonstrated the exciting potential for success [25], issues related to stability arose, a common problem when constructing hybrid climate models. Various methods were tried to improve the stability, such as coupling multiple models together and searching for better model architectures [26, 27]. These efforts led to improved error rates in the predictions. More recently, researchers have expanded this work into real-world settings, using more advanced ML architectures [28–30]. Wang et al. (2022) [31] even managed to create a deep-learning model that showed hybrid stability over a decade under real-world conditions. While this hybrid model had a few biases, it was successful in capturing some aspects of climate variability. Additionally, work has been done to compress input data to avoid causal confounders while maintaining accuracy [17], use latent representations that account for stochasticity [15], and enforce physical constraints within these models [32], all of which could potentially improve their reliability.

# 7 Datasheet

## 7.1 Motivation

1. **For what purpose was the dataset created?** *Our benchmark dataset was created to serve as a foundation for developing robust frameworks that emulate parameterizations for cloud and extreme rainfall physics and their interaction with other sub-resolution processes.*

2. **Who created the dataset and on behalf of which entity?** *The dataset was developed by a consortium of climate scientists and ML researchers listed in the author list.*

3. **Who funded the creation of the dataset?** *The main funding body is the National Science Foundation (NSF) Science and Technology Center (STC) Learning the Earth with Artificial Intelligence and Physics (LEAP). Other funding sources of individual authors are listed in the acknowledgment section of the main text.*

## 7.2 Distribution

1. **Will the dataset be distributed to third parties outside of the entity (e.g., company, institution, organization) on behalf of which the dataset was created?** *Yes, the dataset is open to the public.*

2. **How will the dataset will be distributed (e.g., tarball on website, API, GitHub)?** *The dataset will be distributed through Hugging Face and the code used for developing baseline models through GitHub.*

3. **Have any third parties imposed IP-based or other restrictions on the data associated with the instances?** *No.*

4. **Do any export controls or other regulatory restrictions apply to the dataset or to individual instances?** *No.*

## 7.3 Maintenance

1. **Who will be supporting/hosting/maintaining the dataset?** *NSF-STC LEAP will support, host, and maintain the dataset.*

2. **How can the owner/curator/manager of the dataset be contacted (e.g., email address)?** *The owner/curator/manager(s) of the dataset can be contacted through following emails: Sungduk Yu (sungduk@uci.edu), Michael S. Pritchard (mspritch@uci.edu) and LEAP (leap@columbia.edu).*

3. **Is there an erratum?** *No. If errors are found in the future, we will release errata on the main web page for the dataset (https://leap-stc.github.io/ClimSim).*

4. **Will the dataset be updated (e.g., to correct labeling errors, add new instances, delete instances)?** *Yes, the datasets will be updated whenever necessary to ensure accuracy, and announcements will be made accordingly. These updates will be posted on the main web page for the dataset (https://leap-stc.github.io/ClimSim).*

5. **If the dataset relates to people, are there applicable limits on the retention of the data associated with the instances (e.g., were the individuals in question told that their data would be retained for a fixed period of time and then deleted?)** *N/A*

6. **Will older version of the dataset continue to be supported/hosted/maintained?** *Yes, older versions of the dataset will continue to be maintained and hosted.*

7. **If others want to extend/augment/build on/contribute to the dataset, is there a mechanisms for them to do so?** *No.*

## 7.4 Composition

1. **What do the instance that comprise the dataset represent (e.g., documents, photos, people, countries?)** *Each instance includes both input and output vector pairs. These inputs and outputs are instantaneous snapshots of atmospheric states surrounding detailed numerical calculations to be emulated.*

2. **How many instances are there in total (of each type, if appropriate)?** *The high-resolution dataset (ClimSim_high-res) includes 5,676,480,000 instances, and each low-resolution dataset (ClimSim_low-res and ClimSim_low-res_aqua-planet) includes 100,915,200 instances.*

3. **Does the dataset contain all possible instances or is it a sample of instances from a larger set?** *The datasets contain 80% of all possible instances. The rest 20% are reserved as the holdout test set, which will be released once enough models using ClimSim are developed by independent groups.*

4. **Is there a label or target associated with each instance?** *Yes, each instance includes both input and target (prediction) variables.*

5. **Is any information missing from individual instances?** *No.*

6. **Are there recommended data splits (e.g., training, development/validation, testing)?** *We have a hard split between the training/validation set and the test set. The first 8 simulation years-worth dataset is reserved for the training/validation set, and the last 2 simulation years-worth dataset is reserved for the test set. However, we do not have specific recommendations on the split within the training/validation set.*

7. **Are there any errors, sources of noise, or redundancies in the dataset?** *There is one redundancy. Input variable "state_pmid" is redundant since it is a linear function of "state_ps".*

8. **Is the dataset self-contained, or does it link to or otherwise rely on external resources (e.g., websites, tweets, other datasets)?** *The dataset is self-contained.*

9. **Does the dataset contain data that might be considered confidential?** *No.*

10. **Does the dataset contain data that, if viewed directly, might be offensive, insulting, threatening, or might otherwise cause anxiety?** *No.*

## 7.5 Collection Process

1. **How was the data associated with each instance acquired?** *The data associated with each instance is acquired from a series of simulations of a global climate model called E3SM-MMF. References for E3SM-MMF are provided in Section 3 of the main text.*

2. **What mechanisms or procedures were used to collect the data (e.g., hardware apparatus or sensor, manual human curation, software program, software API)?** *We used many NVIDIA A100 GPU nodes in a high-performance computing cluster called Perlmutter (operated by the U.S. Department of Energy) to run the E3SM-MMF simulations.*

3. **Who was involved in the data collection process (e.g., students, crowdworkers, contractors) and how were they compensated (e.g., how much were crowdworkers paid)?** *Regular employees (e.g., scientists and postdocs) at UC Irvine, LLNL, and SNL were involved in the data collection process. No crowdworkers were involved during the data collection process.*

4. **Does the dataset relate to people?** *No.*

5. **Did you collect the data from the individuals in questions directly, or obtain it via third parties or other sources (e.g., websites)?** *We obtained the dataset from computer simulations of Earth's climate.*

## 7.6 Uses

1. **Has the dataset been used for any tasks already?** *No, this dataset has not been used for any tasks yet.*

2. **What (other) tasks could be the dataset be used for?** *Please refer to Section 5 in the main manuscript for other applications.*

3. **Is there anything about the composition of the dataset or the way it was collected and preprocessed/cleaned/labeled that might impact future uses?** *The current composition of the datasets are self-sufficient to build a climate emulator. However, it misses some extra variables, which are not essential for such climate emulators but necessary to strictly enforce physical constraints (see Section 4.5 of the main text). We plan to include these extra variables in the next release. Any changes in the next release and update to user guidelines will be documented and shared through the dataset webpage (https://leap-stc.github.io/ClimSim).*

4. **Are there tasks for which the dataset should not be used?** *No.*

# 8 Extra Figures and Tables

## 8.1 MLP with Expanded Target Variables

| | (Variables) | MLPv1 | MLPv2 | MLPv1-ne30 | MLPv2-ne30 |
|---|---|---|---|---|---|
| **MAE** | dT/dt | 2.688 | 2.305 | 2.799 | 2.886 |
| | dq/dt | 4.503 | 4.030 | 4.231 | 4.068 |
| | dq$_l$/dt | N/A | 0.689 | N/A | 0.697 |
| | dq$_i$/dt | N/A | 0.384 | N/A | 0.330 |
| | du/dt | N/A | 1.34E-04 | N/A | 2.68E-04 |
| | dv/dt | N/A | 1.09E-04 | N/A | 2.66E-04 |
| | NETSW | 13.47 | 8.339 | 15.47 | 11.04 |
| | FLWDS | 5.118 | 4.134 | 5.318 | 4.891 |
| | PRECSC | 2.645 | 1.539 | 3.115 | 3.009 |
| | PRECC | 33.89 | 23.74 | 42.49 | 29.62 |
| | SOLS | 7.942 | 5.774 | 8.484 | 6.866 |
| | SOLL | 10.30 | 8.190 | 10.582 | 8.993 |
| | SOLSD | 4.587 | 3.230 | 5.056 | 4.360 |
| | SOLLD | 4.834 | 3.977 | 4.963 | 4.553 |
| **R2** | dT/dt | 0.590 | 0.663 | 0.626 | 0.536 |
| | dq/dt | - | - | - | - |
| | dq$_l$/dt | N/A | - | N/A | - |
| | dq$_i$/dt | N/A | - | N/A | - |
| | du/dt | N/A | - | N/A | - |
| | dv/dt | N/A | - | N/A | - |
| | NETSW | 0.982 | 0.993 | 0.977 | 0.988 |
| | FLWDS | 0.927 | 0.945 | 0.914 | 0.924 |
| | PRECSC | - | - | -0.117 | -0.117 |
| | PRECC | -1.494 | 0.833 | -0.115 | -0.115 |
| | SOLS | 0.962 | 0.978 | 0.963 | 0.976 |
| | SOLL | 0.948 | 0.964 | 0.953 | 0.965 |
| | SOLSD | 0.955 | 0.976 | 0.950 | 0.965 |
| | SOLLD | 0.866 | 0.905 | 0.874 | 0.899 |
| **RMSE** | dT/dt | 4.437 | 3.756 | 5.199 | 4.958 |
| | dq/dt | 7.337 | 6.521 | 7.550 | 7.135 |
| | dq$_l$/dt | | 1.192 | | 1.489 |
| | dq$_i$/dt | | 0.812 | | 0.940 |
| | du/dt | | 2.80E-04 | | 6.45E-04 |
| | dv/dt | | 2.25E-04 | | 6.72E-04 |
| | NETSW | 26.95 | 17.24 | 30.48 | 21.18 |
| | FLWDS | 6.803 | 5.532 | 7.136 | 6.540 |
| | PRECSC | 4.656 | 2.955 | 7.791 | 7.509 |
| | PRECC | 73.16 | 53.47 | 119.8 | 83.22 |
| | SOLS | 17.39 | 12.84 | 18.51 | 14.74 |
| | SOLL | 21.96 | 17.89 | 22.71 | 19.27 |
| | SOLSD | 9.474 | 6.837 | 10.42 | 8.724 |
| | SOLLD | 10.14 | 8.486 | 10.62 | 9.526 |

Table 4: Similar to Table 2 in the main text but for comparing MAR, R2, and RMSE of different MLP models: MLP v1 (subset emulation) and the MLP v2 (full vector emulation) built with the low-resolution (ne4) and the high-resolution datasets (ne30). dq$_l$/dt, dq$_i$/dt, du/dt, and dv/dt correspond to the tendencies of state_q0002, state_q0003, state_u, and state_v, respectively, in Table SI1.

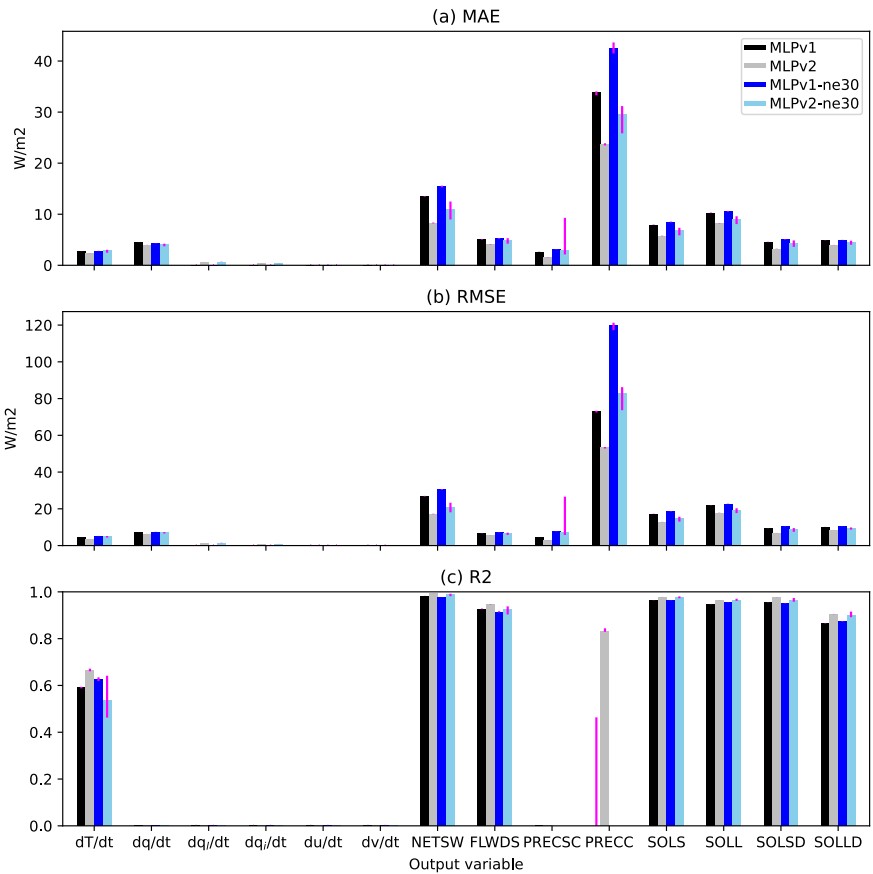

Figure 7: Equivalent to Figure S3, but for comparing the MLPv1 (subset emulation) and the MLPv2 (full vector emulation). In addition, MLP models trained with the high-resolution dataset (ne30) are shown here: MLPv1-ne30 and MLPv2-ne30. Bars show the median of the performance of top-20 models selected from the hyperparamter search (>8,000 trials), and magenta error bars show the range of the top-20 model performance.

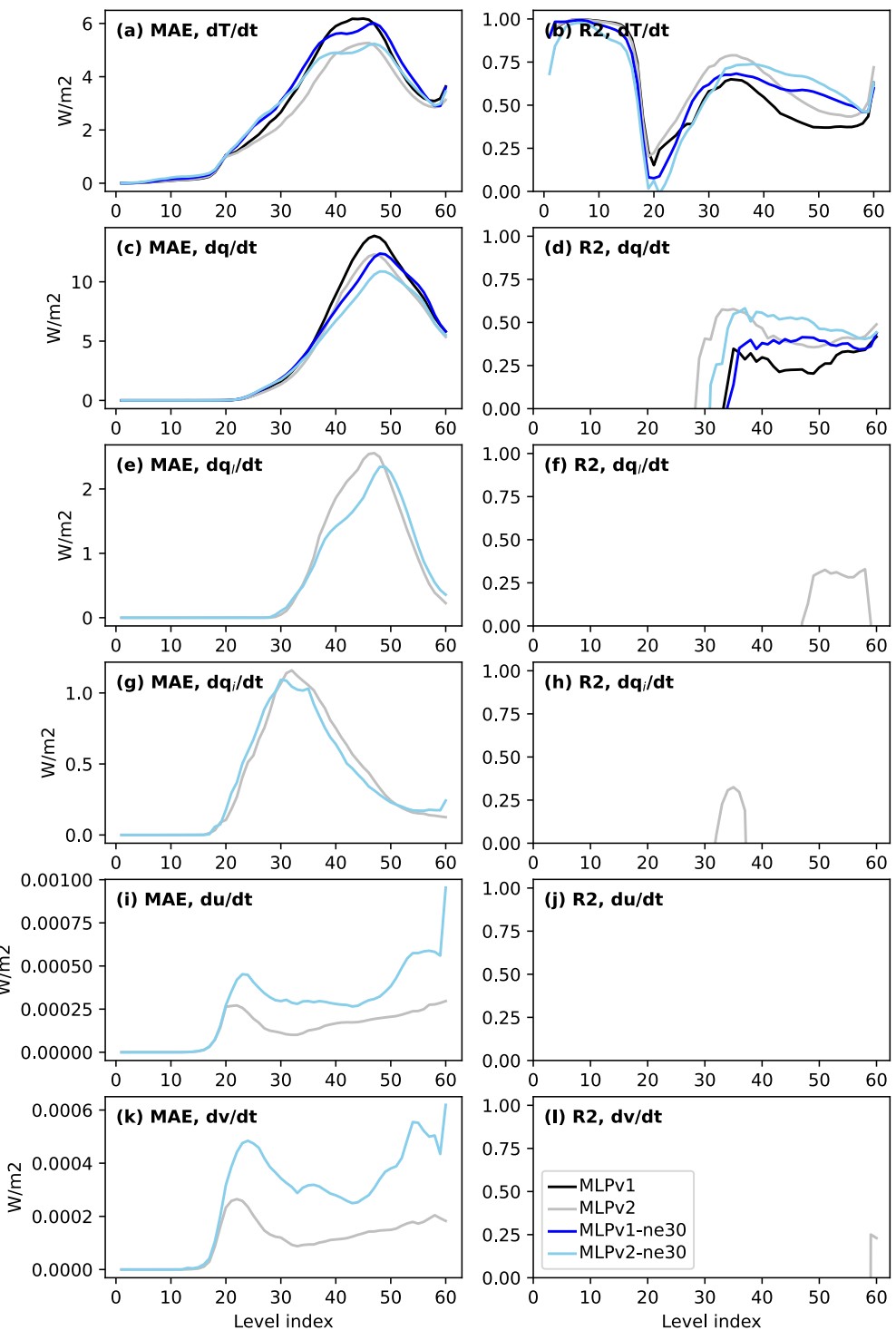

Figure 8: Equivalent to Figure 2, but for comparing the MLP v1 (subset emulation) and the MLP v2 (full vector emulation). In addition, MLP models trained with the high-resolution dataset (ne30) are shown here: MLPv1-ne30 and MLPv2-ne30. Out of the top model pools, MLP models shown in this figure are randomly chosen for visualizatoin.

## 8.2 Scatter Plots

### Net surface shortwave flux, NETSW

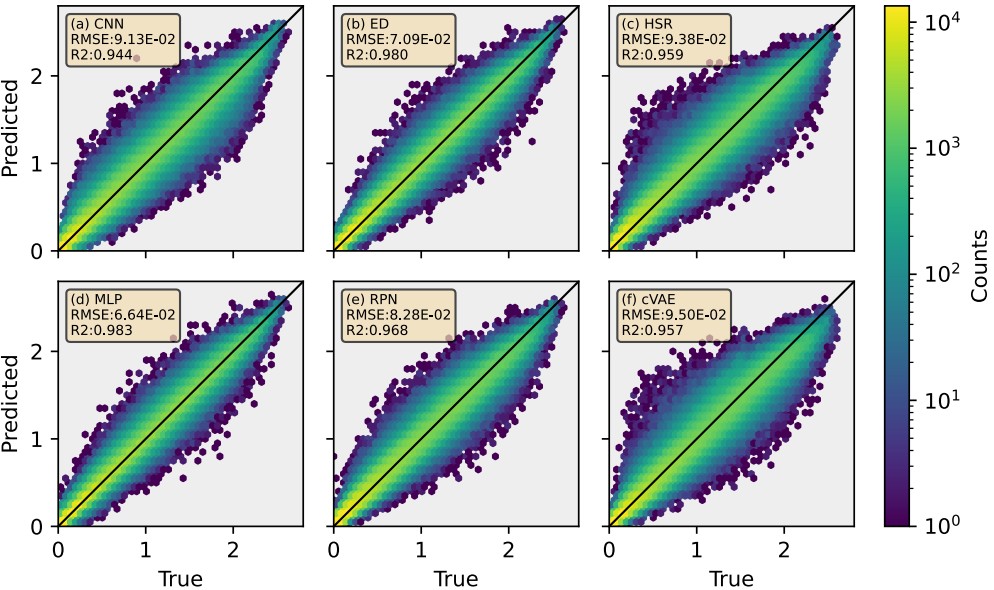

### Downward surface longwave flux, FLWDS

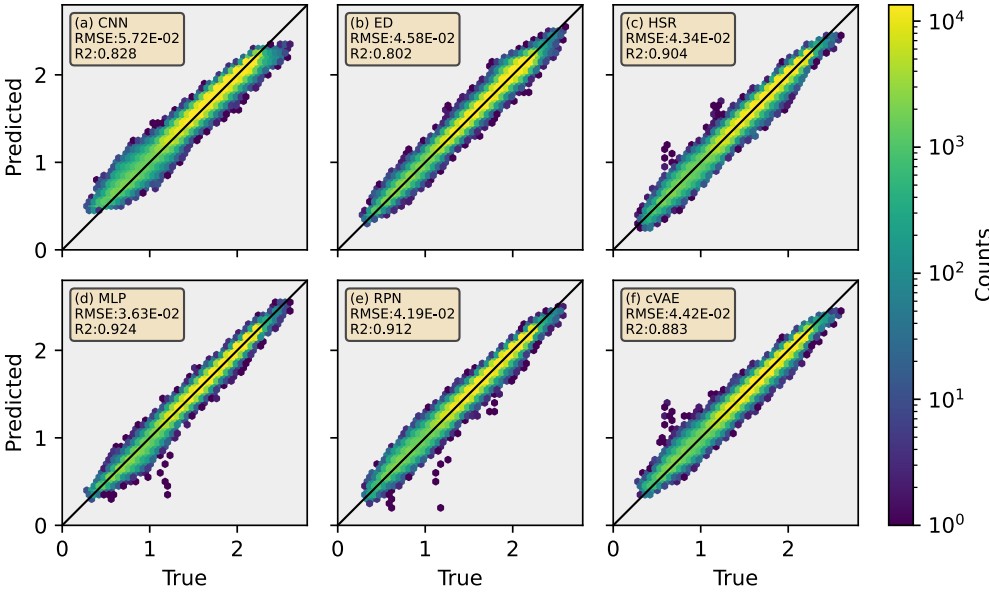

Figure 9: Hexagonally-binned representation of 2D target variables comparing the climate model simulation ("true"; x-axis) with the ML model prediction ("predicted"; y-axis). The color of each hexagonal bin corresponds to the number of data points enclosed.

## Snow rate, PRECSC

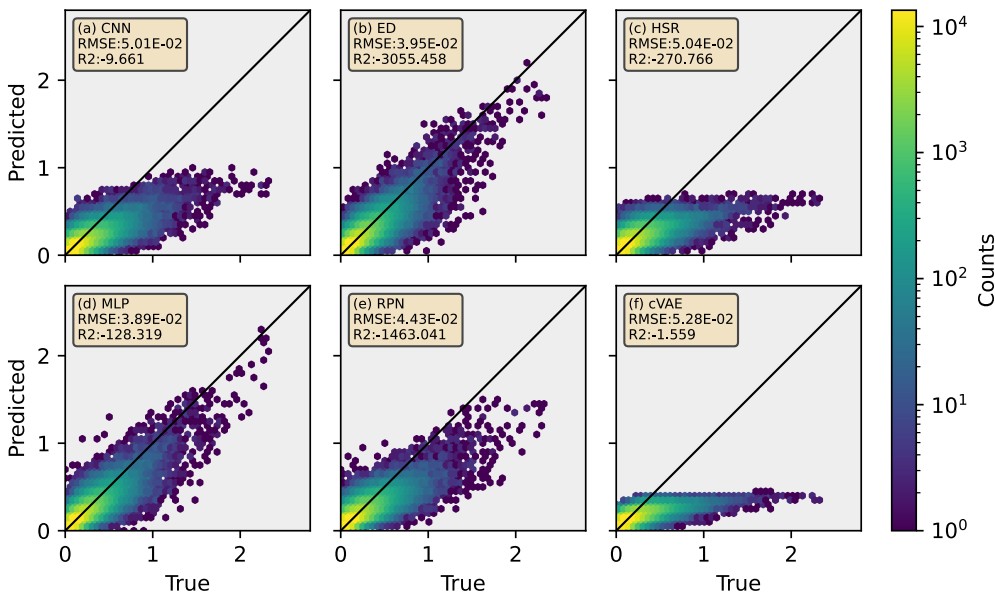

## Rain rate, PRECC

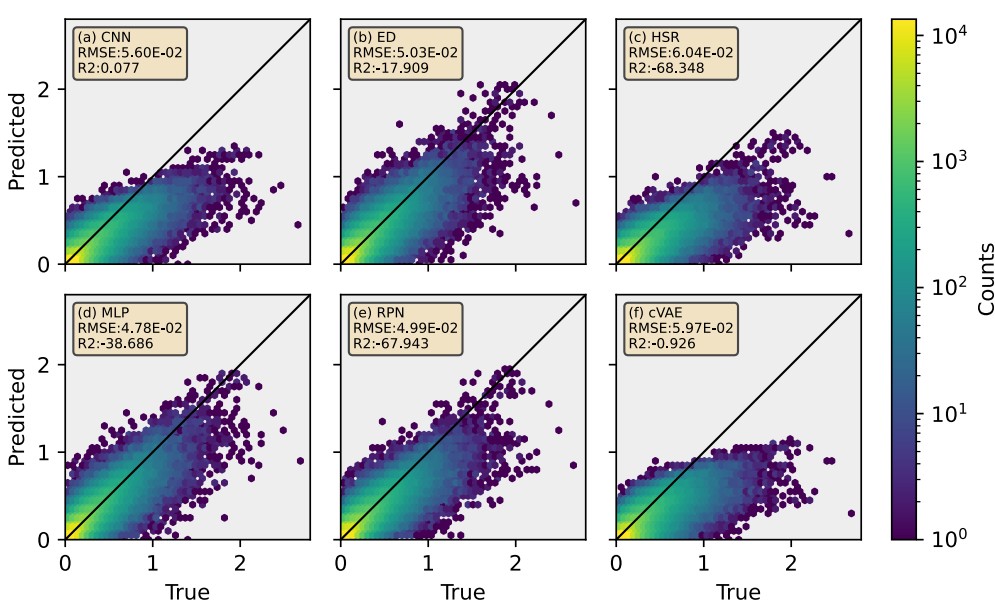

Figure 10: Hexagonally-binned representation of 2D target variables comparing the climate model simulation ("true"; x-axis) with the ML model prediction ("predicted"; y-axis). The color of each hexagonal bin corresponds to the number of data points enclosed.

## Visible direct solar flux, SOLS

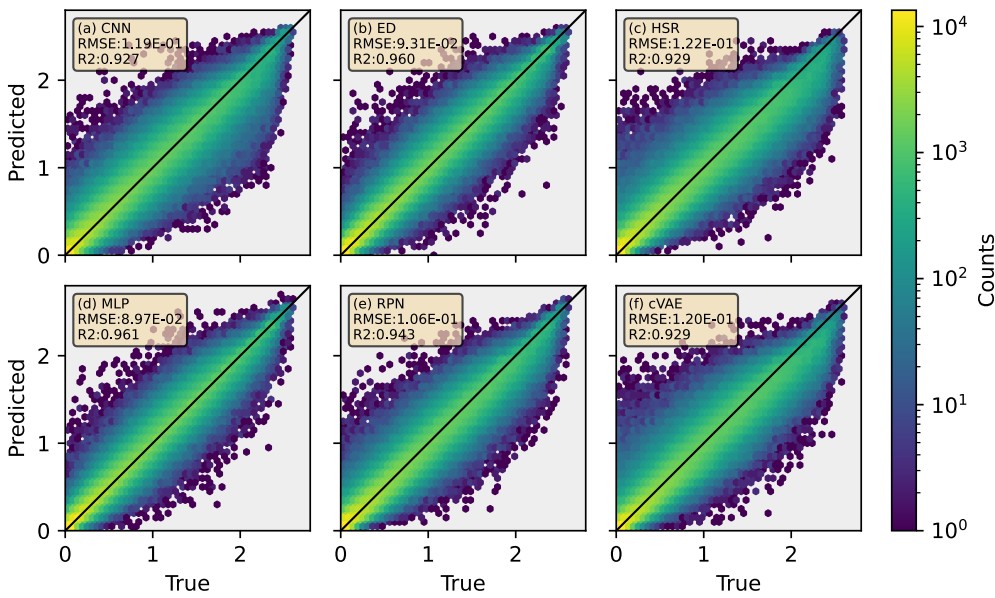

## Near-IR direct solar flux, SOLL

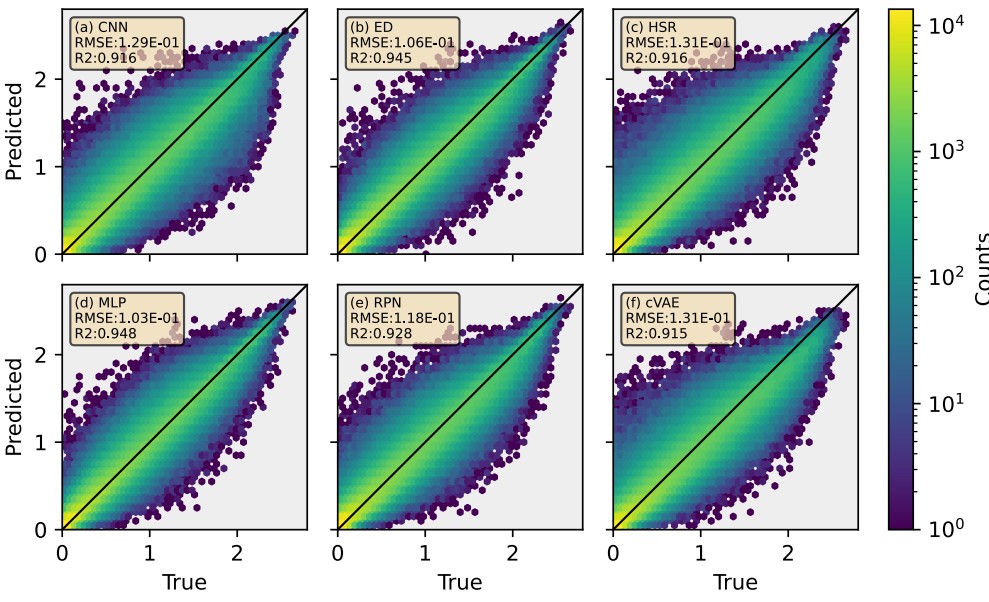

Figure 11: Hexagonally-binned representation of 2D target variables comparing the climate model simulation ("true"; x-axis) with the ML model prediction ("predicted"; y-axis). The color of each hexagonal bin corresponds to the number of data points enclosed.

## Visible diffused solar flux, SOLSD

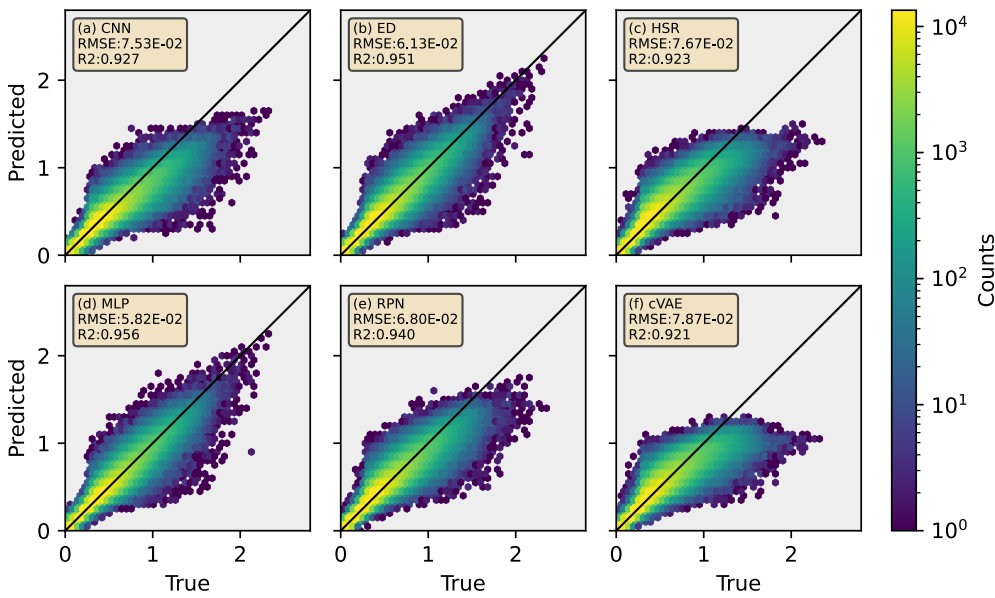

## Near-IR diffused solar flux, SOLLD

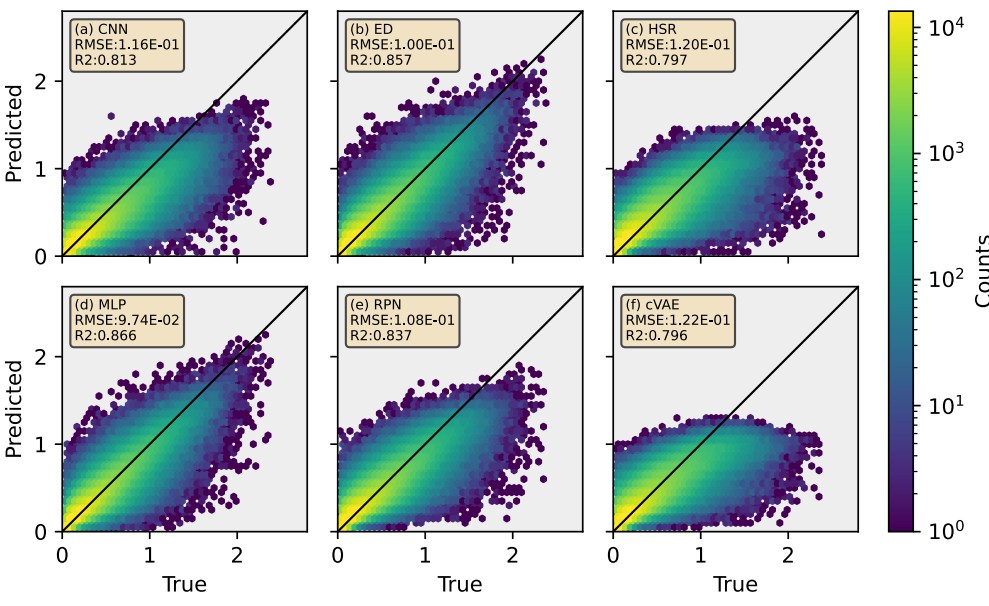

Figure 12: Hexagonally-binned representation of 2D target variables comparing the climate model simulation ("true"; x-axis) with the ML model prediction ("predicted"; y-axis). The color of each hexagonal bin corresponds to the number of data points enclosed.

Heating tendency, ∂T/∂t (level=2)

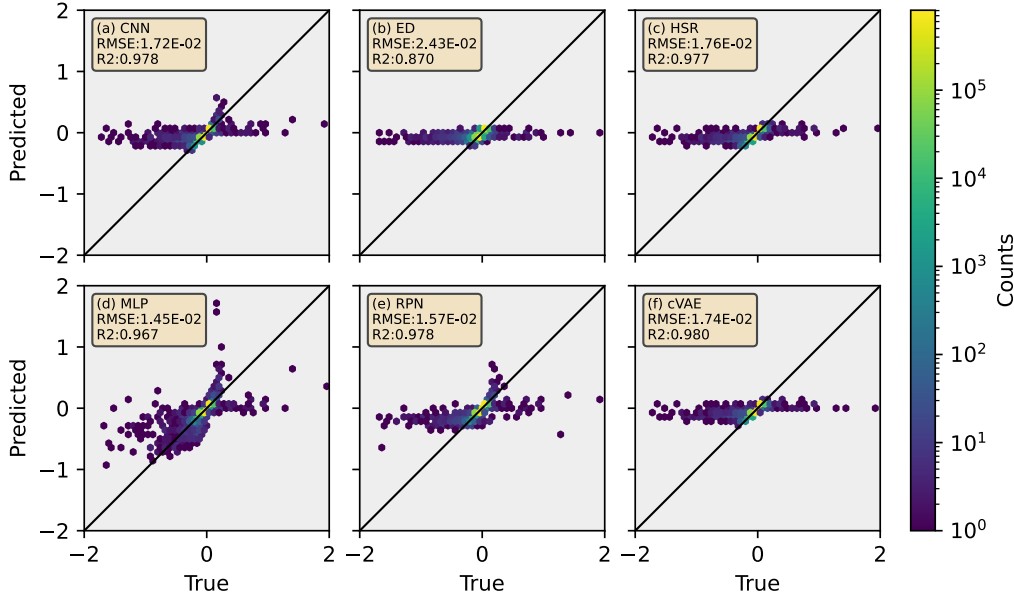

Heating tendency, ∂T/∂t (level=17)

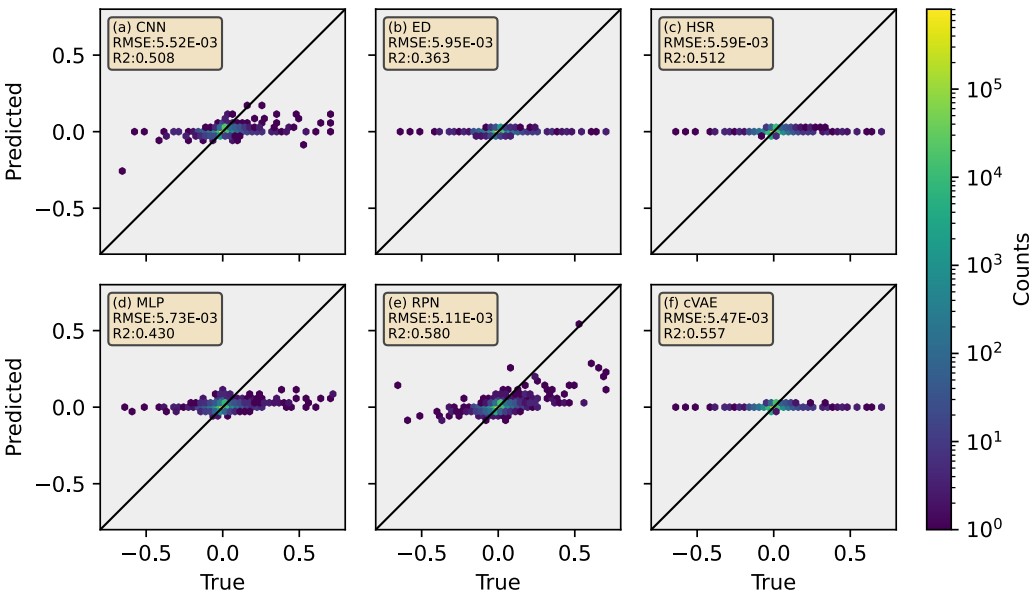

Figure 13: Hexagonally-binned representation of 3D (vertically-resolved) target variables comparing the climate model simulation ("true"; x-axis) with the ML model prediction ("predicted"; y-axis) at four different vertical levels. The color of each hexagonal bin corresponds to the number of data points enclosed.

Heating tendency, ∂T/∂t (level=32)

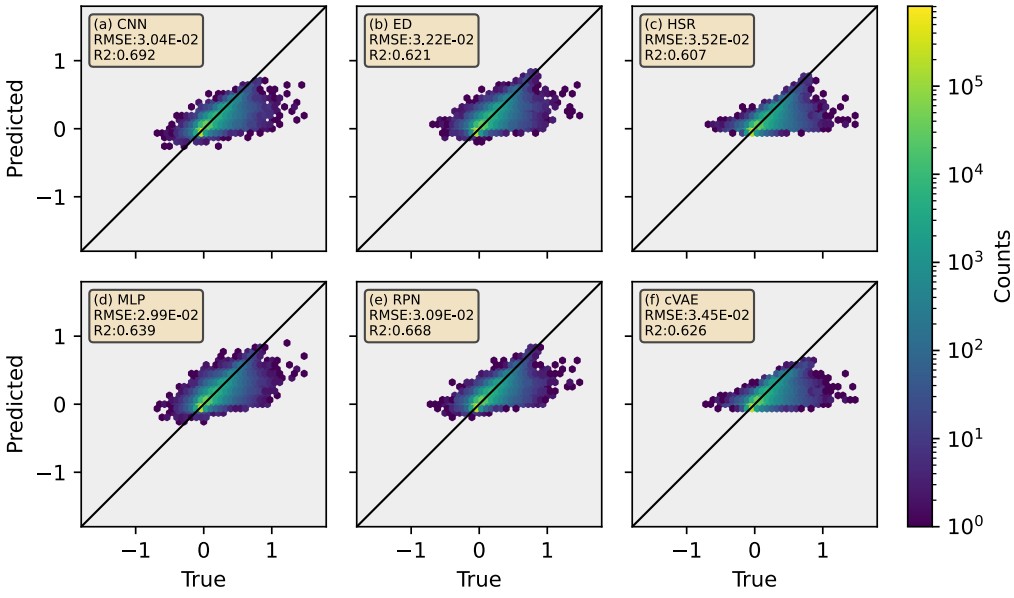

Heating tendency, ∂T/∂t (level=57)

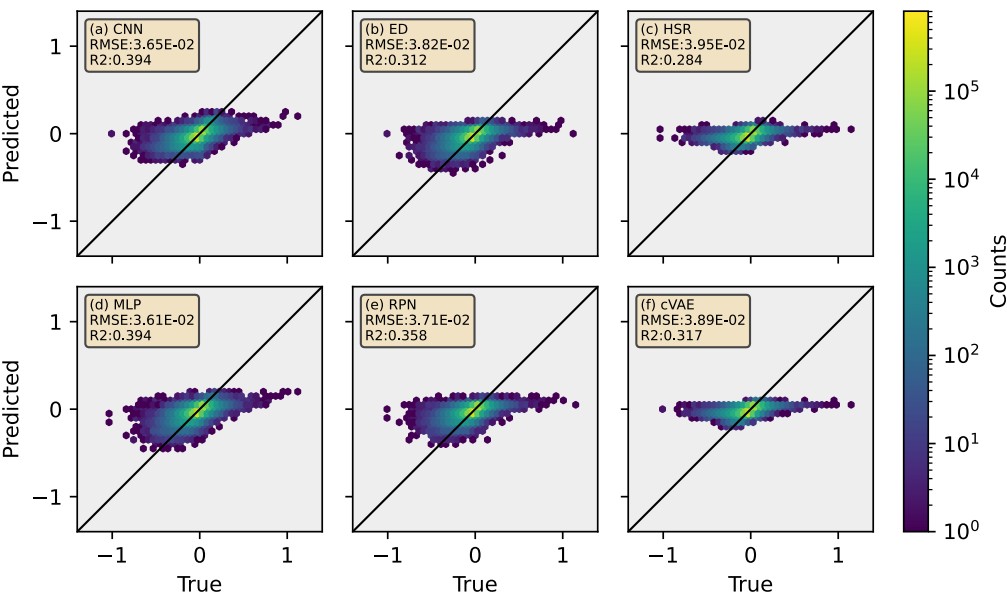

Figure 14: Hexagonally-binned representation of 3D (vertically-resolved) target variables comparing the climate model simulation ("true"; x-axis) with the ML model prediction ("predicted"; y-axis) at four different vertical levels. The color of each hexagonal bin corresponds to the number of data points enclosed.

## Moistening tendency, ∂q/∂t (level=2)

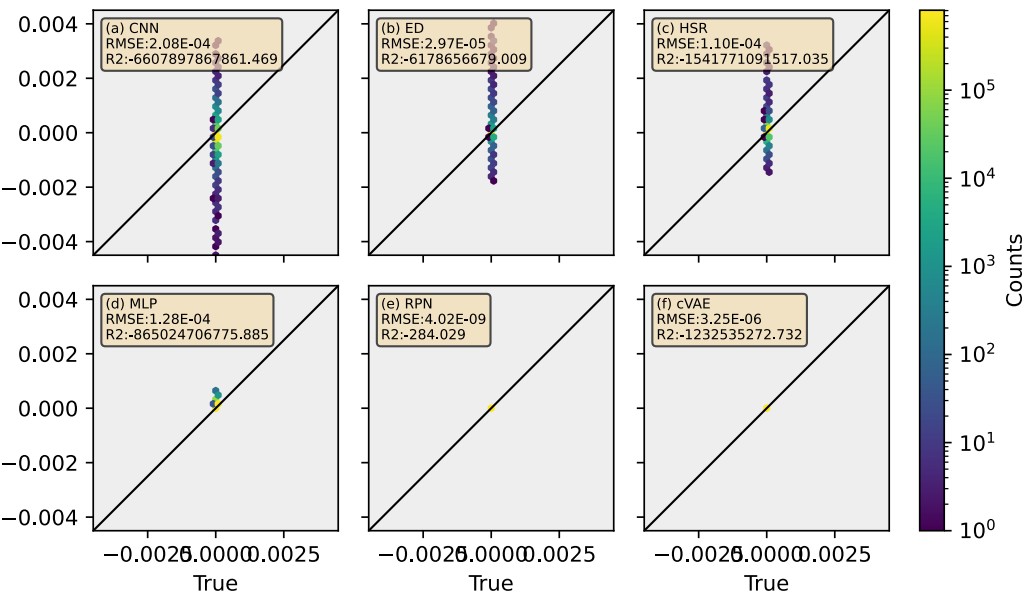

## Moistening tendency, ∂q/∂t (level=17)

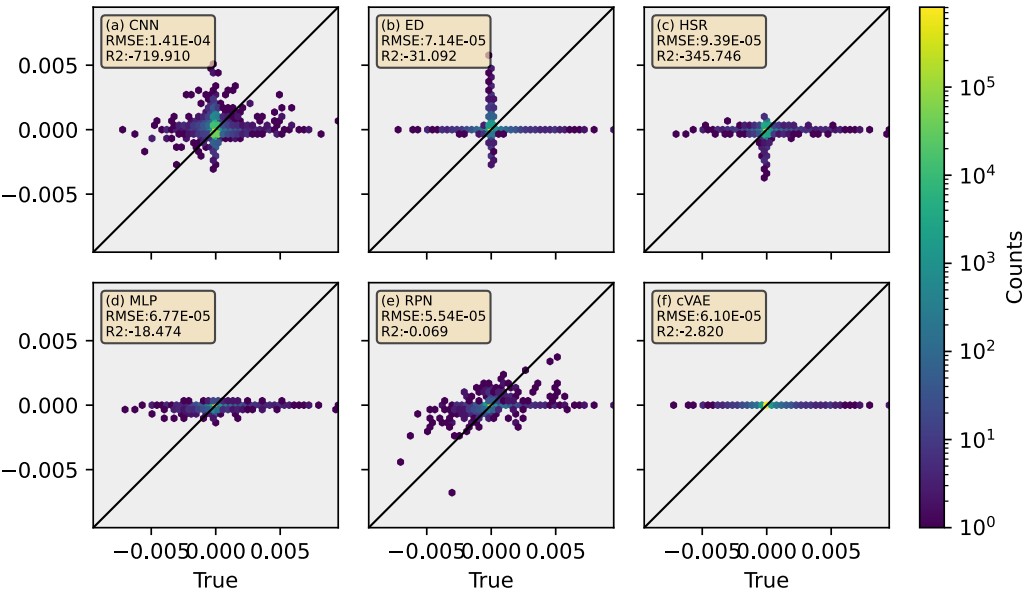

Figure 15: Hexagonally-binned representation of 3D (vertically-resolved) target variables comparing the climate model simulation ("true"; x-axis) with the ML model prediction ("predicted"; y-axis) at four different vertical levels. The color of each hexagonal bin corresponds to the number of data points enclosed.

## Moistening tendency, ∂q/∂t (level=32)

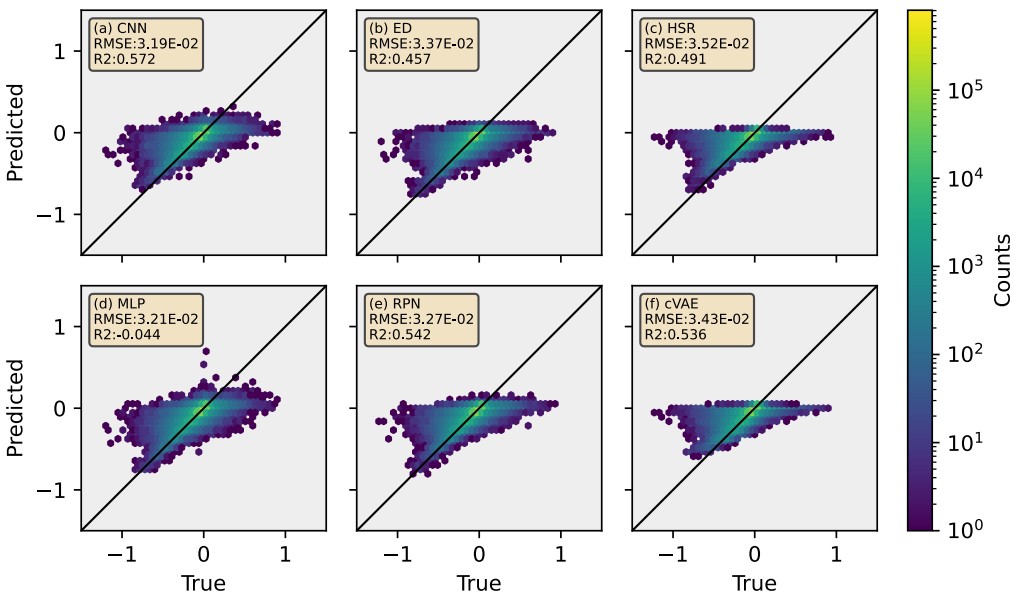

## Moistening tendency, ∂q/∂t (level=57)

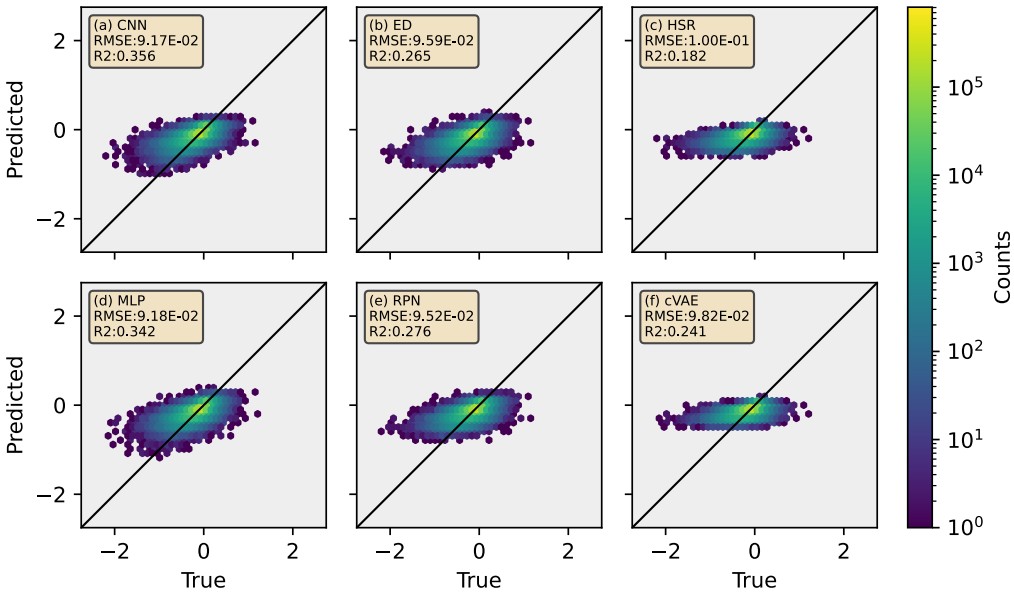

Figure 16: Hexagonally-binned representation of 3D (vertically-resolved) target variables comparing the climate model simulation ("true"; x-axis) with the ML model prediction ("predicted"; y-axis) at four different vertical levels. The color of each hexagonal bin corresponds to the number of data points enclosed.

## 8.3 Global Maps of $R^2$

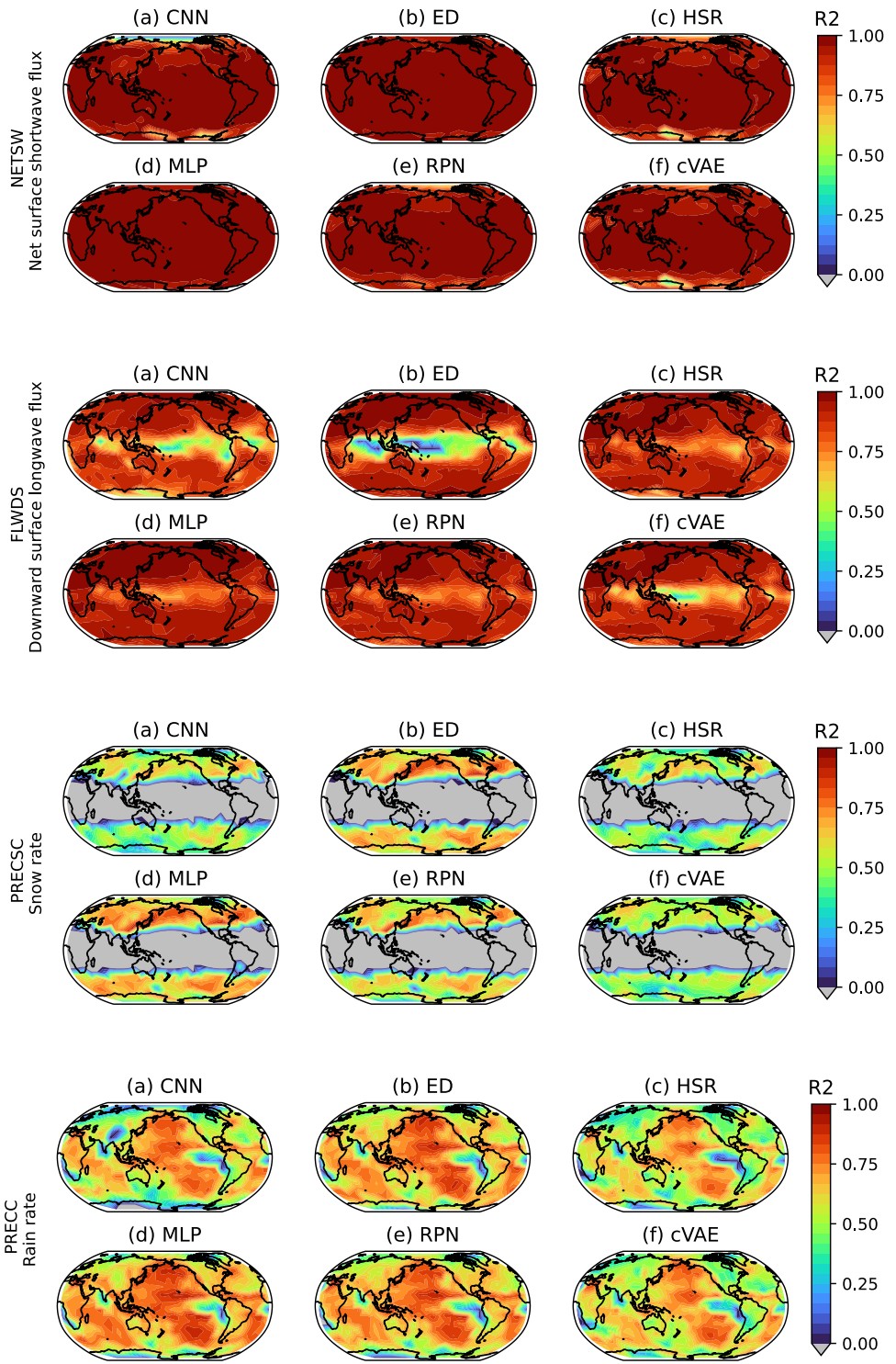

Figure 17: Global maps of $R^2$ of baseline models (built on the low-res, real-geography dataset). Grey shading shows locations with negative $R^2$ values.

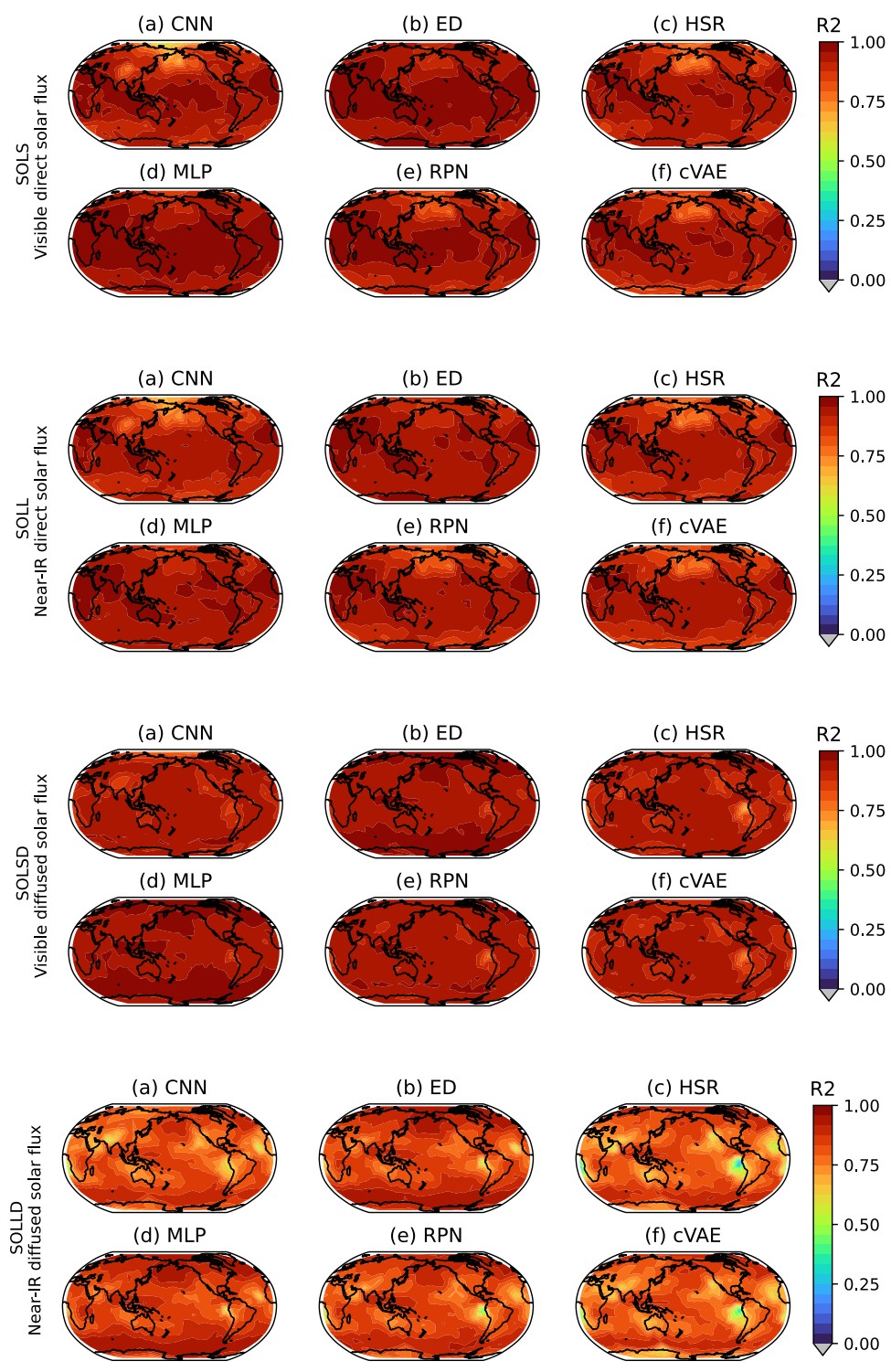

Figure 18: Global maps of $R^2$ of baseline models (built on the low-res, real-geography dataset). Grey shading shows locations with negative $R^2$ values.

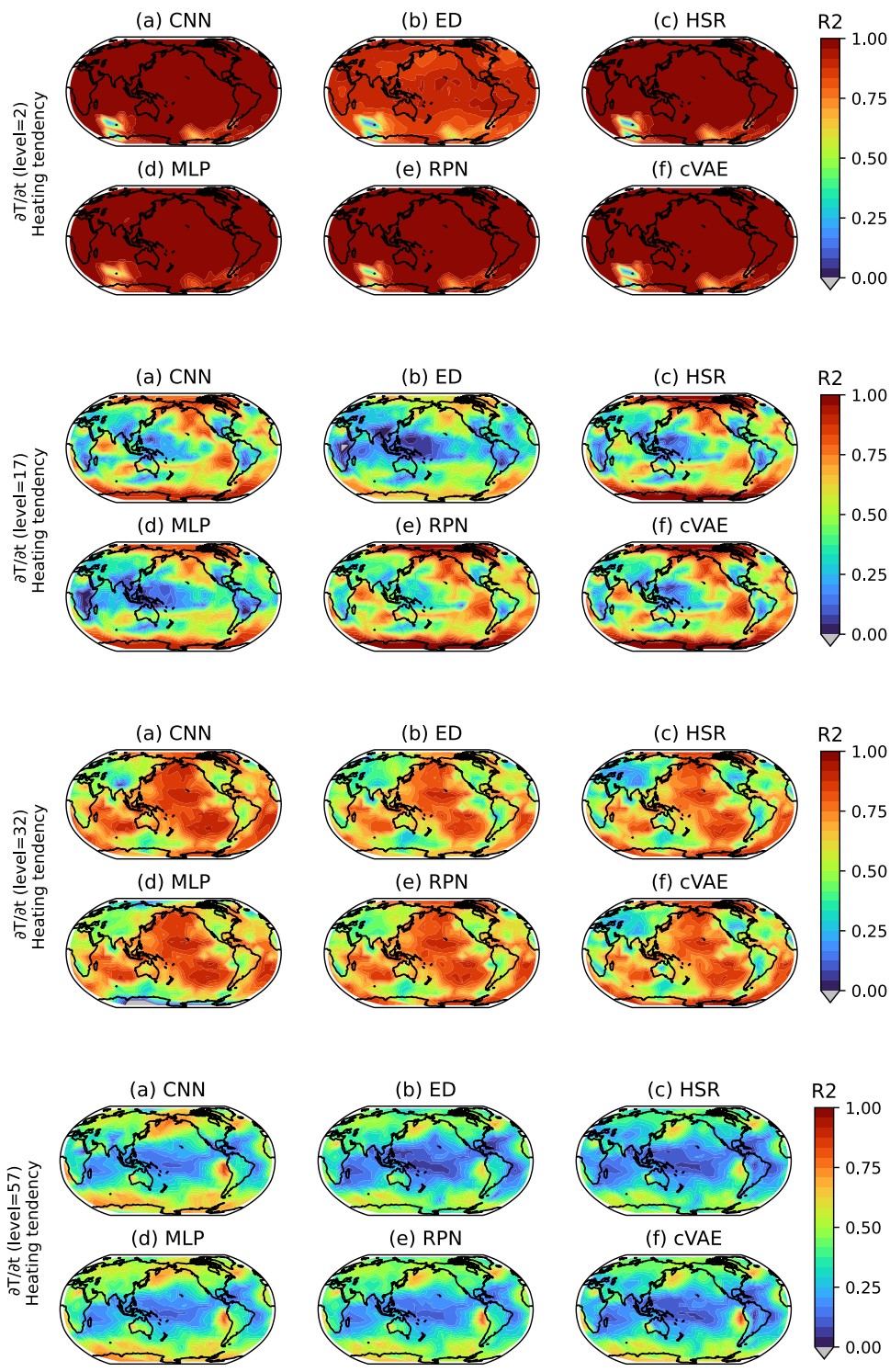

Figure 19: Global maps of $R^2$ of baseline models (built on the low-res, real-geography dataset). Grey shading shows locations with negative $R^2$ values.

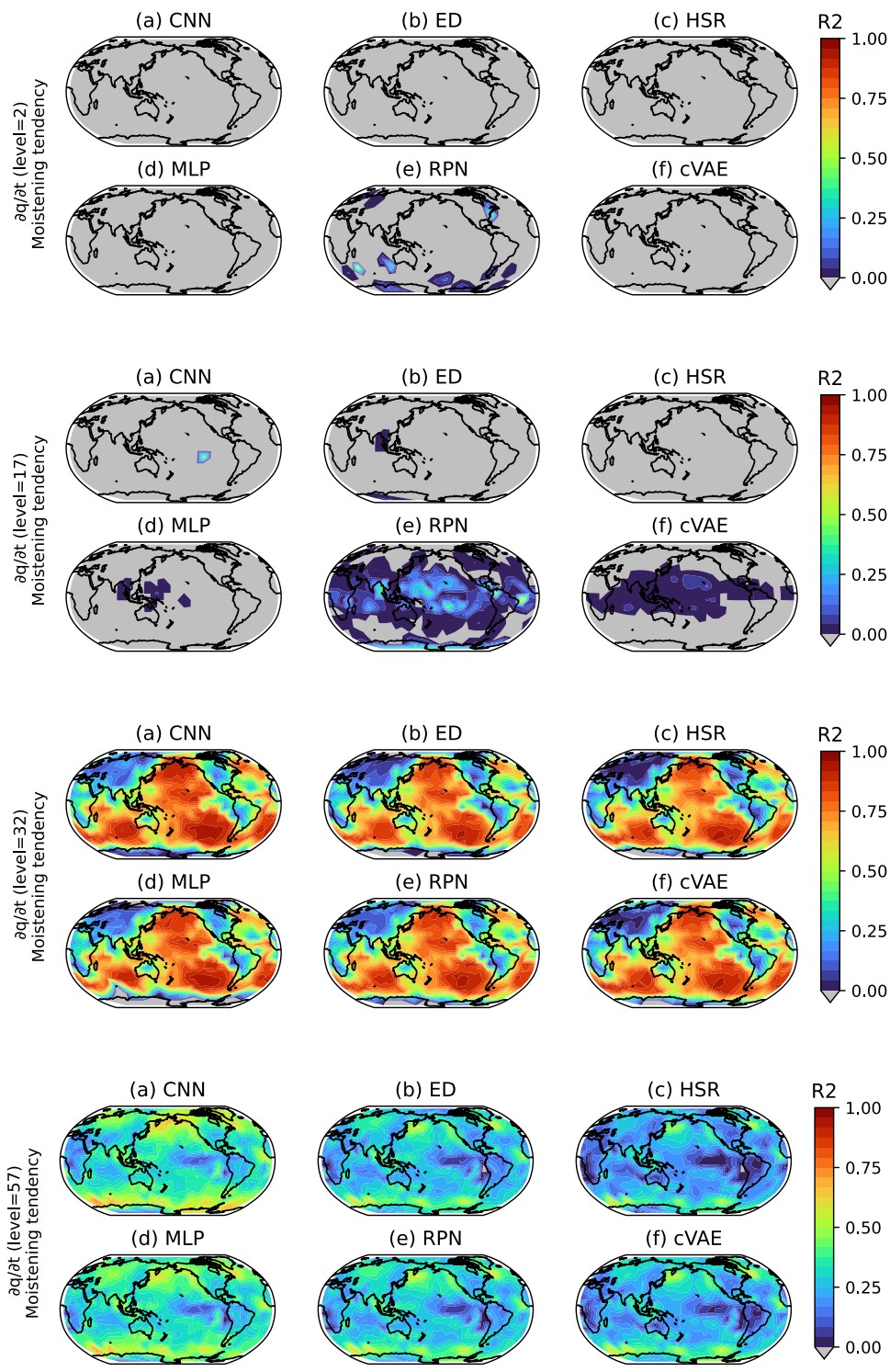

Figure 20: Global maps of $R^2$ of baseline models (built on the low-res, real-geography dataset). Grey shading shows locations with negative $R^2$ values.