# OpenReview forum: "ClimSim: A large multi-scale dataset for hybrid physics-ML climate emulation"
_NeurIPS.cc/2023/Track/Datasets_and_Benchmarks — NeurIPS 2023 Datasets and Benchmarks Oral_

### Official Review · Reviewer_PWUd · 2023-07-08
**High Resolution Climate Data for Hybrid ML Simulations**

**Rating:** 9
**Confidence:** 4
**Correctness:** The paper appears to be correct.

**Strengths:**

There are clear advantages to using a hybrid climate simulation, if the ML models can provide sufficient accuracy for the high-resolution data. This work will facilitate the development of such research. The dataset is comprehensive and the multiple levels of resolution allow researchers to develop ML models on a reasonably-sized dataset and then test their model with the  high-resolution dataset.

**Additional Feedback:**

On line 167, change "24 2D variables and 10, 3D" to either remove the comma after 10 or insert one after 24. (Removing the comma seems correct.)
On line 213, suggest changing "1-D" to "1D" for consistency with the rest of the paper.

**Clarity:**

The paper is well-organized and well-written. The results are explained clearly, although some further discussion of Table 2 should be considered in the final version.

**Documentation:**

The online dataset is available and appears to be easy to access with sufficient explanation to understand how to use it. The code repository is not yet populated.

**Ethics:**

No concerns.

**Limitations:**

The limitations are sufficiently described in the paper. The limitations are ones that should be expected at this level of development of the dataset and do not detract from the contribution.

**Opportunities For Improvement:**

No suggestions.

**Relation To Prior Work:**

The relationship between this work and previous work is explained well. The dataset in the paper opens the door for novel research on hybrid climate models.

**Summary And Contributions:**

A climate dataset with varying levels of high-resolution data is produced. One objective is to support the further development of Hybrid ML climate simulations, where ML can be used for the high-resolution data and these results can be fed into a traditional climate simulator. The other objective is to allow training on relatively low-resolution data and then promising models can be tested on higher-resolution data.

---

> ### Author Response · Authors · 2023-08-22
> **Authors' response to Reviewer PWUd**
>
> We appreciate the reviewer's time and thoughtful feedback on our manuscript. It's great to hear that our ClimSim dataset came across as comprehensive and promising for advancing the development of hybrid ML climate simulators. We have taken your feedback into account and made revisions accordingly. If you have any more comments or if something in our response isn't clear, feel free to let us know. We are committed to addressing any potential points of ambiguity and providing additional information to the best of our abilities.
>
> Point-by-point response:
>
> > Clarity: The paper is well-organized and well-written. The results are explained clearly, although some further discussion of Table 2 should be considered in the final version
>
> We have added some further discussion on the baseline model results in Section 4.4 “Baseline Model Results” [Lines 267-278].
>
> [Lines 267-278] “ For the global mean MAE we see the largest averaged errors for PRECC and NETSW (mean MAE > 15 W/m², Figure 2 and Table 2), where MLP clearly has the best the best skill compared to all other benchmark models. For the other variables, the global mean MAE is considerably smaller and the skill of the benchmarks model appears to be more similar in absolute numbers. While for the global mean R$^\text{2}$ we find the lowest measurable performance for dT/dt and PRECC (mean R$^\text{2} <$ 0.7) and in these cases, CNN gives the most skillful predictions. The other variables have larger R$^\text{2}$ of order 0.8 or higher, which suggests that these quantities are easier to deep-learn (Table 2). For dq/dt and PRECSC global mean R$^\text{2}$ is not an ideal evaluation metric due to negligible variability in dq/dt in the upper atmosphere and for PRECSC in the tropics in the dataset (Table 2}).
>
> Additional tables and figures that reveal the geographic and vertical structure of these errors, fit quality, and analysis of stochastic metrics, are included in SI (Sections 4.3, 8.1, and 8.2 in SI).”
>
> > Documentation: The online dataset is available and appears to be easy to access with sufficient explanation to understand how to use it. The code repository is not yet populated.
>
> We apologize for the inconvenience that the reviewer did not find the way to connect to the repository during the review period. Our code repository has been online (https://github.com/leap-stc/ClimSim/) since the initial submission of our manuscript; however, the code repository weblink was not shown clearly on the landing page due to the small button size (https://leap-stc.github.io/ClimSim/intro.html). We have updated the landing page and the github documentation to fix this issue. For details, please refer to our initial comments on the repo reorganization and quickstart.
>
> >On line 167, change "24 2D variables and 10, 3D" to either remove the comma after 10 or insert one after 24 (removing the comma seems correct).
>
> We have fixed the typo by removing the comma, as suggested.
>
> >On line 213, suggest changing "1-D" to "1D" for consistency with the rest of the paper.
>
> We have fixed the typo,  as suggested.

---

> > ### Comment · Reviewer_PWUd · 2023-08-28
> > **Reply**
> >
> > Thank you for addressing my comments. I'm satisfied that these changes address my points.

---

> > > ### Author Response · Authors · 2023-08-31
> > >
> > > We would like to reiterate our gratitude. Thank you very much again for your invaluable feedback. We anticipate your continued suggestions and feedback whenever they arise. Your input is truly helpful for future releases of ClimSIm.

---

### Official Review · Reviewer_ujho · 2023-07-21
**A Massive Simulation Dataset for Investigating Parameterizations with ML methods**

**Rating:** 9
**Confidence:** 4
**Correctness:** I have some comments above in the opp…

**Strengths:**

* The scale of the dataset is massive. They address the biggest problem with training extensive ML methods is the fact that they are data hungry. This is an excellent start to training large-scale models.
* They target a wide range of relevant variables, which could be helpful for different people involved in climate science applications. There could be an overlap between the variables used and the case studies one wishes to address.
* There are detailed preprocessing steps (which people may want to modify).
* The data is hosted on the HuggingFace platform, which is excellent for visibility and ease of access. This is the first climate-based data.
From an ML perspective, it answers some key questions:  1) The problem we are trying to solve, 2) The subsequent toy problem/reduced problem we wish to solve, 3) The training data (and structure) provided, 4) Results comparison.
* There are some solid benchmark models which span a wide range of assumptions were included. They were evaluated fairly and consistently.
* The authors include an excellent detailed overview of improving the proposed models based on physics-informed considerations.
* Lastly, there was a good overview of some of the limitations and potential applications for which this dataset can be used.

**Additional Feedback:**

N/A

**Clarity:**

The paper has a good balance of domain expertise and machine learning.
It is also easy enough for ML researchers to understand the task.

**Documentation:**

The authors have a complete overview of the data generation and collection.
From an ML perspective, there are demo notebooks for users to start.
They also share the training procedure and resulting weights from their baseline models.


A few suggestions for the code base:

* The benchmark system should be more ML-like, where things are organized in a more ML fashion. While browsing, I found no clear introduction for new users as to what to look at first. For example, a better template could be - https://github.com/ashleve/lightning-hydra-template. The dataset is massive, and the challenge is great, so a more ML-like organization is necessary.
* I would like to see a more complete introduction on the website posted. I assume there will be more work done (which is fine). However, I want to reiterate that although the paper is quite dense with information that may be helpful conceptually, I think the code base should also exhibit the same completeness and cater to new users.

**Ethics:**

To the best of my ability, I see that the authors addressed some of the ethics.

**Limitations:**

The authors addressed some of the limitations of their work. However, I'm very surprised there was no mention of the societal impact due to the processing power required. There could even be positive ones,  for example:
* a positive could be the fact that the simulation was done on a very streamlined/efficient system
* a large consortium of interested parties agreed on a dataset to ensure it isn't wasted.

A brief statement might be worth adding.

**Opportunities For Improvement:**

I want to preface this by saying that the paper is excellent and completes all of the requirements to be accepted. However, I have a few things to suggest to increase the impact.

* I understand that the target audience is those interested in specific parameterizations. However, there are some overlapping uses that this dataset can be used for, e.g., surrogates, hybrid model training/exploration, and process understanding. Many authors are on the paper from different fields, so other use cases can be explored using this dataset alone. This is mentioned in the limitations and other applications, but there should be more positive statements (appendix worthy) of the potential applications.
* Often there is a gap between research and the operational world; thus, many, many ML systems don’t get used in real-world operational systems. A brief statement about who this work is for, I.e., operationally, research only, etc, might be inspirational. While keeping experiments within a self-contained system is excellent, it’s also essential to know the context system and where it will finally be used as it might spark some thought about the overall experimental design (which may sometimes warrant a reformulation of the problem or pivot).
* Concerning the previous point, some characteristics of the final goal could be highlighted. Are there any real-world operational constraints to consider? For example, memory and computational budget, fast inference, stability? In general, I think one should we include these into the assumptions and metrics catalogue because often scale and stability/consistency are some determining factors for whether a model gets adopted or not.
* The metrics are complete. However, are there any spectral metrics to consider (as is often done with fields)? In addition, I didn’t see any breakdown of the metrics per region, time period, etc. Optimizing the best metric based on the mean is a “simple task,” but there could also be other spatiotemporal/spectral diagnostic quantities that showcase whether the model did a good job or not.

**Relation To Prior Work:**

This work covers all of the literature reviews about this work.

**Summary And Contributions:**

The authors present ClimSim, a dataset designed for researchers to train hybrid ML models on high-fidelity climate simulations.
In particular, some applications can include training emulators for cloud parameterizations and extreme rainfall physics.
A considerable effort was made to generate an extensive dataset with over 5.7 billion pairs of physical states of different variables at a global coverage.
They also include some benchmark baseline ML regression methods (including some stochastic methods).
Some standard metrics are provided to assess the performance.
The dataset is hosted on and easily accessible from the HuggingFace platform.
The code is open-source on a GitHub repo where they feature data loaders, pre-trained models, and getting-started tutorials.

---

> ### Author Response · Authors · 2023-08-22
> **Authors' response to Reviewer ujho (1/2)**
>
> We want to express our genuine appreciation to the reviewers for taking the time to provide such thorough feedback. We're really excited about the review's positive note that "the paper is excellent and meets all the requirements for acceptance." We have been working hard to tackle the reviewer's suggestions and make our manuscript even better in this revision. If any aspect of our response requires clarification (or additional questions or comments arise), please let us know. We'd be very grateful for any further feedback, and we will promptly address any such inputs to the best of our ability.
>
> Point-by-point response:
>
> >I understand that the target audience is those interested in specific parameterizations. However, there are some overlapping uses that this dataset can be used for, e.g., surrogates, hybrid model training/exploration, and process understanding. Many authors are on the paper from different fields, so other use cases can be explored using this dataset alone. This is mentioned in the limitations and other applications, but there should be more positive statements (appendix worthy) of the potential applications.
>
> We agree and have added a new section to the SI (Section 2.4 “Dataset Applications”; Lines SI 178-187), which discusses the potential use of ClimSim dataset beyond our target parameterization problem.
>
> [Lines SI 178-187] “2.4 Data Applications
> Our data can benefit a broader audience beyond climate modelers wishing to explore ML for sub-grid  parameterization. For climate studies, while high-frequency timestep-level outputs from simulations are rarely archived, they offer insights into convective extremes and diurnal variability. Such data opens the path to explore multi-scale interactions between rapid dynamics and broader weather and climate fluctuations. This includes a detailed examination of variables needed to constrain vertically resolved energy and water budgets and understand their variability. For the machine learning community, this dataset addresses the scarcity of large-scale regression benchmarks, common in the sciences. Such benchmarks are less common compared to prevalent industrial datasets that emphasize classification, computer vision, and NLP tasks.”
>
>
> >Often there is a gap between research and the operational world; thus, many, many ML systems don’t get used in real-world operational systems. A brief statement about who this work is for, I.e., operationally, research only, etc., might be inspirational. While keeping experiments within a self-contained system is excellent, it’s also essential to know the context system and where it will finally be used as it might spark some thought about the overall experimental design (which may sometimes warrant a reformulation of the problem or pivot).
>
> We have added a new section in SI (Section 2.5 “Target Audiences”; Lines SI 188-203), which is dedicated to discussing the potential pool of Climsim users.
>
> [Lines SI 188-203] “2.5 Target Audiences
> In essence, this benchmark aims to democratize and expand access to advanced climate modeling. High-potential architectures will undergo testing in the superparameterized version of the DOE's primary climate model, E3SM. Successful integration would substantially reduce computational costs for the DOE when contemplating the deployment of MMF technology in climate prediction. E3SM's external user community, typically deterred by the extensive computational demands of superparameterized simulators, also stands to benefit. Currently, only a minority with substantial computing resources can engage with such models. A successful recipe for ClimSim could thus democratize the use of explicit convection for a broader user base. If performant architectures also prove effective in the NCAR Community Earth System Model (CESM) - the world’s most widely used open source climate simulator - the user base could expand significantly. Given its software similarities to E3SM, it is logical to expect that ClimSim's learnt parameterizations will be readily adaptable to CESM. Moreover, we anticipate that a successful hybrid machine learning climate simulator will bring benefits to a diverse range of industry sectors, including those vulnerable to climate risks (such as agriculture, energy, and tourism), as well as the climate risk industry itself (such as insurance and risk assessment).”

---

> ### Author Response · Authors · 2023-08-22
> **Authors' response to Reviewer ujho (2/2)**
>
> >Concerning the previous point, some characteristics of the final goal could be highlighted. Are there any real-world operational constraints to consider? For example, memory and computational budget, fast inference, stability? In general, I think one should we include these into the assumptions and metrics catalogue because often scale and stability/consistency are some determining factors for whether a model gets adopted or not.
>
> To be operationally useful, ML models should avoid using more memory than the cloud-resolving subcomponents of the MMF they are meant to replace, and should ideally infer one or more orders of magnitude faster (e.g. Rasp et al. 2018 reported 20x speedup). We estimate this memory ceiling at 1 MB per grid column and inference floor at 300 ms per inference. These numbers are based on various performance tests of the climate simulator using GPUs at various HPC centers. Besides, we have included a table summarizing the inference cost (FLOPS) (see Table 2 in SI).
>
>
> >The metrics are complete. However, are there any spectral metrics to consider (as is often done with fields)? In addition, I didn’t see any breakdown of the metrics per region, time period, etc. Optimizing the best metric based on the mean is a “simple task,” but there could also be other spatiotemporal/spectral diagnostic quantities that showcase whether the model did a good job or not.
>
> We agree those are important but nontrivial to assess at sufficient scope. Thankfully, the published benchmark and diagnostic code repository of Mooers et al. (2021) provides a convenient path to adding regional and spectral metrics with precedent in the literature. We have added a Github issue to build the necessary connector from ClimSim’s metrics to those of Mooers et al. On a related note, in the revised manuscript, we have included comprehensive 2-D global metric (R2) maps to allow some assessment of the horizontal patterns of model performance [Figures SI17 to SI20].
>
> (Ref: Mooers, G., Pritchard, M., Beucler, T., Ott, J., Yacalis, G., Baldi, P., & Gentine, P. (2021). Assessing the potential of deep learning for emulating cloud superparameterization in climate models with real-geography boundary conditions. Journal of Advances in Modeling Earth Systems, 13, e2020MS002385. https://doi.org/10.1029/2020MS002385)
>
> >Limitations: The authors addressed some of the limitations of their work. However, I'm very surprised there was no mention of the societal impact due to the processing power required. There could even be positive ones, for example:
> \* positive could be the fact that the simulation was done on a very streamlined/efficient system;
> \* large consortium of interested parties agreed on a dataset to ensure it isn't wasted.
> A brief statement might be worth adding.
>
> We have added a brief positive statement to Section 3 “ClimSim Data Construction” [Lines 186-193].
>
> [Lines 186-193] “Energy use: The computing and energy costs of generating ClimSim could be viewed as wasteful and having a negative consequence for society through associated emissions. We emphasize that while it can appear large, the compute used is actually orders of magnitude less than what is consumed by operational climate prediction. Associated emissions are minimized given that our integrations were performed on energy-efficient GPU hardware. The cost must also be weighed against the potential social benefit of mitigating future energy consumption by eliminating end users' need for costly physics-based MMF simulations. Meanwhile, a large consortium of interested parties have helped agree on this dataset, to help ensure it is not wasted.”
>
>
> >The benchmark system should be more ML-like, where things are organized in a more ML fashion. While browsing, I found no clear introduction for new users as to what to look at first. For example, a better template could be - https://github.com/ashleve/lightning-hydra-template. The dataset is massive, and the challenge is great, so a more ML-like organization is necessary.
>
> We have updated the repo and its documentation. For details, please refer to our initial comments to all reviewers on the repo reorganization and quickstart.
>
>
> >I would like to see a more complete introduction on the website posted. I assume there will be more work done (which is fine). However, I want to reiterate that although the paper is quite dense with information that may be helpful conceptually, I think the code base should also exhibit the same completeness and cater to new users.
>
> We have updated the repo and its documentation. For details, please refer to our initial comments to all reviewers on the repo reorganization and quickstart.

---

> > ### Comment · Reviewer_ujho · 2023-08-30
> >
> > I thank the authors for addressing all of my comments. I also greatly appreciate the improvement in the repo and documentation. I think this paper is definitely in the upper echelon of Dataset and Benchmark papers and I cannot wait to see what the community produces in the future.

---

> > > ### Author Response · Authors · 2023-08-31
> > >
> > > Thank you very much once again for your encouraging message and the invaluable feedback you've provided, which led to the critical improvement of our manuscript. We are committed to maintaining and expanding our dataset through future releases. If you have any further suggestions or feedback in the future, we would be more than grateful to receive them.

---

### Official Review · Reviewer_uz8R · 2023-07-21
**A dataset for multi-scale climate simulations with potential impact on the accuracy of (global) climate simulations**

**Rating:** 10
**Confidence:** 4

**Strengths:**

* An impressive 40 TB of data, split into training and test sets
* 3 scenarios at different scales or complexity
* Baseline results for several ML approaches (MLP, RPN, CNN, HSR, cVAE)
* A set of Python scripts that were used to produce the baseline results, which can serve as a template for new methods


**Additional Feedback:**

* See the comments above.
* The review was not a blind one, as all authors' names and affiliations were listed in the submission.
* The supplementary material as well as the information on the github website are really helpful.

**Clarity:**

The paper is well written.

One minor issue: "10, 3D variables" -> "10 3D variables"


**Correctness:**

The paper lives up to the claims in the abstract.

The applied methodology for the creation of the dataset seems to be sound.
I do not have the knowledge and possibility to reproduce the datasets.

The benchmark data is explained with helpful details on the datasets' characteristics in Figure 1 and more information in the appendix.


**Documentation:**

The documentation is sound.

Suggestions for improvement:
* The documentation on github has a section on Preprocessing the Data. I would recommend to reformulate it to give a clear guideline on what to do for non-domain-experts.
* The Readme.txt files on huggingface should at least contain a back-ref to the github page of the benchmark dataset.

**Ethics:**

Not applicable. Synthetic simulation data.


**Limitations:**

Limitations of the given approach and data set are discussed in section 5.


**Opportunities For Improvement:**

* Instructions on how to generate more data with the help of the simulation code that was used would be a valuable add-on for, e.g. active learning. This could be scope of a further extension.
* Do improvements based on the presented dataset only improve the performance of climate simulations based on the E3SM-MMF multi-scale climate simulator of the authors? I was missing some information on the potential impact and the possibility to use the local sub-model results for other climate simulators.
* It is great to have raw data, including a discussion in Section 4.4. of how preprocessing could improve the performance of predictive models. However, this makes it also difficult to use the data as a benchmark for ML methods. It will be difficult to judge whether performance improvements have been achieved by better ML models or sophisticated preprocessing. Here a clear suggestion of how to use the data set by default would be helpful for the ML community.
* I had to start reading and interpreting the Python scripts provided to get started. An easy access for non-domain experts to just try out their own methods is missing.


**Relation To Prior Work:**

Information on prior work is provided.


**Summary And Contributions:**

The ClimSim dataset provides data to substitute highly-resolved computationally-expensive spatially-local simulation in climate simulations by ML models. The overall simulation framework is a two-scale simulation, where an ML-based surrogate is sought to replace expensive local simulations. They are required to provide macroscopic input parameters mapping inputs such as temperature fields and humidity distribution to heating tendency and rain rate, e.g.

The authors additionally provide baseline results for five methods and exemplary Python Worksheets to get started. However, a discussion of the basic performance of the baseline results is missing. The only comment is that MLP, which outperforms the other approaches in pretty much all metrics, has been "highly tuned".

While I am not a climate expert, I assume that significant improvements in the ML-based surrogate models can really have a real-world impact on the accuracy of climate simulations with a relevant code (E3SM-MMF multi-scale climate simulator).

---

> ### Author Response · Authors · 2023-08-22
> **Authors' response to Reviewer uz8R (1/2)**
>
> We appreciate the reviewer's time and insightful feedback on our paper. It's rewarding to see the recognition of key aspects of our ClimSim dataset—volume, complexity hierarchy, baseline models, and Python script repository—which we dedicated time and effort to. Our revision now reflects the reviewer's suggestions. If any points in our response are unclear or if you have new comments or questions, please inform us. We are committed to addressing them promptly.
>
> Point-by-point response:
>
> >Instructions on how to generate more data with the help of the simulation code that was used would be a valuable add-on for, e.g. active learning. This could be the scope of a further extension.
>
> Unfortunately, the climate simulation code is tailored for specific national supercomputers and requires specialized configuration and domain expertise. Nonetheless, we already plan on providing guidance on generating more data that would benefit practitioners in the ClimSim webpage. We have added this action item to github issues. Besides, in Section 5 “Limitations and Other Applications” [Lines 356-359], we have added active learning and relevance determination as challenges of interest that can be addressed with the current data and metrics provided.
>
> [Lines 356-359] “Relevance determination and active learning: While the climate simulator code offers data generation flexibility, guidance on ideal regimes to target for improved learning would benefit the domain scientists able to run it. This question can be addressed with the current data and metrics of interest provided.”
>
> >Do improvements based on the presented dataset only improve the performance of climate simulations based on the E3SM-MMF multi-scale climate simulator of the authors? I was missing some information on the potential impact and the possibility to use the local sub-model results for other climate simulators.
>
> Insights about which models work will be of interest to other climate modeling centers, such as the effect of stochasticity, spatial dependencies and runtime-efficient architectures. ClimSim's learnt parameterizations should also be directly adaptable to the world’s most widely used open source climate simulator - the NCAR Community Earth System Model (CESM) - given its software similarities to E3SM through common code lineage. We have added this information to the SI under a new subsection on “2.5 Target Audiences” [Lines SI 188-203].
>
> [Lines SI 188-203] “2.5 Target Audiences
> In essence, this benchmark aims to democratize and expand access to advanced climate modeling. High-potential architectures will undergo testing in the superparameterized version of the DOE's primary climate model, E3SM. Successful integration would substantially reduce computational costs for the DOE when contemplating the deployment of MMF technology in climate prediction. E3SM's external user community, typically deterred by the extensive computational demands of superparameterized simulators, also stands to benefit. Currently, only a minority with substantial computing resources can engage with such models. A successful recipe for ClimSim could thus democratize the use of explicit convection for a broader user base. If performant architectures also prove effective in the NCAR Community Earth System Model (CESM) - the world’s most widely used open source climate simulator - the user base could expand significantly. Given its software similarities to E3SM, it is logical to expect that ClimSim's learnt parameterizations will be readily adaptable to CESM. Moreover, we anticipate that a successful hybrid machine learning climate simulator will bring benefits to a diverse range of industry sectors, including those vulnerable to climate risks (such as agriculture, energy, and tourism), as well as the climate risk industry itself (such as insurance and risk assessment).”
>
> >It is great to have raw data, including a discussion in Section 4.4. of how preprocessing could improve the performance of predictive models. However, this makes it also difficult to use the data as a benchmark for ML methods. It will be difficult to judge whether performance improvements have been achieved by better ML models or sophisticated preprocessing. Here a clear suggestion of how to use the data set by default would be helpful for the ML community.
>
> We direct you to our initial comments regarding the repo reorganization and quickstart. While we concur on the value of a user-friendly default dataset for ease of comparison, our work uniquely offers a large-scale regression dataset to fill existing gaps.
>
> >I had to start reading and interpreting the Python scripts provided to get started. An easy access for non-domain experts to just try out their own methods is missing.
>
> We have added a quickstart. For details, please refer to our initial comments on the repo reorganization and quickstart.

---

> ### Author Response · Authors · 2023-08-22
> **Authors' response to Reviewer uz8R (2/2)**
>
> >Clarity: The paper is well written. One minor issue: "10, 3D variables" -> "10 3D variables"
>
> We have corrected the typo as suggested.
>
> >The documentation on github has a section on Preprocessing the Data. I would recommend reformulating it to give a clear guideline on what to do for non-domain-experts.
>
> We have updated github documentation. For details, please refer to our initial comments on the repo reorganization and quickstart.
>
> >The Readme.txt files on huggingface should at least contain a back-ref to the github page of the benchmark dataset.
>
> We have added a back-ref on Hugging Face repositories. For details,  please refer to our initial comments on the repo reorganization and quickstart.

---

> > ### Comment · Reviewer_uz8R · 2023-08-27
> > **Reply**
> >
> > Thank you for the detailed answers and the improvements in the paper, appendix and online repo.

---

> > > ### Author Response · Authors · 2023-08-31
> > >
> > > We would like to extend our gratitude once more for your insightful feedback, which was critical for us to revise our manuscript for better clarity. Please don't hesitate to reach out if you have any further suggestions, feedback, or requirements.

---

### Official Review · Reviewer_pLrq · 2023-07-21

**Rating:** 8
**Confidence:** 3
**Correctness:** As far as I can tell, yes. However, I…

**Strengths:**

I see several clear strengths of the paper. My comments on the strengths are brief because there is not much to say.

### Public well-done runs from a simulator

Getting credible simulators to work properly requires expertise that is usually beyond the ML community. I do not work in climate science, but do work in another area with similar characteristic (astrophysics). Having this data available at the given scale and with the given format is incredibly helpful  and a great opportunity for the community.

### Extensive benchmarking

The authors have tried a variety of flavors of methods for solving the problem and have the results in the paper. These results will be helpful for the community.

### Good analysis and many extra potential directions pointed to

The paper does a good job of pointing to issues and potential opportunities for researchers.
- While there are some additional analyses that would improve the work, the current analysis is more than adequate for a paper.
- There are plenty of ideas for how to extend the the proposed baselines. Sometimes it is claimed that the onus falls on the researchers producing the dataset to run these extensions themselves. I'd preemptively push-back on this point. Simply having a basic version of methods is extremely helpful and often enough to get the community started.

**Additional Feedback:**

I like this paper a lot and would like to see it accepted. I have a feedback that's meant to improve the paper, not meant as a criticism.

Various experimental quibbles:

- The baselines aren't quite apples to apples. Having done similar projects myself, I recognize that it's genuinely hard to do this, but there are some basic gaps. However: (a) the MLP uses RAdam, which does decoupled weight-decay (i.e. following Loshchilov and Hutter) while the CNN uses original Adam and not AdamW; (b) the MLP is trained with a MSE, which is minimized by the conditional expectation and spreads the weights equally across the target outputs; the CNN is trained to minimize a MAE, which is minimized by the conditional median (the weighting here is required for apples-to-apples comparison).
- The paper would probably benefit from a separated validation and test set.

Various paper suggestions:
- The claim that most image-to-image problems don't have as many outputs isn't really true, as many problems are multi-class problems (eg for segmentation or regression-by-classification) and often have as many outputs
- Fig 1. The top-most node of the second layer of the MLP doesn't connect to the third-from-the-top node in the third layer of the MLP.
- L135 "21,600" comes out of nowhere. I can't figure this number out -- the paper suggests the discretization is 1.5 deg x 1.5 deg. If it's 1.5x1.5 degree, then a 180x360 degree map should be (180/1.5) * (360/1.5) = 28,000 pixels.
- Table 2: Table 2 could benefit from re-explanation of what the particular variables are, so that the reader does not have to flip back to Table 1. I think this is feasible in the space, given the extra whitespace. The caption might benefit from more details

**Clarity:**

Largely yes, apart from input/output sizes as discussed in opportunities for improvement

**Documentation:**

Yes, I believe so.

**Ethics:**

No ethics concerns.

**Limitations:**

Yes, the manuscript clearly addresses these, far more so than most papers in my stack.

**Opportunities For Improvement:**

I see three opportunities for improvement with the paper These are longer than my strengths out of an attempt to help the authors understand my points. However, I think the weaknesses are not reasons to reject the paper. The first is a clarity issue that just requires rewriting (which I trust the authors are capable of doing). The second is more thorough analysis, which can be fixed via re-analysis of the existing data. The final ones are things that the authors may wish to do. I do not see them as critical to accepting the paper, but I do think that they would improve the final product or may be useful to provide for the community.

### Clarity about inputs/outputs and precise problems solved

The paper would be substantially improved by being more direct and clear about input/output-sizes and terminology.
Here are a few examples:
- Figure 1 suggests that there are 124 inputs and 128 outputs. The text in Section 3 suggests that it's 617 inputs and 368 outputs until Section 4 is reached and then it is explained that only 124 inputs and 128 outputs are handled in the paper and benchmark. I think the paper would be stronger if it focused from the start with the smaller number and explained clearly why the larger task couldn't be tackled, or perhaps showed one method on the full data.
- Much of the data is suggested to be 2D "For 2D tensors, the dimensions represent time and horizontal location". I'm not sure what horizontal location means. Is this longitude, latitude, the data after lat/lon is flattened into a single vector?
- The preprocessing steps suggest that data "collapse horizontal location and time into a single sample dimension". This was enormously confusing at first. Part of this is perhaps the use of "horizontal location" and part of it is confusion as to what sample dimension means -- is it a dimension along which the sample index runs? is it the dimensionality of the sample?  If I understand correctly, what this actually means is that the atmospheric columns at each location and time are treated as independent samples. So if there are T simulation steps used for training, and each simulation has K grid cells, then there are T*K samples.
- if the data is treating each grid sample as independent, then it's not clear where the 1D CNN comes in. It's mentioned that the inputs are 60x6, but where 60 comes from requires reading the paper carefully, and assuming that the 60 elsewhere is the same 60. Adding a simple signpost that the number of dimensions is the discretization of the atmospheric column would help tremendously.

Each issue is possible to resolve via some guess-work and understanding of how these sorts of simulations work (albeit in astrophsyics, not climate science), but a less interested/invested reader may give up.

### Results could use more thoroughness

The results would use an increase in thoroughness of the analysis. Here are a few comments:
- Given the hope of using ML-based models to improve simulations, I'm surprised that there are no plots of the joint density of the outputs and the ground-truth (i.e., plt.scatter(y,model(x))). I'd assume many of these target variables may be multimodal in terms of output, and many ML models may split the difference, leading to nonsense results.
- At a minimum, it may be good to compute bias for over/under prediction. It is true that that biases can often be correct post-facto via a calibration factor, but knowing what out of the box performance would be is important. This is important for the non-negative variables, where errors may not be able to average out to zero: e.g., the lower tails of the noise will be truncated at zero.
- Error bars would be helpful, especially given that the paper is proposing a dataset + benchmark. Having these would ensure there was public knowledge of the sort of variability expected. This can help identify which gains are real and which ones are likely due to lucky seeds.

### Various experiments would improve the paper

The paper will, I think, likely to be a strong contribution to the community. There are a few things that I wish had been done, which the authors may wish to do before their release.
- Results with the full data. It's surprising, at least to me, that the data doesn't use the full set of variables. I'd understand if the data were spatially-varying too (e.g., HxWx617 tensors), but if it's 1D, then I don't see why 600 inputs is too much: 600 inputs is about the size of a typical feature.
- Results with even adjacent cells tacked on to analyze spatial information.
- Even a basic integration into a downstream task might be tremendously helpful for impact. A full integration is the type of project deserving a large multi-year, multi-PI program, but even a basic test would be helpful for motivation and for going beyond the current metrics for performance.

**Relation To Prior Work:**

I believe so, but again I am not a climate scientist.

**Summary And Contributions:**

Update: in light of the authors' strong response, I've increased my score to 8.

---------------

The paper proposes a climate simulation dataset and benchmark. ML models for climate simulation are likely to provide substantial gains. Past work in this area has been on simplified simulations, with limited baselines, and evaluations. The proposed work produces a dataset from a large simulation as well as multiple baselines that are evaluated for their ability to predict the simulation.

---

> ### Author Response · Authors · 2023-08-22
> **Authors' response to Reviewer pLrq (1/4)**
>
> We extend our heartfelt appreciation to the reviewer for their generous commitment of time in meticulously evaluating our paper and providing us with invaluable feedback.  It is rewarding to hear that the review  “liked this paper a lot and would like to see it accepted.” We have taken the reviewer’s comments seriously and have worked hard to further improve our manuscript. We eagerly welcome ongoing discussions during this open review phase. If any of our responses warrant additional clarification or if new comments and questions come up, please let us know without hesitation. We will try our best to address these matters promptly. Thank you once again for your valuable feedback.
>
> Point-by-point response:
>
> >Figure 1 suggests that there are 124 inputs and 128 outputs. The text in Section 3 suggests that it's 617 inputs and 368 outputs until Section 4 is reached and then it is explained that only 124 inputs and 128 outputs are handled in the paper and benchmark. I think the paper would be stronger if it focused from the start with the smaller number and explained clearly why the larger task couldn't be tackled, or perhaps showed one method on the full data.
>
> We have acted on this by retraining a new  MLP baseline to map the complete 617 inputs to 368 outputs for both low- and high-resolution dataset, including a series of ~10,000 trial hyperparameter sweeps (refer to Section 3.1 in SI for further details). Full vector emulation improves predictions uniformly and especially for precipitation. These results have been incorporated into a new subsection in the main narrative focusing on the effects of expanding features and targets (Section 4.2 “Skill boost from Expanding Features and Targets; See Table SI4 and Figures SI7-8)
>
>
> >Much of the data is suggested to be 2D "For 2D tensors, the dimensions represent time and horizontal location". I'm not sure what horizontal location means. Is this longitude, latitude, the data after lat/lon is flattened into a single vector?
>
> We admit the original manuscript lacked a clear description on the ‘horizontal location’. We have added more description on the climate simulator’s horizontal grid in “Section 3 - Dataset Collection” [LINES 150-154]. Briefly, the computational grid of the climate simulator uses an “unstructured” cube-sphere mesh, not a regularly gridded latitude-longitude Cartesian grid such that in a 2D tensor [time, ncol], each ncol is mapped to a specific location, e.g, (lat[ncol], lon[ncol]).
>
> [Lines 150-154] “... the high-resolution simulation that uses an unstructured ``cube-sphere" horizontal grid with 21,600 grid columns spanning the globe. This grid yields an \textit{approximate} horizontal grid spacing of 1.5$^\circ$, but unlike a traditional climate model that maps points across the sphere using two dimensions aligned with cardinal north/south and east/west directions, unstructured grids use a single dimension to organize the horizontal location of points.”
>
>
> >The preprocessing steps suggest that data "collapse horizontal location and time into a single sample dimension". This was enormously confusing at first. Part of this is perhaps the use of "horizontal location" and part of it is confusion as to what sample dimension means–is it a dimension along which the sample index runs? is it the dimensionality of the sample? If I understand correctly, what this actually means is that the atmospheric columns at each location and time are treated as independent samples. So if there are T simulation steps used for training, and each simulation has K grid cells, then there are T*K samples.
>
> We apologize for the confusion. Your understanding is correct and we have adopted your suggested text to make it clearer. Please see the updated text in Lines 148-158.
>
> [Lines 148-158] “These data were saved at 20-minute intervals (i.e. the time step of the climate model) for 10 simulated years, resulting in 5.7 billion samples for the high-resolution simulation that uses an unstructured ``cube-sphere" horizontal grid with 21,600 grid columns spanning the globe. This grid yields an \textit{approximate} horizontal grid spacing of 1.5$^\circ$, but unlike a traditional climate model that maps points across the sphere using two dimensions aligned with cardinal north/south and east/west directions, unstructured grids use a single dimension to organize the horizontal location of points. The atmospheric columns at each location and time are treated as independent samples. Thus, the total number of samples can be understood by considering that atmospheric columns at each location and time are treated as independent samples, such that 5.7 billion $\approx$ 21,600 horizontal locations per time step $\times$ 72-time steps per simulated day $\times$ 3,650 simulated days).”

---

> ### Author Response · Authors · 2023-08-22
> **Authors' response to Reviewer pLrq (2/4)**
>
> >if the data is treating each grid sample as independent, then it's not clear where the 1D CNN comes in. It's mentioned that the inputs are 60x6, but where 60 comes from requires reading the paper carefully, and assuming that the 60 elsewhere is the same 60. Adding a simple signpost that the number of dimensions is the discretization of the atmospheric column would help tremendously.
>
> We agree this was not adequately explained. We have modified the main text to clearly specify the discretization of the atmospheric column in Lines 158-159
>
> [Lines 158-159] “It is important to note that each sample retains a 1D structure corresponding to the vertical variation across 60 levels.”
>
>
> >Each issue is possible to resolve via some guess-work and understanding of how these sorts of simulations work (albeit in astrophysics, not climate science), but a less interested/invested reader may give up.
>
> We apologize for this blind spot, and are very grateful for the suggestions that have reduced the barrier this will pose to future readers, and for the time spent making constructive suggestions that have improved clarity.
>
>
> >Given the hope of using ML-based models to improve simulations, I'm surprised that there are no plots of the joint density of the outputs and the ground-truth (i.e., plt.scatter(y,model(x))). I'd assume many of these target variables may be multimodal in terms of output, and many ML models may split the difference, leading to nonsense results.
>
> Our prediction problem's multidimensional and multivariate nature posed concerns about overwhelming readers. However, in alignment with the reviewer's feedback, we have created a new section in the SI (Section 4.3 “Fit Quality”; Lines SI 431-436) to add pertinent hexagonally-binned histograms (Fig. SI9 to SI16). While many variables exhibit consistent fit quality, some show notable variability between baselines, as seen with snow precipitation rate predictions. The multi-variate performance of our optimized deterministic baseline (MLP) suggests these issues are avoidable. We appreciate the reviewer's insight into the challenges that can be revealed in certain vector subsets. Our new scatterplot visualizations in the SI emphasize these and will be helpful to guide future optimization efforts.
>
> [Lines SI 431-436] “4.2 Fit Quality
>  Scatter plots of truth versus prediction are shown in this section (Figures SI9 to SI16 in SI Section 8). While many variables exhibit consistent fit quality, some show notable variability between baselines, as seen with snow precipitation rate predictions. The performance of our optimized deterministic baseline (MLP) suggests these issues are avoidable. However, note that our prediction problem has a multi-variate and multi-dimensional nature.”
>
>
> >At a minimum, it may be good to compute bias for over/under prediction. It is true that biases can often be correct post-facto via a calibration factor, but knowing what out of the box performance would be is important. This is important for the non-negative variables, where errors may not be able to average out to zero: e.g., the lower tails of the noise will be truncated at zero.
>
> We thank the review again for this comment. As answered above, the newly added figures provide the information requested in this comment. Please let us know in through OpenReview comments if you need to see other types of visual presentations. We will be happy to provide more analysis and visualizations.
>
>
> >Error bars would be helpful, especially given that the paper is proposing a dataset + benchmark. Having these would ensure there was public knowledge of the sort of variability expected. This can help identify which gains are real and which ones are likely due to lucky seeds.
>
> We agree and thank the reviewer for the constructive suggestion. We added a measure of variability in model performance by scoring 160 of the top-performing MLP models selected from our ~8k-member HPO ensemble; the revealed variability was added as error bars and grey shadings to Figure 2, and we hope the reviewer agrees the variability is reassuringly small. New text in the SI elaborates on how the model variability was measured [SI Lines 238-242].
>
> (Quotes to be deleted)
> [SI Lines 238-242] “To provide some context on the amount of variance in model performance that can be attributed to random effects of optimization, the top 160 models were selected from our pool of 8,257 trials and scored on the validation set; the 5th to 95th percentile range of this ensemble is shown by the error bars in Figures 2a and SI3, and by the grey shading in Figures 2b-e, SI4, and SI5.”

---

> ### Author Response · Authors · 2023-08-22
> **Authors' response to Reviewer pLrq (3/4)**
>
> >Results with the full data. It's surprising, at least to me, that the data doesn't use the full set of variables. I'd understand if the data were spatially-varying too (e.g., HxWx617 tensors), but if it's 1D, then I don't see why 600 inputs is too much: 600 inputs is about the size of a typical feature.
>
> We concur. A distinctive feature of our dataset is its enhanced input and output vector dimensions, aiming to capture all causatively pertinent factors. The revised manuscript now features an MLP baseline for full vector emulation in Section 4.2 [Lines 242-254].
>
> [Lines 242-254] “4.2 Skill Boost from Expanding Features and Targets
> We performed an ablation of our best performing MLP baseline to demonstrate the added value of the expanded inputs and targets available in \dataset{}, i.e. using inputs $x$ of size $d_i=617$ and targets $y\in\mathbb R^{d_o}$ of size $d_o=368$; see Table 1 in SI for the full list of variables. We use the same transformation described in our preprocessing workflow to compute and add condensate (cloud liquid and cloud ice) and momentum (zonal and meridional winds) tendencies to the target vector. We conducted this ablation study with both the low-resolution and the high-resolution datasets (see Section 3.1 in SI for further details regarding these MLP variants). For common elements of the target vector, using all available variables leads to a uniform improvement in prediction accuracy, especially for precipitation, in both resolutions (Figures SI7, SI8 and Table SI4). The larger errors (e.g., MAE and RMSE) observed in the high-resolution emulators are anticipated due to the increased variance of higher-resolution data. Nevertheless, the similarity of their R$^\text{2}$ values to those of the corresponding low-resolution emulators confirms their adequate performance.”
>
> >Results with even adjacent cells tacked on to analyze spatial information.
>
> We acknowledge the importance of considering spatial adjacency. However, integrating it can complicate the downstream simulation pipeline due to the parallel domain decomposition challenges of the host climate simulator. To facilitate future exploration in this direction, we've added a git issue to provide examples of how to identify the spatially adjacent grid columns from the simulator’s unstructured mesh, for potential pre-processing enhancements.
>
> >Even a basic integration into a downstream task might be tremendously helpful for impact. A full integration is the type of project deserving a large multi-year, multi-PI program, but even a basic test would be helpful for motivation and for going beyond the current metrics for performance.
>
> We are glad the reviewer shares our own enthusiasm for pursuing the downstream integration test. Regretfully, even a rudimentary test demands software engineering beyond the constraints of the current review period. We are committed to undertaking this effort in the long term, with intentions to launch a subsequent competition phase centered on tests of the downstream signal. In the interim, for the proactive developers, we've added a git issue to provide necessary references to relevant literature and external code repositories exemplifying hybrid simulation, and have made a plan to generate initial documentation highlighting the distinct approach that our expanded input and output vectors will require.

---

> ### Author Response · Authors · 2023-08-22
> **Authors' response to Reviewer pLrq (4/4)**
>
> >The baselines aren't quite apples to apples. Having done similar projects myself, I recognize that it's genuinely hard to do this, but there are some basic gaps. However: (a) the MLP uses RAdam, which does decoupled weight-decay (i.e. following Loshchilov and Hutter) while the CNN uses original Adam and not AdamW; (b) the MLP is trained with a MSE, which is minimized by the conditional expectation and spreads the weights equally across the target outputs; the CNN is trained to minimize a MAE, which is minimized by the conditional median (the weighting here is required for apples-to-apples comparison).
>
> We apologize for the inconsistencies. Due to difficulties in coordinating across so many authors, it was not feasible to control for all elements such as fully isolating effects of optimizer choice from effects of architecture. We will attempt to standardize more model parameters (including optimization) for baseline models in future ClimSim releases. Meanwhile, we have added the following text to acknowledge the inconsistencies in the beginning of Secion 3 “Baseline Models” in the SI:
>
> [Lines SI 205-211] “This section offers a detailed depiction of six baseline models. Every facet of model designs, excluding the dimensions of the input and output layers, differs among the models. We recognize that while this approach maximizes the differentiation among baseline models, such extensive degrees of freedom complicate the complete isolation of the effects arising from optimization parameter choices and those originating from the model architecture itself. In future ClimSim releases, baseline models will share more constraints (including optimization parameters) to highlight the performance difference due to model architectures.”
>
>
> >The paper would probably benefit from a separated validation and test set.
>
> As described in Section 3, our dataset comes with a typical train/test split common in machine learning. However, in this paper, we did not use the test set in any model development / evaluation stage and used the validation set for model performance evaluation presented in our manuscript.  We will provide an additional public leaderboard on Huggingface for final performance reporting purposes, to exploit a completely independent test set, which we have held out for this purpose. Only then, the test set will be released publicly.
>
>
> >The claim that most image-to-image problems don't have as many outputs isn't really true, as many problems are multi-class problems (eg for segmentation or regression-by-classification) and often have as many outputs
>
> We agree and have removed the text that misleadingly suggested that we have more channels in this problem compared to image-to-image problems.
>
> >Fig 1. The top-most node of the second layer of the MLP doesn't connect to the third-from-the-top node in the third layer of the MLP.
>
> We have edited the schematic in Figure 1 to implement the desired connections.
>
>
> >L135 "21,600" comes out of nowhere. I can't figure this number out -- the paper suggests the discretization is 1.5 deg x 1.5 deg. If it's 1.5x1.5 degree, then a 180x360 degree map should be (180/1.5) * (360/1.5) = 28,000 pixels.
>
> We agree that this number is not adequately explained in the text. As mentioned in our above responses, we have added more description regarding the “unstructured” grid [Lines 148-158].
>
> [Lines 148-158] “These data were saved at 20-minute intervals (i.e. the time step of the climate model) for 10 simulated years, resulting in 5.7 billion samples for the high-resolution simulation that uses an unstructured ``cube-sphere" horizontal grid with 21,600 grid columns spanning the globe. This grid yields an \textit{approximate} horizontal grid spacing of 1.5$^\circ$, but unlike a traditional climate model that maps points across the sphere using two dimensions aligned with cardinal north/south and east/west directions, unstructured grids use a single dimension to organize the horizontal location of points. The atmospheric columns at each location and time are treated as independent samples. Thus, the total number of samples can be understood by considering that atmospheric columns at each location and time are treated as independent samples, such that 5.7 billion $\approx$ 21,600 horizontal locations per time step $\times$ 72-time steps per simulated day $\times$ 3,650 simulated days).”
>
>
> >Table 2: Table 2 could benefit from re-explanation of what the particular variables are, so that the reader does not have to flip back to Table 1. I think this is feasible in the space, given the extra whitespace. The caption might benefit from more details.
>
> We agree, and have added the descriptions of each variable to the Table 2 caption along with a more detailed explanation of the contents of the table.

---

> > ### Comment · Reviewer_pLrq · 2023-08-28
> >
> > I thank the authors for their their improvements to the paper. I continue to be very excited about this paper and will argue for its acceptance. I think having these additional details will improve the impact and will update my score accordingly.
> >
> > One minor detail that is just me being a bit picky: some of the figures would be much better as a pdf or re-exported with higher dpi (e.g., supplement figures 9-13). Right now, the rasterization is pretty easy to spot and isn't particularly aesthetically appealing. It's up to the authors as to whether they wish to do this though.

---

> > > ### Author Response · Authors · 2023-08-29
> > > **We have updated our figures as suggested.**
> > >
> > > Thank you once again for your encouraging comments!
> > >
> > > We agree that the figures (especially the texts in them) were not clear enough. We have re-uploaded both the main document and the Supplementary Information after replacing the original figures with vectorized versions. We hope this adjustment will improve the clarity and readability of our figures.

---

### Official Review · Reviewer_HQ4J · 2023-07-23
**This paper presents a largest-ever and multi-scale climate simulation dataset designed for ML-physics research.**

**Rating:** 7
**Confidence:** 4
**Correctness:** Yes, the claims in this paper are cor…
**Clarity:** Yes.

**Strengths:**

- This climate dataset is the largest and most physically comprehensive for hybrid-ML research. Also, a benchmark is provided to serve as a foundation for future research.

- This paper is well-written and very detailed in the context of dataset and benchmark.

**Additional Feedback:**

- On Page 5, line 147, could the authors describe more on the horizontal grid? For the climate dataset, I thought it would be 2D snapshots/images with temporal evolution. I am curious why only 1D signals are extracted and tested.

- On Page 5, line 185, the bracket directly goes from (3) to (5).

**Documentation:**

Yes, the data detail is sufficient.

**Ethics:**

No.

**Limitations:**

Yes, the authors have included a comprehensive discussion on limitations.

**Opportunities For Improvement:**

- In Section Experiments, the authors only test on a low-resolution dataset. It would help the readers have a better sense of this climate dataset if the authors could also test on one subset of the high-resolution dataset.

- In Section 4.2, this paper considers MAE and $R^2$. These metrics measure pixel-wise differences. For climate data, it would be good to consider some multi-scale metrics, e.g., Anomaly Correlation Coefficient (ACC) [1].

Reference:

[1] Pathak, J., Subramanian, S., Harrington, P., Raja, S., Chattopadhyay, A., Mardani, M., ... & Anandkumar, A. (2022). Fourcastnet: A global data-driven high-resolution weather model using adaptive fourier neural operators. arXiv preprint arXiv:2202.11214.

**Relation To Prior Work:**

Yes, this paper clearly discussed the difference between this work and the existing papers.

**Summary And Contributions:**

This paper introduces a substantial and high-resolution climate dataset that encompasses a comprehensive range of associated factors, both in terms of temporal and spatial dimensions. The primary objective of this dataset is to address the existing gap in macro-scale hybrid-ML climate simulation research, which has been attributed to the limitations in dataset inclusiveness and the necessary expertise within the climate field. The author additionally furnishes baseline experiments of climate prediction with one of their low-resolution subsets, showcasing the versatility and utility of ClimSim.

---

> ### Author Response · Authors · 2023-08-22
> **Authors' response to Reviewer HQ4J**
>
> We thank the reviewer for taking time to assess our manuscript and offering valuable suggestions. We are excited to hear that the reviewer found our ClimSim dataset comprehensive and our manuscript well-written and very detailed. Following the reviewer’s feedback, we have updated our manuscript accordingly. Please let us know if any of our response is unclear or if a new question or comment comes up. We will do our best to clarify any ambiguities and further comments.
>
> Point-by-point response:
>
> >In Section Experiments, the authors only test on a low-resolution dataset. It would help the readers have a better sense of this climate dataset if the authors could also test on one subset of the high-resolution dataset.
>
> We fully agree with the review are have been working on building and testing MLP using the high-resolution dataset. We have just finished selecting the best models from a HPO with ~10k trials and are about to evaluate them. Once finishing this task, we will incorporate the results to the manuscript and post updates here on OpenReview.
>
> >In Section 4.2, this paper considers MAE and R2. These metrics measure pixel-wise differences. For climate data, it would be good to consider some multi-scale metrics, e.g., Anomaly Correlation Coefficient (ACC) [1].
>
> ACC is a popular metric for evaluating the prediction skill of global weather forecasting models, including FourCastNet, by summarizing the decorrelation of global weather patterns with lead time. However, this metric is not well-suited for ClimSim, which instead focuses on learning a spatially local machine learning parameterization rather than forecasting. To allow some assessment of the horizontal patterns, in lieu of ACC, we have generated global metric (R2) maps that reveal the geospatial structure of model performances. Please refer to Figures SI17 to SI20 in the SI of our revised manuscript.
>
> >On Page 5, line 147, could the authors describe more on the horizontal grid? For the climate dataset, I thought it would be 2D snapshots/images with temporal evolution. I am curious why only 1D signals are extracted and tested.
>
> The predictor-corrector multi-scale scaffold of the E3SMF is spatially and temporally local with the only remaining degree of freedom being altitude. The default ClimSim experiments are aligned with this (see Fig. 1). We agree that the description of the climate model’s computational grid was not adequately described. We have added more description about the horizontal grid under Section 3 “ClimSim Dataset Construction – Dataset collection” [Lines148-159]:
>
> [Lines 148-159] “These data were saved at 20-minute intervals (i.e. the time step of the climate model) for 10 simulated years, resulting in 5.7 billion samples for the high-resolution simulation that uses an unstructured ``cube-sphere" horizontal grid with 21,600 grid columns spanning the globe. This grid yields an \textit{approximate} horizontal grid spacing of 1.5$^\circ$, but unlike a traditional climate model that maps points across the sphere using two dimensions aligned with cardinal north/south and east/west directions, unstructured grids use a single dimension to organize the horizontal location of points. The atmospheric columns at each location and time are treated as independent samples. Thus, the total number of samples can be understood by considering that atmospheric columns at each location and time are treated as independent samples, such that 5.7 billion $\approx$ 21,600 horizontal locations per time step $\times$ 72-time steps per simulated day $\times$ 3,650 simulated days). It is important to note that each sample retains a 1D structure corresponding to the vertical variation across 60 levels.”
>
> > On Page 5, line 185, the bracket directly goes from (3) to (5).
>
> We have fixed the typo.

---

> > ### Comment · Reviewer_HQ4J · 2023-08-28
> >
> > Thanks for the responses. The authors have addressed my concerns. I have increased my confidence score: 3 -> 4.

---

> > > ### Author Response · Authors · 2023-08-31
> > > **We have added baseline models that uses full variable set with both low-resolution and high-resolution**
> > >
> > > Thank you once again for your constructive feedback that helped us improve our manuscript!
> > >
> > > We would like to bring to your attention an additional update. Building upon your first comment, we have just included MLP baselines that use the comprehensive list of variables in our revised manuscript. For further details, I kindly direct your attention to the authors' shared remark #3.

---

### Author Response · Authors · 2023-08-22
**Authors' shared remarks [1/3]: Repo reorganization and quickstart**

We thank all reviewers for the valuable comments and suggestions on our GitHub Repo [https://github.com/leap-stc/ClimSim/tree/main] and landing page [https://leap-stc.github.io/ClimSim], which has led to a thorough discussion among our team and a plan to reorganize our repo and set up clear guides on quick-start and community contributions. The purpose of this plan is three-fold: 1) to offer easy onboarding workflows for both machine learning and climate science researchers, 2) to build robust structures and policies for long-term data and code maintenance and 3) to enable rigorous version control for research reproducibility.

We have added the followings while preparing this response:
- Automated unit test infrastructure,
- Standardized code unifying preprocessing and evaluation,
- Quickstart datasets for non-power users, and
- Comprehensive introductory YouTube video [https://www.youtube.com/watch?v=M3Vz0zR1Auc].

Furthermore, we have tidied up our GitHub repository (e.g., restructuring code trees and removing unused and redundant files and directories) and added a list of actions as GitHub issues (under “Issues” tab in our repo) to track our progress. Additional issues from community researchers wishing to build off ClimSim or make use of our tools are welcome and can be raised using the GitHub issues page (preferred) or by directly emailing ClimSim maintainers.

Finally, we wish to assure reviewers that ClimSim will be an actively maintained set of data and tools with continued user support and future updates. Plans for the future include, but are not limited to:

- Copying datasets on HuggingFace into zarr-formatted ARCO (Analysis-Ready, Cloud Optimized) versions of the ClimSim dataset,
- Adding the ClimSim dataset to the UCI Machine Learning Repository,
- Quickstarts for non-domain-experts who wish to quickly try out their methods on reasonably-sized datasets,
- An online Kaggle-style competition that makes use of the to-date held-out test set we have for a private leaderboard, and
- Adding additional unit tests to streamline future refactoring.

---

### Author Response · Authors · 2023-08-22
**Authors' shared remarks [2/3]: A newly added baseline encoder-decoder (ED)**

In addition to five baseline models included in the initial submission, we have added a new baseline model based on an encoder-decoder (ED) architecture. The figures and tables are updated through the manuscript accordingly (e.g., Main Figure 2; SI Figures 3, 4, 5, 6; Main Table 2; SI Tables 2, 3, 4).

---

### Author Response · Authors · 2023-08-31
**Authors' shared remarks [3/3]: Newly added MLP baseline variants**

We have incorporated supplementary experiments involving the MLP baseline that include:
- MLP using a subset of variables with the low-resolution dataset
- (*New*) MLP using a subset of variables with the high-resolution dataset
- (*New*) MLP using the complete variable set with the low-resolution dataset
- (*New*) MLP using the complete variable set with the high-resolution dataset

Please refer to newly added Section 4.2 in the main text  (as excerpted below) for details. These additional experiments aim to enhance the comprehensiveness of our Climsim dataset, thus further reinforcing its valuable contribution to the ML community.

"**4.2 Skill Boost from Expanding Features and Targets**
We performed an ablation of our best performing MLP baseline to demonstrate the added value of the expanded inputs and targets available in \dataset{}, i.e. using inputs $x$ of size $d_i=617$ and targets $y\in\mathbb R^{d_o}$ of size $d_o=368$; see Table 1 in SI for the full list of variables. We use the same transformation described in our preprocessing workflow to compute and add condensate (cloud liquid and cloud ice) and momentum (zonal and meridional winds) tendencies to the target vector. We conducted this ablation study with both the low-resolution and the high-resolution datasets (see Section 3.1 in SI for further details regarding these MLP variants). For common elements of the target vector, using all available variables leads to a uniform improvement in prediction accuracy, especially for precipitation, in both resolutions (Figures SI7, SI8 and Table SI4). The larger errors (e.g., MAE and RMSE) observed in the high-resolution emulators are anticipated due to the increased variance of higher-resolution data. Nevertheless, the similarity of their R$^\text{2}$ values to those of the corresponding low-resolution emulators confirms their adequate performance."

---

### Decision · Program_Chairs · 2023-09-22

**Decision:**

Accept (Oral)

**Comment:**

The paper proposes a climate simulation dataset and benchmark. A climate dataset with varying resolutions of climate simulated data is produced. ML models for climate simulation are likely to provide substantial gains. Past work in this area has been on simplified simulations, with limited baselines, and evaluations. The proposed work produces a dataset from a large simulation as well as multiple baselines that are evaluated for their ability to predict the simulation. The authors additionally provide baseline results for five methods and exemplary Python worksheets to get started.

All reviewers, while not climate experts, see a potential for this benchmark,  due to its size, to enable the advancement of machine learning research for climate model surrogates.

All reviewers agree for the quality of the paper. Its a well written paper, and a well-organised benchmark dataset, with associated code, documentation, and Python workbooks to compare ML surrogate model performance for climate research. The work is original and stands apart the existing available datasets related to climate research, due to its big size, varying resolutions and comprehensive evaluation setup.

Key strengths include:
- An impressive 40 TB of data, split into training and test sets
- Three scenarios at different scales or complexity
- Baseline results for several ML approaches (MLP, RPN, CNN, HSR, cVAE)
- Data is hosted on the HuggingFace platform, which is excellent for visibility and ease of access.
-  A set of Python scripts that were used to produce the baseline results, which can serve as a template for new methods

Reviewers did not identify major concerns, and several suggestions for improvements have been a result of in-depth interactions between authors and reviewers, during the review process. Authors further improved this work based on the suggestions of the reviewers.